# Four decades of Antarctic surface elevation changes from multi-mission satellite altimetry

Ludwig Schröder[1,2], Martin Horwath[1], Reinhard Dietrich[1], Veit Helm[2], Michiel R. van den Broeke[3], and Stefan R. M. Ligtenberg[3]

[1]Technische Universität Dresden, Institut für Planetare Geodäsie, Dresden, Germany
[2]Alfred Wegener Institute, Helmholtz Centre for Polar and Marine Research, Bremerhaven, Germany
[3]Institute for Marine and Atmospheric Research Utrecht, Utrecht University, Utrecht, The Netherlands

*Correspondence to:* Ludwig Schröder (ludwig.schroeder@tu-dresden.de)

**Abstract.** We developed a multi-mission satellite altimetry analysis over the Antarctic Ice Sheet which comprises Seasat, Geosat, ERS-1, ERS-2, Envisat, ICESat and CryoSat-2. After a consistent reprocessing and a stepwise calibration of the inter-mission offsets, we obtain monthly grids of multi-mission surface elevation change (SEC) with respect to the reference epoch 09/2010 from 1978 to 2017. A validation with independent elevation changes from in situ and airborne observations as well as a comparison with a firn model proves that the different missions and observation modes have been successfully combined to a seamless multi-mission time series. For coastal East Antarctica, even Seasat and Geosat provide reliable information and, hence, allow to analyze four decades of elevation changes. The spatial and temporal resolution of our result allows to identify when and where significant changes in elevation occurred. These time series add detailed information to the evolution of surface elevation in key regions as Pine Island Glacier, Totten Glacier, Dronning Maud Land or Lake Vostok. After applying a density mask, we calculated time series of mass changes and find that the Antarctic Ice Sheet north of 81.5°S was losing mass at an average rate of -85±16 Gt/yr between 1992 and 2017, which accelerated to -137±25 Gt/yr after 2010.

## 1 Introduction

Satellite altimetry is fundamental for detecting and understanding changes in the Antarctic ice sheet (AIS, Rémy and Parouty, 2009; Shepherd et al., 2018). Since 1992, altimeter missions have revealed dynamic thinning of several outlet glaciers in West Antarctica and have put narrow limits on elevation changes in most parts of East Antarctica. Rates of surface elevation change are not constant in time (Shepherd et al., 2012). Ice flow acceleration has caused dynamic thinning to accelerate (Mouginot et al., 2014; Hogg et al., 2017). Variations in surface mass balance (SMB) and firn compaction rate also cause interannual variations of surface elevation (Horwath et al., 2012; Shepherd et al., 2012; Lenaerts et al., 2013). Consequently, different rates of change have been reported from altimeter missions that cover different time intervals. For example, ERS-1 and ERS-2 data over the interval 1992-2003 revealed negative elevation rates in eastern Dronning Maud Land and Enderby Land (25-60°E) and positive rates in Princess Elizabeth Land (70-100°E) (Wingham et al., 2006b), while Envisat data over the interval 2003-2010 revealed the opposite pattern (Flament and Rémy, 2012). Two large snowfall events in 2009 and 2011 have induced stepwise elevation changes in Dronning Maud Land (Lenaerts et al., 2013; Shepherd et al., 2012).

As a consequence, mean linear rates derived from a single mission have limited significance in characterizing the long-term evolution of the ice sheet (Wouters et al., 2013). Data from different altimeter missions need to be linked over a time span as long as possible in order to better distinguish and understand the long-term evolution and the natural variability of ice sheet volume and mass.

Missions with similar sensor characteristics have been combined e.g. by Wingham et al. (2006b, ERS-1 and ERS-2) and Li and Davis (2008, ERS-2 and Envisat). Fricker and Padman (2012) use Seasat, ERS-1, ERS-2 and Envisat to determine elevation changes of Antarctic ice shelves. They apply constant biases, determined over open ocean, to cross-calibrate the missions. In contrast to ocean-based calibration, Zwally et al. (2005) found significant differences for the biases over ice sheets with a distinct spatial pattern (see also Frappart et al., 2016). Khvorostovsky (2012) showed that the correction of inter-

mission offsets over an ice sheet is not trivial. Paolo et al. (2016) cross-calibrated ERS-1, ERS-2 and Envisat on each grid cell using overlapping epochs and Adusumilli et al. (2018) extended these time series by including CryoSat-2 data. We use a very similar approach for conventional radar altimeter measurements with overlapping mission periods. Moreover, we also include measurements of the non-overlapping missions Seasat and Geosat and measurements with different sensor characteristics, such as ICESat laser altimetry or CryoSat-2 interferometric Synthetic Aperture Radar (SARIn) mode, making the combination of

the observations even more challenging.

    Here we present an approach to combine seven different satellite altimetry missions over the AIS. By a refined waveform retracking and slope correction of the radar altimetry (RA) data we ensure consistency of the surface elevation measurements and improve their precision by up to 50%. In the following stepwise procedure, we first process the measurements from all missions jointly using the repeat altimetry method. We then form monthly time series for each individual mission data set.

Finally, we merge all time series from both radar and laser altimetry. For this last step, we employ different approaches of inter-mission offset estimation, depending on the temporal overlap or non-overlap of the missions and on the similarity or dissimilarity of their altimeter sensors.

    We arrive at consistent and seamless time series of gridded surface elevation differences with respect to a reference epoch (09/2010) which we made publicly available at https://doi.pangaea.de/10.1594/PANGAEA.897390. The resulting monthly

grids with a $10\,\mathrm{km}$ spatial resolution were obtained by smoothing with a moving window over three months and a spatial gaussian weighting with $2\sigma = 20\,\mathrm{km}$. We evaluate our results and their estimated uncertainties by a comparison with independent in-situ and airborne datasets, satellite gravimetry estimates, and regional climate model outputs. We illustrate that these time series of surface elevation change (SEC) allow to study geometry changes and derived mass changes of the AIS in unprecedented detail. The recent elevation changes of Pine Island Glacier in West Antarctica, Totten Glacier in East Antarctica,

and Shirase Glacier of Dronning Maud Land in East Antarctica are put in context with the extended time series from satellite altimetry. Finally, we calculate ice sheet mass balances from these data for the respectively covered regions. A comparison with independent data indicates a high consistency of the different data sets but reveals also remaining discrepancies.

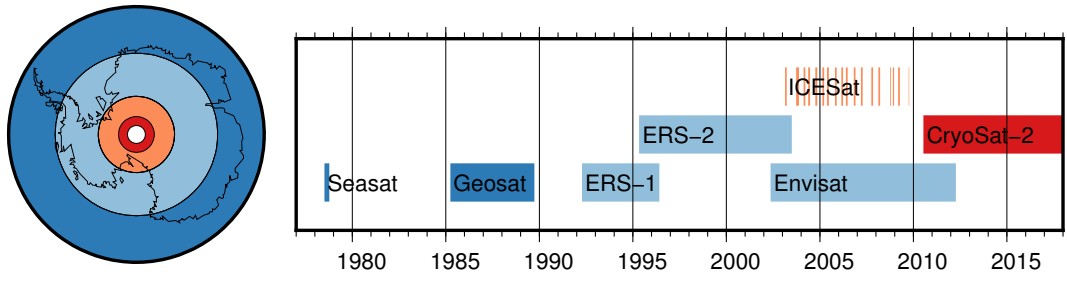

**Figure 1.** Spatial and temporal coverage of the satellite altimetry data used in this study. The colors denote the maximum southern extent of the measurements (dark blue: 72°S, light blue: 81.5°S, orange: 86°S, red: 88°S) and thus the size of the respective polar gap.

## 2  Data

### 2.1  Altimetry data

We use the ice sheet surface elevation observations from seven satellite altimetry missions: Seasat, Geosat, ERS-1, ERS-2, Envisat, ICESat and CryoSat-2. Figure 1 gives an overview over their temporal and spatial coverage. The data of the two early missions, Seasat and Geosat, were obtained from the Radar Ice Altimetry project at Goddard Space Flight Center (GSFC). For the ESA missions, we used the data of the REAPER reprocessing project (Brockley et al., 2017) of ERS-1 and ERS-2, the RA-2 Sensor and Geophysical Data Record (SGDR) of Envisat in version 2.1 and Baseline C Level 2I data of CryoSat-2. For ICESat we used GLA12 of release 633 from the National Snow and Ice Data Center (NSIDC). Further details concerning the data set versions used and the data editing criteria, applied to remove corrupted measurements in a preprocessing step are given in the supplement.

As illustrated in Fig. 1, due to the inclination of 108°, Seasat and Geosat measurements cover only the coastal regions of the East Antarctic Ice Sheet (EAIS) and the northern tip of the Antarctic Peninsula Ice Sheet (APIS) north of 72°S, which is about 25% of the total ice sheet area. With the launch of ERS-1, the polar gap was reduced to areas south of 81.5°S, resulting in a coverage of 79% of the area. The polar gap is even smaller for ICESat (86°S) and CryoSat-2 (88°S), leading to a nearly complete coverage of the AIS in recent epochs.

ERS-1 and ERS-2 measurements were performed in two different modes, distinguished by the width of the tracking time window and the corresponding temporal resolution of the recorded waveform. The ice mode is coarser than the ocean mode, in order to increase the chance of capturing the radar return from rough topographic surfaces (Scott et al., 1994). While the ice mode was employed for the majority of measurements, a significant number of observations has also been performed in ocean mode over Antarctica (22% for ERS-1, 2% for ERS-2). We use the data from both modes, as the ocean mode provides a higher precision while the ice mode is more reliable in steep terrain (see Fig. S1 and S3). However, as there is a regionally varying bias between the modes, we treat them as two separate data sets, similar to Paolo et al. (2016).

## 2.2 Reprocessing of radar altimetry

Compared to measurements over the global oceans, pulse limited radar altimetry (PLRA) over ice sheets requires a specific processing to account for the effects of topography and the dielectric properties of the surface (Bamber, 1994). To ensure consistency in the analysis of PLRA measurements, processed and provided by different institutions, we applied our own method for retracking and slope correction.

The slope correction is applied to account for the effect of topography within the beam limited footprint (Brenner et al., 1983). Different approaches exist to apply a correction (Bamber, 1994) but this effect is still a main source of error in RA over ice sheets. The "direct method" uses the surface slope within the beam limited footprint to obtain a correction for the measurement in nadir direction. In contrast, the "relocation method" relates the measurement towards the likely true position up slope. While the direct method has the advantage that the measurement location is unchanged, which allows an easier calculation of profile crossovers or repeat track parameters, the relocation method has lower intrinsic errors (Bamber, 1994). A validation using crossovers with kinematic GNSS-profiles (Schröder et al., 2017) showed that, especially in coastal regions, the direct method leads to significantly lager offsets and standard deviations, compared to the relocation method. Roemer et al. (2007) developed a refined version of the relocation method, using the full information of a digital elevation model (DEM) to locate the Point of Closest Approach (POCA) within the approximately $20\,\mathrm{km}$ beam limited footprint. We applied this method in our reprocessing chain using the DEM of Helm et al. (2014). The CryoSat-2 measurements, used for this DEM, have a very dense coverage, and hence, very little interpolation is necessary. Compared to the DEM of Bamber et al. (2009), this significantly improves the spatial consistency. We optimized the approach of Roemer et al. (2007) with respect to computational efficiency for the application over the entire ice sheet. Instead of searching the POCA with the help of a moving window of $2\,\mathrm{km}$ (which represents the pulse limited footprint) in the DEM-to-satellite grid, we applied a Gaussian filter with $\sigma$=1 km to the DEM itself to resemble the coverage of a pulse limited footprint. Hence, instead of the closest window average, we can simply search for the closest cell in the smoothed grid, which we use as the coarse POCA location. In order to achieve a sub-grid POCA location, we fit a biquadratic function to the satellite-to-surface distance within a 3x3 grid cell environment around the coarse POCA grid cell and determine the POCA according to this function.

The retracking of the return signal waveform is another important component in the processing of RA data over ice sheets (Bamber, 1994). Functional fit approaches (e.g. Martin et al., 1983; Davis, 1992; Legrésy et al., 2005; Wingham et al., 2006b) are well established and allow the interpretation of the obtained waveform shape parameters with respect to surface and sub-surface characteristics (e.g. Lacroix et al., 2008; Nilsson et al., 2015). However, the alternative approach of threshold retrackers has proven to be more precise in terms of repeatability (Davis, 1997; Nilsson et al., 2016; Schröder et al., 2017). A very robust variant is called ICE-1, using the "Offset Center of Gravity" (OCOG) amplitude (Wingham et al., 1986). Compared to the waveform maximum, the OCOG-amplitude is significantly less affected by noise (Bamber, 1994). Davis (1997) compared different retrackers and showed that a threshold based retracker produces a remarkably higher precision (especially with a low threshold as 10%), compared to functional fit based results. We implemented three threshold levels (10%, 20% and 50%) for the OCOG-amplitude, which allowed us to analyze the influence of the choice of this level similar to Davis (1997).

In addition to PLRA, we also use the SARIn mode data of CryoSat-2, reprocessed by Helm et al. (2014). The difference with respect to the processing by ESA mainly consisted in a refined determination of the interferometric phase and in the application of a threshold retracker.

## 2.3 Accuracy and precision

The accuracy of RA-derived ice surface elevation measurements has been assessed previously by a crossover comparison with independent validation data such as the ICESat laser observations (Brenner et al., 2007), airborne lidar (Nilsson et al., 2016) and ground based GNSS profiles (Schröder et al., 2017). Besides the offset due to snow pack penetration and instrumental calibration over flat terrain, these assessments revealed that with increasingly rough surface topography, the RA measurements show systematically higher elevations than the validation data. These topography related offsets can be explained by the fact

that for surfaces that undulate within the ~20 km beam-limited footprint, the radar measurements tend to refer to local topographic maxima (the POCA), while the validation data from ground-based GNSS profiles or ICESat-based profiles represent the full topography. The standard deviation of differences between RA data and validation data contains information about the measurement noise but is additionally influenced by the significantly different sampling of a rough surface as well. While over flat terrains, this standard deviation is below 50 cm for most satellite altimeter data sets, it can reach ten meters and more in

coastal regions. However, both types of error relate to the different sampling of topography of the respective observation techniques. An elevation change, detected from within the same technique, is not influenced by these effects. Hence, with respect to elevation changes, not the accuracy but the precision (i.e. the repeatability) has to be considered.

This precision can be studied using intra-mission crossovers between ascending and descending profiles. Here, the precision of a single measurement is obtained by $\sigma_H = |\Delta H|/\sqrt{2}$ as two profiles contribute to this difference. To reduce the influence

of significant real surface elevation changes between the two passes, we consider only crossovers with a time difference of less than 31 days. In stronger inclined topography, the precision of the slope correction dominates the measurement error (Bamber, 1994). Hence, to provide meaningful results, the surface slope needs to be taken into consideration. We calculate the slope from the CryoSat-2 DEM (Helm et al., 2014). The absence of slope-related effects on flat terrain allows to study the influence of the retracker (denoted as noise here). With increasing slope, the additional error due to topographic effects can be identified.

A comparison of the crossover errors of our reprocessed data and of the standard products shows significant improvements achieved by our reprocessing. Figure 2 shows this comparison for Envisat (similar plots for each data set can be found in the supplement Fig. S1), binned into groups of 0.05° of specific surface slope. The results for a flat topography show that a 10% threshold provides the highest precision,which confirms the findings of Davis (1997). For higher slopes, we see that our refined slope correction also contributed to a major improvement. A constant noise level $\sigma_{noise}$ and a quadratic, slope related term

$\sigma_{slope}$ has been fitted to the data according to $\sigma_H = \sigma_{noise} + \sigma_{slope} \cdot s^2$, where $s$ is in the unit of degrees. The results in Tab. 1 show that for each of the PLRA data sets of ERS-1, ERS-2 and Envisat, the measurement noise could be reduced by more than 50% compared to the ESA product which uses the functional fit retracker ICE-2 (Legrésy and Rémy, 1997). With respect to the CryoSat-2 standard retracker (Wingham et al., 2006a), the improvement is even larger. Improvements are also significant for the slope-related component. For the example of Envisat and a slope of 1°, the slope-related component is 1.03 m for the ESA

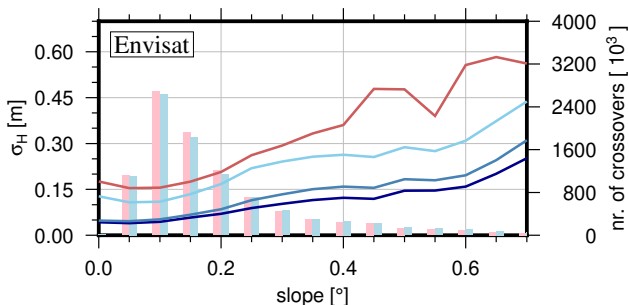

**Figure 2.** Precision of different processing versions of Envisat measurements from near time (<31 days) crossovers, binned against slope. Red curve: ESA version with ICE-2 retracker and relocated by mean surface slope. Light, medium and dark blue curves: Data reprocessed in this study with 50%-, 20%- and 10%-threshold retracker, relocated using the refined method. Vertical bars: number of crossovers for the ESA (red) and our 10% threshold retracked data (blue).

**Table 1.** Noise level and slope related component ($s$ in degrees) of the measurement precision, fitted to near time crossovers (unit: m) of the data from the respective data center and our reprocessed data (with a 10% threshold retracker applied).

| Data set | Data center | Reprocessed |
|---|---|---|
| Seasat | $0.21 + 1.91s^2$ | $0.25 + 0.70s^2$ |
| Geosat | $0.17 + 0.86s^2$ | $0.18 + 1.16s^2$ |
| ERS-1 (ocean) | $0.25 + 0.90s^2$ | $0.09 + 0.18s^2$ |
| ERS-1 (ice) | $0.36 + 2.37s^2$ | $0.17 + 0.57s^2$ |
| ERS-2 (ocean) | $0.23 + 0.75s^2$ | $0.07 + 0.14s^2$ |
| ERS-2 (ice) | $0.38 + 2.57s^2$ | $0.15 + 0.53s^2$ |
| Envisat | $0.17 + 1.03s^2$ | $0.05 + 0.37s^2$ |
| ICESat | $0.05 + 0.25s^2$ | |
| CryoSat-2 (LRM) | $0.18 + 2.46s^2$ | $0.03 + 1.06s^2$ |
| CryoSat-2 (SARIn) | $0.38 + 2.01s^2$ | $0.11 + 0.79s^2$ |

Note that the slope dependent component is weakly determined for data sets with a poor tracking in rugged terrain such as Seasat, Geosat or the ERS ocean mode and for the LRM mode of CryoSat-2.

product and only $0.37\,\mathrm{m}$ for the reprocessed data. The advanced interferometric processing of the SARIn data achieved similar improvements. For the two early missions Seasat and Geosat, the crossover error of our reprocessed profiles is similar to that of the original dataset from GSFC. However, the number of crossover points is significantly increased, especially for Geosat (see Fig. S1). This means that our reprocessing obtained reliable data where the GSFC processor rejected the measurements.

5    In addition to measurement noise, reflected in the crossover differences, a consistent pattern of offsets between ascending and descending tracks has been observed previously (A-D bias, Legrésy et al., 1999; Arthern et al., 2001). Legrésy et al. (1999)

interpret this pattern as an effect of the interaction of the linearly polarized radar signal with wind-induced surface structures, while Arthern et al. (2001) attribute the differences to anisotropy within the snowpack. Helm et al. (2014) showed that a low threshold retracker significantly reduces the A-D bias. We observe a similar major reduction (from ±1 m in some regions for a functional fit retracker to ±15 cm when using a 10% threshold, see Fig. S2). The remaining bias is not larger, in its order of magnitude, than the respective noise. Moreover, near the ice sheet margins, the determination of meaningful A-D biases is complicated by the broad statistical distribution of A-D differences and the difficulty to discriminate outliers. We therefore do not apply a systematic A-D bias as a correction but rather include its effect in the uncertainty estimate of our final result.

## 3   Multi-mission SEC time series

### 3.1   Repeat track parameter fit

We obtain elevation time series following the repeat track approach, similar to Legrésy et al. (2006) and Flament and Rémy (2012). As the orbits of the missions used here have different repeat track patterns, instead of along-track boxes we perform our fit on a regular grid with 1 km spacing (as in Helm et al., 2014). For each grid cell we analyze all elevation measurements $h_i$ within a radius of 1 km around the grid cell center. This size seems reasonable as for a usual along track spacing of about 350 m for PLRA (Rémy and Parouty, 2009), each track will have up to 5 measurements within the radius. Due to the size of the pulse limited footprint a smaller search radius would contain only PLRA measurements with very redundant topographic information and thus would not be suitable to fit a reliable correction for the topography. As specified in Eq. (1), the parameters contain a linear trend ($dh/dt$), a planar topography ($a_0, a_1, a_2$) and a regression coefficient ($dBS$) for the anomaly of backscattered power ($bs_i - \overline{bs}$) to account for variations in the penetration depth of the radar signal.

For a single mission, the parameters are adjusted according to the model

$$
\begin{aligned}
h_i = \quad & dh/dt(t_i - t_0)+ \\
& a_0 + a_1 x_i + a_2 y_i + \\
& dBS(bs_i - \overline{bs})+ \\
& res_i
\end{aligned}
\tag{1}
$$

Here, $t_i$ denotes the time of the observation. The reference epoch $t_0$ is set to 09/2010. $x_i$ and $y_i$ are the Polar Stereographic coordinates of the measurement location, reduced by the coordinates of the cell's center. The residual $res_i$ describes the misfit between the observation and the estimated parameters.

To account for varying penetration depths due to variations in the electromagnetic properties of the ice sheet surface, different approaches exist. Wingham et al. (1998), Davis and Ferguson (2004), McMillan et al. (2014) or Zwally et al. (2015) apply a linear regression using the backscattered power. Flament and Rémy (2012), Michel et al. (2014) or Simonsen and Sørensen (2017) use two additional waveform shape parameters, obtained from functional fit retrackers. Nilsson et al. (2016) showed, that a low threshold retracker mitigates the need for a complex waveform shape correction. Hence, we decided to use a solely backscatter-related correction.

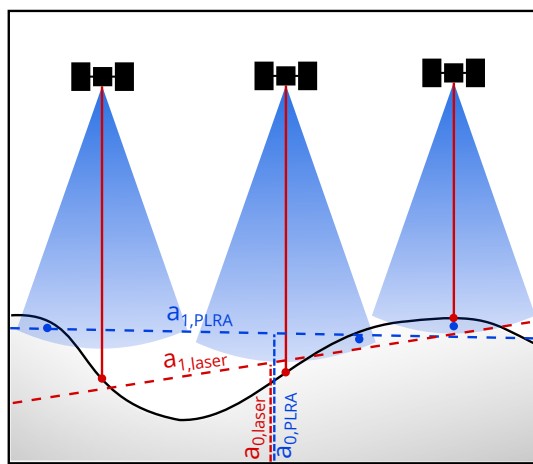

**Figure 3.** Illustration of the technique-dependent topographic sampling. The laser (red) measures the surface elevation in the nadir of the instrument while for radar altimetry (blue), the first return signal originates from the POCA (marked by the blue point). Hence, planar surface approximations to the measured heights (dashed lines) as in Eq. (2) are intrinsically different for the different techniques.

Besides the parameters in Eq. (1), McMillan et al. (2014) and Simonsen and Sørensen (2017) estimate an additional orbit direction related parameter to account for A-D biases. In Sect. 2.3 we showed that these biases are significantly reduced due to the reprocessing with a low threshold retracker. A further reduction of possible remaining artifacts of A-D biases is achieved by the smoothing in the merging step in Sect. 3.3.3. The weighted averaging of the results over a diameter of 60 km leads to a

balanced ratio of ascending and descending tracks. Our choices concerning the correction for local topography, time-variable penetration effects and A-D biases were guided by the principle to prefer the simplest viable model in order to keep the number of parameters small compared to the number of observations.

In contrast to this single mission approach, here we perform a combined processing of all data from different missions and even different altimeter techniques. Thus, some of the parameters may vary between the data sets. To allow for offsets between

the missions, the elevation at the cell center $a_0$ is fitted for each mission individually. The same applies to $dBS$, which might relate to specific characteristics of a mission as well. For Seasat, covering less than 100 days, this parameter is not estimated, as we assume that during the mission life time no significant changes occurred. For ICESat, $dBS$ is not estimated either, as signal penetration is negligible for the laser measurements.

Between different observation techniques (i.e. PLRA, SARIn and laser altimetry), the effective surface slope may also differ.

Considering the specific footprint sizes and shapes, the topography is sampled in a completely different way as illustrated in Fig. 3. While PLRA refers to the closest location anywhere within the ~20 km beam-limited footprint (i.e. the POCA), CryoSat's SARIn measurement can be attributed within the narrow Doppler stripe in cross-track direction. For ICESat the ~70 m laser spot allows a much better sampling of local depressions. Hence, the slope parameters $a_1$ and $a_2$ are estimated for each of the techniques independently.

Considering these sensor-specific differences, the model for the least squares adjustment in Eq. (1) is extended for multi-mission processing

$$
\begin{aligned}
h_i = \; & dh/dt(t_i - t_0) + \\
& a_{0,M(i)} + a_{1,T(i)}x_i + a_{2,T(i)}y_i + \\
& dBS_{M(i)}(bs_i - \overline{bs}_{M(i)}) + \\
& res_i
\end{aligned}
\tag{2}
$$

where $M(i)$ and $T(i)$ denote to which mission or technique, the measurement $h_i$ belongs.

We define a priori weights for the measurements $h_i$ based on the precision of the respective mission and mode from crossover analysis (Tab. 1) and depending on the surface slope at the measurement location. This means that in regions with a more distinctive topography, ICESat measurements (with a comparatively low slope-dependent error component) will obtain stronger weights, compared to PLRA as Envisat. Over regions of flat topography, such as the interior of East Antarctica, the weights between PLRA and ICESat are comparable

In order to remove outliers from the data and the results we apply different outlier filters. After the multi-mission fit, we screen the standardized residuals (Baarda, 1968) to exclude any $res_i$ that exceed five times its a posteriori uncertainty. We iteratively repeat the parameter fit until no more outliers are found. Furthermore, in order to exclude remaining unrealistic results from further processing, we filter our repeat track cells and reject any results where we obtain an absolute elevation change rate $|dh/dt|$ which is larger than $20\,\mathrm{m/yr}$ or where the standard deviation of this rate is higher than $0.5\,\mathrm{m/yr}$.

## 3.2   Single-mission time series

After fitting all parameters according to the multi-mission model (Eq. 2), we regain elevation time series by recombining the parameters $a_0$ and $dh/dh$ with monthly averages of the residuals ($\overline{res}$). For each month $j$ and each mission $M$, the time series are constructed as

$$
h_{j,M} = a_{0,M} + dh/dt(t_j - t_0) + \overline{res}_{j,M}.
\tag{3}
$$

This recombination of parameters from Eq. (2) and averages of residuals does not include the parameters of topography slope and backscatter regression. Hence, each time series of $h_{j,M}$ relates to the cell center and is corrected for time-variable penetration effects. Due to the reference elevation $a_{0,M}$, which may also contain the inter-mission offset, the penetration depth and a component of the topography sampling within the cell, this results in individual time series for each single mission. A schematic illustration of the results of this step is given in Fig. 5a. The temporal resolution of these time series is defined by

using monthly averages of the residuals. With typical repeat cycle periods of 35 days or more, these $\overline{res}$ represent the anomalies of typically a single satellite pass towards all parameters including the linear rate of elevation change. The standard deviation of the residuals in these monthly averages are used as uncertainty measure for $h_{j,M}$ (see C.2 for further details).

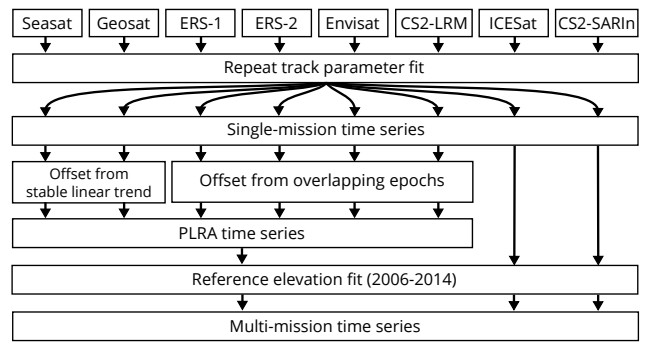

**Figure 4.** Schematic diagram of the processing steps from the combined repeat track parameter fit over single-mission time series towards a combined multi-mission time series.

## 3.3 Combination of the single-mission time series

In order to merge data from different missions into a joint time series, inter-mission offsets have to be determined and eliminated. In the ERS reprocessing project (Brockley et al., 2017), mean offsets between the ERS missions and Envisat have been determined and applied to the elevation data. However, for ice sheet studies inter-satellite offsets are found to be regionally

varying (Zwally et al., 2005; Thomas et al., 2008; Khvorostovsky, 2012). When merging data from different observation techniques (PLRA, SARIn and laser) the calibration gets even more challenging. We chose an approach in different steps which is depicted in Figs. 4 and 5. The following section gives an overview and explains the different steps to merge the single mission time series. A detailed description of the parameters used in each step can be found in the supplement.

### 3.3.1 Merging PLRA time series

In a first step, we merge the PLRA time series. For these missions the topographic sampling by the instruments is similar and thus the offsets are valid over larger regions. For overlapping missions (ERS-1, ERS-2, Envisat, CryoSat-2 LRM) the offsets are calculated from simultaneous epochs (blue area in Fig. 5b), as performed by Wingham et al. (1998) or Paolo et al. (2016). Smoothed grids of these offsets are generated, summed up if necessary to make all data sets comparable with Envisat (see Fig. S4) and applied to the respective missions. For the ERS missions, we find significant differences in the offsets for

ice and ocean mode, hence, we determine separate offsets for each mode. Comparing our maps with similar maps of offsets between ERS-2 (ice mode) and Envisat shown by Frappart et al. (2016) reveals that the spatial pattern agrees very well but we find significantly smaller amplitudes. We interpret this as a reduced influence of volume scattering due to our low retracking threshold. In accordance with Zwally et al. (2005), we did not find an appropriate functional relationship between the offset and the waveform parameters.

To calibrate Geosat and Seasat, a gap of several years without observations has to be bridged. As depicted by the dashed blue lines in Fig. 5b, we do this using the trend corrected reference elevations $a_{0,M}$ from the joint fit in Eq. (2). This, however, can only be done if the rate is sufficiently stable over the whole period. Therefore, we use two criteria. First, we check the

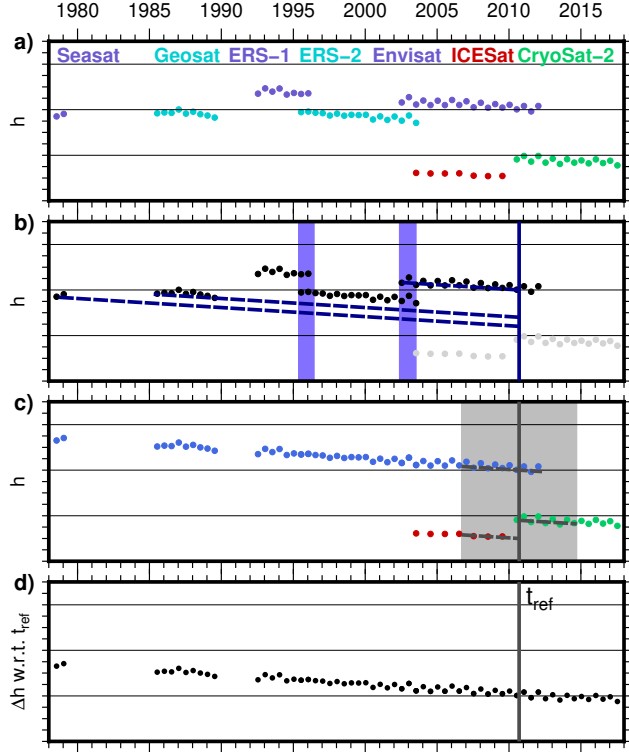

**Figure 5.** Schematic illustration of the combination of the missions. **a)** Single-mission time series of PLRA missions (blue and cyan), CryoSar-2 in SARIn mode (green) and the laser altimetry measurements of ICESat (red) with inter-mission offsets. **b)** Offsets between the PLRA data are determined from overlapping epochs (blue area) or trend-corrected elevation differences (according to Eq. 2) where $dh/dt$ is sufficiently stable. **c)** The specific offset between PLRA, SARIn and laser data depends on the sampling of the topography within each single cell. These different techniques are aligned by reducing each elevation time series by the specific elevation at the reference epoch $t_{ref}$. Due to possible non-linear surface elevation changes, this reference elevation is obtained from a 8-year interval only (gray area). **d)** The combined multi-mission time series contains SECs with respect to $t_{ref}$.

standard deviation of the fit of $dh/dt$. This $\sigma_{dh/dt}$ indicates the consistency of the observations towards a linear rate during the observational period. However, anomalies during the temporal gaps between the missions (i.e. 1978-1985 and 1989-1992) cannot be detected in this way. Therefore, we furthermore utilize a firn densification model (FDM, Ligtenberg et al., 2011; van Wessem et al., 2018). This model describes the anomalies in elevation due to atmospheric processes against the long-term mean. The RMS of the FDM time series is hence a good measure for the magnitude of the non-linear variations in surface elevation. Consequently, only cells where $\sigma_{dh/dt} < 1\,\mathrm{cm/yr}$ and $RMS_{FDM} < 20\,\mathrm{cm}$, indicating a highly linear rate, are used to calibrate the two historic missions. Maps of the offsets with respect to Envisat are shown in the supplement Fig. S5. The FDM criterion is not able to detect changes in ice dynamics. However, as regions where both stability criteria are fulfilled are mainly found on the plateau where flow velocity are below $30\,\mathrm{m/yr}$ (Rignot et al., 2017), we expect no significant non-linear elevation changes due to ice flow. The mean of the offsets over all cells amounts to -0.86 m for Seasat and -0.73 m for Geosat.

The corresponding standard deviations of 0.85 m and 0.61 m are mainly a result of the regional pattern of the offsets. The true offsets are likely to have spatial variations. However, we are not able to distinguish spatial variations of the offset from residual effects of temporal height variations in the regions meeting the stability criterion. In the regions not meeting this criterion, we are not able to estimate the spatial variations of the offset at all. Therefore, our final estimate of the offset, applied to

the measurements, is a constant, calculated as the average offset over the regions meeting the stability criterion. The spatial variability not accounted for by the applied offset is included, instead, in the assessed uncertainty. Our bias between Seasat and Envisat (-0.86±0.85 m) agrees within uncertainties with the ocean-based bias of -0.77 m used by Fricker and Padman (2012). However, we prefer the offset determined over the ice sheet because this kind of offsets may depend on the reflecting medium (see Sect. C.2.2 for a more detailed discussion).

With the help of these offsets, all PLRA missions were corrected towards the chosen reference mission Envisat. Uncertainty estimates of the offsets are applied to the respective time series to account for the additional uncertainty. Hence, the PLRA time series are combined (blue in Fig. 5c with additional CryoSat-2 LRM mode where available). At epochs when more than one data set exists, we apply weighted averaging using the uncertainty estimates.

### 3.3.2  Technique-specific surface elevation changes

In contrast to the PLRA data in the previous step, when merging data from different observation techniques such as CryoSat's SARIn mode, ICESat's laser observations and PLRA, also the different sampling of topography has to be considered. As noted in Sect. 3.1 this might lead to completely different surfaces fitted to each type of elevation measurements and the time series need to be calibrated for each cell individually. However, not all cells have valid observations of each data set. Therefore, instead of calibrating the techniques against each other, we reduce each time series by their elevation at a common reference

epoch and hence obtain time series of surface elevation changes (SEC) w.r.t. this reference epoch instead of absolute elevation time series. This step eliminates offsets due to differences in firn penetration or due to the system calibration between the techniques as well.

We chose September 2010 as the reference epoch. This epoch is covered by the observational periods of PLRA and CryoSat SARIn and also is exactly one year after the last observations of ICESat, which reduces the influence of an annual cycle.

As discussed in Sect. 3.3.1, non-linear elevation changes will adulterate $a_0$ from Eq. (2), obtained over the full time span. Therefore, we applied another linear fit to a limited time interval of 8 years only (09/2006-09/2014, gray area in Fig. 5c). We subtract the variation of the FDM over this period to account for short-term variations. The limited time interval reduces the influence of changes in ice dynamics. We estimate the individual reference elevations $a_{0,T}$ for each technique $T$ and a joint $dh/dt$. After subtracting the technique-specific reference elevations $a_{0,T}$ from the respective time series, they all refer to

09/2010 and can be combined.

### 3.3.3  Merging different techniques

We perform the final combination of the techniques using a weighted spatio-temporal averaging with $10\,\mathrm{km}\,\sigma$ gaussian weights in spatial domain (up to a radius of $3\sigma = 30\,\mathrm{km}$) and over 3 epochs (i.e. including the two consecutive epochs) in the temporal

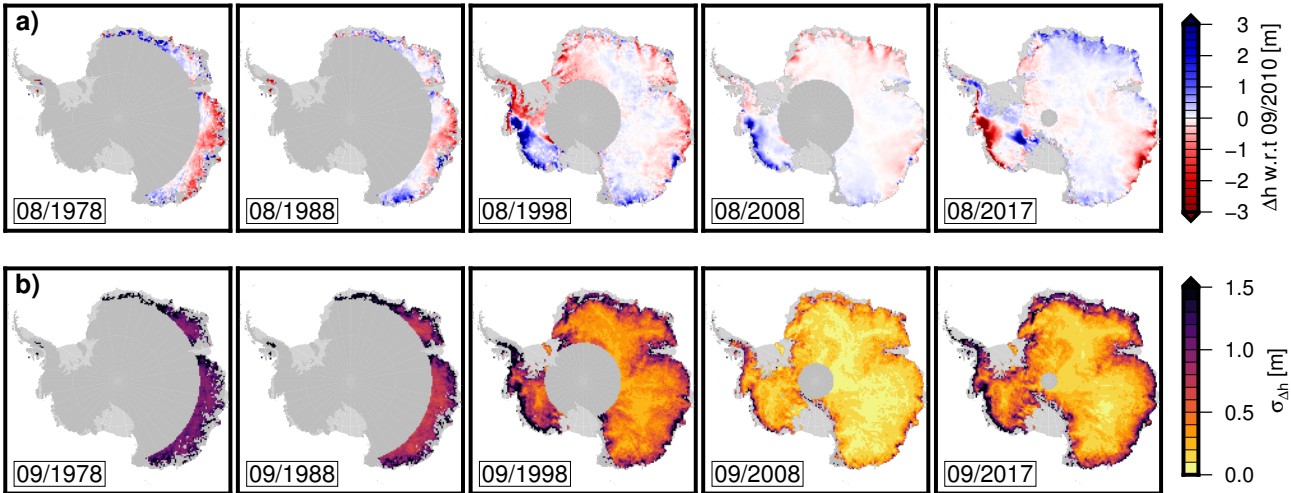

**Figure 6.** Five example snapshots of the resulting combined surface elevation time series (**a**) and their corresponding uncertainty (**b**). The height differences refer to our reference epoch 09/2010.

domain. Hence, we obtain grids of surface elevation change (SEC) with respect to 09/2010 for each month observed. Due to the smoothing of the weighting function, we reduce our spatial SEC grid resolution to $10 \times 10$ km. The respective uncertainties are calculated according to the error propagation. To avoid extrapolation and to limit this merging step to the observed area only, we calculate a value for an epoch in the $10 \times 10$ km grid cells only if we have data within 20 km around the cells center (which

is about the size of a beam-limited radar footprint). The five examples in Fig. 6 demonstrate the spatio-temporal coverage of the resulting SEC grids at different epochs. The corresponding uncertainty estimates, given in Fig. 6b (further details in the supplement) reach values of one meter and more. Besides the measurement noise and the uncertainties of the offsets, these uncertainty estimates contain a further component from the weighted averaging. In regions with large variations in $\Delta h$ over relatively short spatial scales (such as at fast-flowing outlet glaciers), such variations can add a significant contribution to $\sigma_{\Delta h}$.

As the magnitude of $\Delta h$ grows with the temporal distance to the reference epoch, the largest contributions to $\sigma_{\Delta h}$ can be expected for the earliest epochs. This also explains why the epoch 09/2008 provides the lowest uncertainty estimate in these examples, even lower than the CryoSat-2 based epoch 09/2017.

## 4   Comparison of SEC with independent data

### 4.1   In situ and airborne observations

To validate our results, we used inter-profile crossover differences of 19 kinematic GNSS profiles (Schröder et al., 2017, available at https://doi.org/10.1594/PANGAEA.869761) and elevation differences from Operation IceBridge (OIB ATM L4, Studinger, 2014). The ground based GNSS profiles were observed between 2001 and 2015 on traverse vehicles of the Russian Antarctic Expedition and most of them cover more than 1000 km. The accuracy of these profiles has been determined in

Schröder et al. (2017) to 4-9 cm. One profile (K08C) has not been used due to poorly determined antenna height offsets. For each crossover difference between kinematic profiles from different years, we compare the differences of the corresponding altimetric SEC epochs in this location ($\delta\Delta h = \Delta h_{KIN} - \Delta h_{ALT}$). The same analysis has been performed with the elevation changes obtained from differences of measurements of the scanning laser altimeter (Airborne Topographic Mapper, ATM) of

OIB. As described by Studinger (2014), the Level 4 $\Delta h$ product is obtained by comparing planes fitted to the laser scanner point clouds. The flights, carried out between 2002 and 2016, were strongly concentrated along the outlet glaciers of West Antarctica and the Antarctic Peninsula. Hence, they cover much more rugged terrain, which is more challenging for satellite altimetry. Over the tributaries of the Amundsen Sea glaciers and along the polar gap of ICESat, some repeated measurements have also been performed over flat terrain. The accuracy of these airborne measurements has been validated e.g. near summit

station in Greenland. Brunt et al. (2017) used ground based GNSS profiles of snowmobiles for this task and obtained offsets in the order of only a few centimeters and standard deviations between 4 and 9 cm.

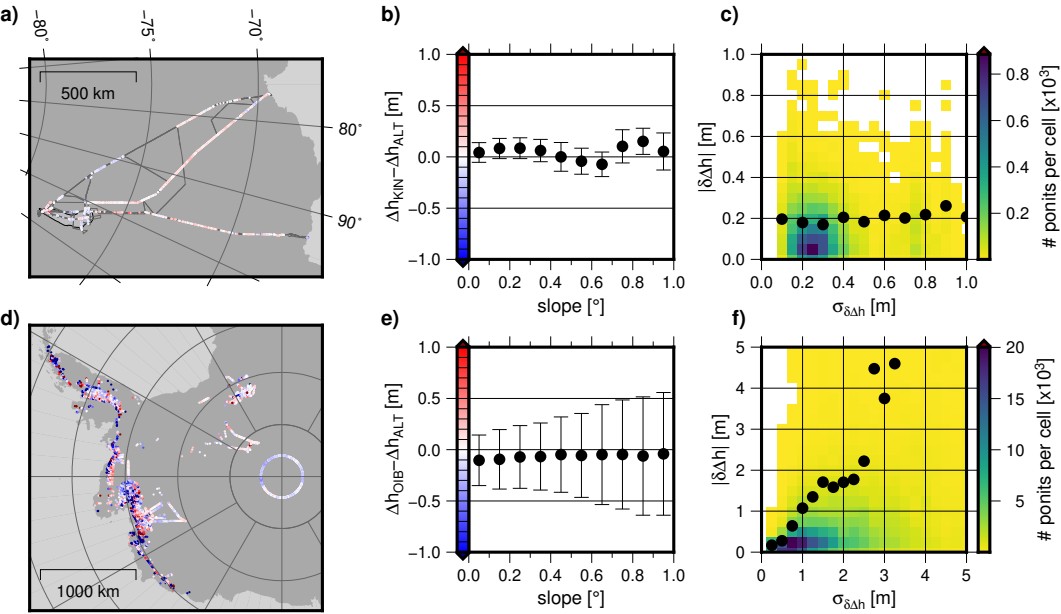

**Figure 7.** Validation with elevation differences observed by kinematic GNSS between 2001-2015 (**a,b,c**) and Operation IceBridge between 2002-2016 (**d,e,f**). Differences between elevation changes observed by the validation data and altimetry are shown on the maps (**a,d**, color scale in **b,e**). Median and MAD of these differences, binned by different surface slope, are shown in the center (**b,e**). The right diagrams (**c,f**) show the comparison of these differences with the respective uncertainty estimate, obtained from both data sets. The point density is plotted from yellow to blue and the black dots show the root mean square, binned against the estimated uncertainty.

Figures 7a and d show the results of our validation (more detailed maps for several regions at Fig. S6). A satellite calibration error would lead to systematic biases between the observed elevation differences if $\Delta h_{ALT}$ is obtained from data of two different missions. However, such biases may also be caused by systematic errors in the validation data. Furthermore, in

contrast to the calibration data, the RA measurements may systematically miss some of the most rapid changes if those are

located in local depressions (Thomas et al., 2008). With an overall median difference of 6±10 cm for the GNSS profiles and -9±42 cm for OIB, however, the observed elevation changes show only moderate systematic effects and agree within their error bars. The median absolute deviation (MAD) for different specific surface slopes (Fig. 7b and e) reveal the influence of topography in this validation. The GNSS profiles show only a very small increase of this variation with slope. The IceBridge data covers the margins of many West Antarctic glaciers, where elevation changes differ over relatively short distances. Hence, it is not surprising that we see a significantly larger spread of $\delta\Delta h$ at higher slopes here. However, also in the less inclined regions, the MAD of the differences is still at a level of 25 cm, which is significantly larger than in the comparison with the GNSS profiles. This large spread for regions with low slopes originates mainly from the tributaries of Pine Island Glacier, where many campaigns of OIB are focused (see Fig. 7d and Fig. S6d for details). While still relatively flat, the surface elevation in this area is comparatively low, which leads to a stronger influence of precipitation (Fyke et al., 2017). This induces higher short-term variations in surface elevation, which might explain the higher differences between the IceBridge results and our 3-month temporally smoothed altimetry data. In contrast, the differences around the South Pole or in Queen Elizabeth Land (see also Fig. S6c) are significantly smaller. For the 2016 campaign of OIB, Brunt et al. (2018) furthermore report a spurious elevation variation of 10 to 15 cm across the wide-scan ATM swath which indicate a bias in the instrumental tilt angle. This could explain the systematic differences along the 88°S circle profiles where this campaign is involved.

The observed $\delta\Delta h$ can further be used to evaluate the uncertainty estimates. In Fig. 7c and f, the uncertainty estimates of the four contributing data are combined and compared to the observed differences. The comparison with both validation data sets supports that the uncertainty estimates are reasonable. For $\Delta h_{ALT}$ we expect higher errors in coastal regions due to the increased uncertainty of the topographic correction in radar altimetry. A similar relation to topography is expected for $\Delta h_{OIB}$ due to the plane fit to the ATM point cloud but also surface roughness and crevassing play an important role here. In contrast, the errors of the GNSS-derived $\Delta h_{KIN}$ are almost independent of topography. Instead, $\Delta h_{KIN}$ tends to be more uncertain on the plateau, where the soft snow causes large variations of the subsidence of the vehicles into the upper firn layers. The relatively low differences in $\delta\Delta h$ even in regions that imply a higher uncertainty, are likely just incidental for the small sample of validation data along the GNSS profiles.

In conclusion, this validation shows that remaining systematic biases (originating from satellite altimetry or the validation data) are less than a decimeter in the observed regions and that our uncertainty estimate is realistic. However, only altimetric SEC within the interval 2001-2016 can be validated in this way. For the earlier missions, no spatially extensive high precision in situ data are publicly available.

## 4.2 Firn model

Another data set, which covers almost the identical spatial and temporal range as the altimetric data, is the IMAU Firn Densification Model (FDM, Ligtenberg et al., 2011), forced at the upper boundary by accumulation and temperature of the Regional Atmospheric Climate Model, version 2.3p2 (van Wessem et al., 2018). The IMAU-FDM has been updated to the period 1979-2016, modeling the firn properties and the related surface elevation changes on a 27 km grid. However, as the FDM contains elevation anomalies only, any long-term elevation trend over 1979-2016, e.g. due to changes in precipitation on longer time

scales (as e.g. observed in some regions of West Antarctica, Thomas et al., 2015) would not be included in the model. Furthermore, due to the nature of the model, it cannot give information about ice dynamic thinning/thickening. Hence, to compare the FDM and the SEC from altimetry, we first remove a linear trend from both data sets respectively. This is performed for the period 1992-2016 (depicted in Fig. S7). The trends are only calculated from epochs where both data sets have data, i.e. in the polar gap this comparison is limited to 2003-2016 or 2010-2016, depending on the first altimetry mission providing data here. After the detrending, the anomalies are used to calculate correlation coefficients for each cell, depicted in Fig. 8a. Figure 8b shows the average magnitude of the seasonal and interannual variations (non-linear SEC), calculated as the RMS of the anomalies from the altimetry data. Comparing the two maps shows that the correlation is around 0.5 or higher, except in regions where the magnitude of the anomalies is small (i.e. where the signal-to-noise ratio of the altimetric data is low) and where large accelerations in ice velocity are observed (such as near the grounding zone of Pine Island Glacier). The relationship between the correlation coefficient and the magnitude of the non-linear SEC is depicted in Fig. 8c, where we see that for the vast majority of cells the correlation is positive. For anomalies with a non-linear SEC > 0.5 m, the average correlation is between 0.3 and 0.6.

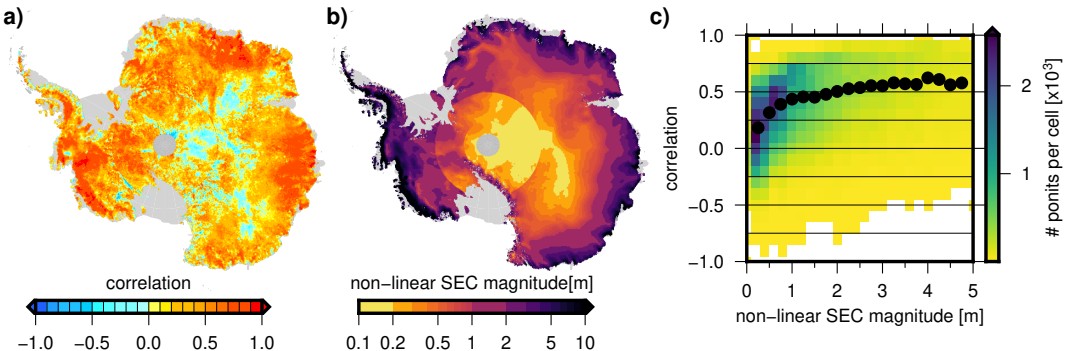

**Figure 8. a)** Correlation coefficient between the SEC anomalies of the altimetry grids and the FDM over 1992-2016 after detrending. **b)** average magnitude of anomalies of the altimetry time series. **c)** Correlation coefficient plotted against the non-linear SEC. The point density is color coded from yellow to blue. The black dots show the binned mean values.

Anomalies against the simultaneously observed long-term trend (1992-2016) can also be computed for earlier epochs. Assuming no significant changes in ice dynamics here, these anomalies allow a comparison of Geosat and Seasat with the FDM. The median difference between the anomalies according to Geosat and the anomalies according to the FDM amounts to 0.12±0.21 m (see Fig. S8). Considering that this difference is very sensitive to extrapolating the long-term trends, this is a remarkable agreement. With a median of 0.26±0.32 m, the difference between anomalies from Seasat and from the FDM is larger, but this comparison is also more vulnerable to potential errors due to the extrapolation. As the FDM starts in 1979 while Seasat operated in 1978, we compare the Seasat data with the FDM anomalies from the respective months of 1979, which might impose additional differences. Finally, the FDM model has its own inherent errors and uncertainties. Therefore, only part of the differences originates from errors in the altimetry results.

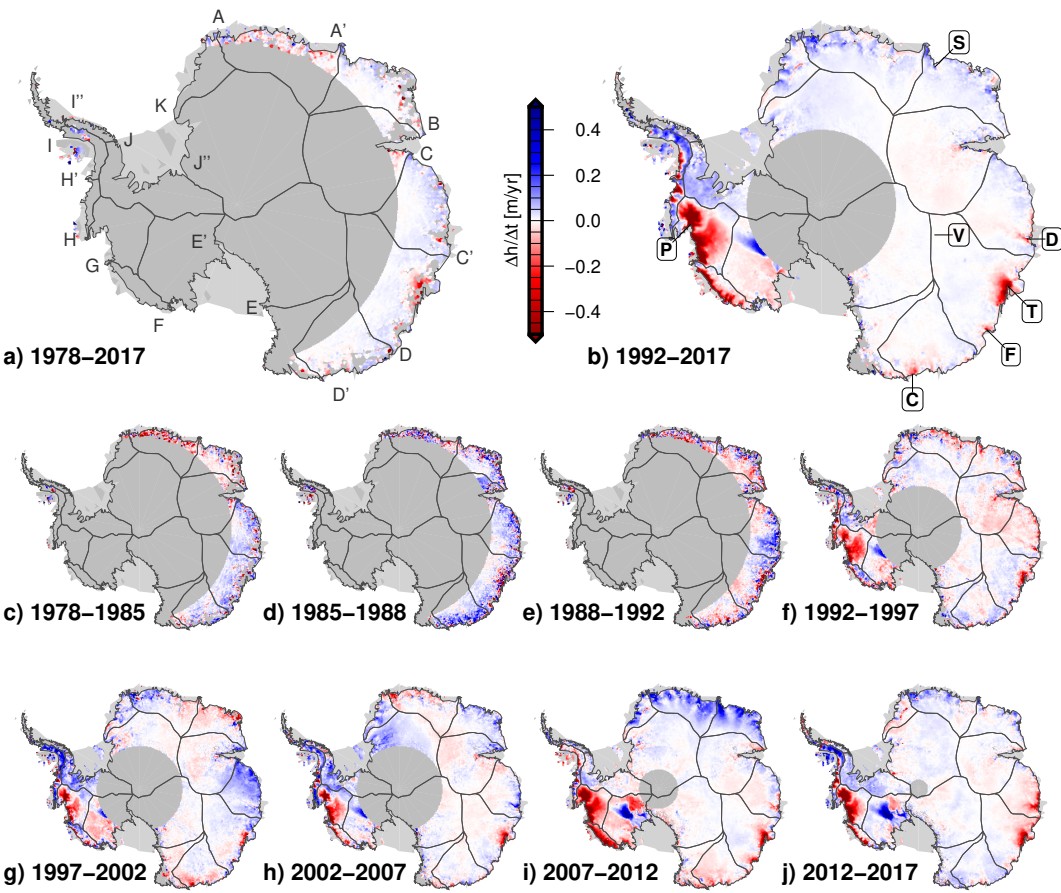

**Figure 9.** Multi-mission surface elevation change from the combined SEC time series over different time intervals. **a** and **b**) The long-term surface elevation change between 1978 and 2017 and 1992 and 2017 for the respectively covered area. **c-j**) Elevation change over consecutive time intervals reveal the interannual variability. Thin lines mark the drainage basin outlines, denoted in **a**. Bold letters in boxes in **b** denote areas mentioned in the text and in Fig. 10.

## 5    Results

### 5.1    Surface elevation changes

The average rates of elevation change over different time intervals of our multi-mission time series are shown in Fig. 9. To calculate these rates, we first averaged the data over the first year and the last year of the interval to reduce the noise, then subtracted the respective averages from each other and finally divided these differences by their time difference in years. If one of the years does not cover the full annual cycle, we calculate the average only from the months covered in both years (July-October for 1978-2017, April-December for 1992-2017). We calculate the SEC rate from epoch differences instead of fitting a rate to all epochs because the first observations at specific latitudes start in different years, the observations have different

precisions and the large gap between 1978 and 1985 is not covered by observations at all. These three points would lead to a bias towards the later epochs in a fit, so that the rates would not be representative for the true average elevation change over the full interval.

The long-term elevation changes over 25 years (Fig. 9b) show the well known thinning in the Amundsen Sea Embayment
and at Totten Glacier, as well as the thickening of Kamb Ice Stream (cf. e.g. Wingham et al., 2006b; Flament and Rémy, 2012; Helm et al., 2014). In contrast, 60% of East Antarctica north of 81.5°S shows surface elevation changes of less than ±1 cm/yr. Several coastal regions of the EAIS, however, show significant elevation changes. Totten Glacier (T in Fig. 9b) is thinning at an average rate of 72±18 cm/yr at the grounding line (cf. Fig. 10b). Several smaller glaciers in Wilkes Land also show a persistent thinning. We observe SEC rates of -26±10 cm/yr at Denman Glacier (D), -41±19 cm/yr at Frost Glacier (F) and
-33±12 cm/yr near Cook Ice Shelf (C). Rignot (2006) showed that the flow velocity of these glaciers, which are grounded well below sea level, was above the balance velocity for many years. Miles et al. (2018) analyzed satellite images since 1973 and found that the flow velocity of Cook Glacier has significantly accelerated since then. In contrast, the western sector of the EAIS (Coats Land, Dronning Maud Land and Enderby Land; basins J"-B) shows thickening over the last 25 years at rates of up to a decimeter per year.

Comparing the long-term elevation changes over 40 years (Fig. 9a) with those over 25 years shows the limitations of the
early observations, but also the additional information they provide. There were relatively few successful observations at the very margins. However, for Totten and Denman Glaciers, the 40-year rates at a distance of approximately 100 km inland from the grounding line are similar to the rates over the 1992–2017 interval, which indicates a persistent rate of thinning. Another benefit of our merged time series is that they allow to calculate rates over any sub-interval, independent of mission periods as
demonstrated in Figs. 9c-j. For most of the coastal regions of the AIS, these rates over different intervals reveal that there is significant interannual variation. Such large scale fluctuations in elevation change have been previously reported by Horwath et al. (2012) or Mémin et al. (2015) for the Envisat period. Our combined multi-mission time series now allow a detailed analysis of such signals on a temporal scale of up to 40 years.

Four examples for elevation change time series in the resulting multi-mission SEC grids are shown in Fig. 10 (coordinates
in Tab. S2). Pine Island Glacier (PIG) is located in the Amundsen Sea Embayment, which is responsible for the largest mass losses of the Antarctic Ice Sheet (e.g. McMillan et al., 2014). In East Antarctica, the largest thinning rates are observed at Totten Glacier. The region of Dronning Maud Land and Enderby Land in East Antartica has been chosen as an example for interannual variation. Here, Boening et al. (2012) reported two extreme accumulation events in 2009 and 2011, which led to significant mass anomalies. We chose a profile at Shirase Glacier as an example for this region. In contrast to the previous
locations, a very stable surface elevation has been reported for Lake Vostok (e.g. Richter et al., 2014). This stability, however, has been a controversial case recently (Zwally et al., 2015; Scambos and Shuman, 2016; Richter et al., 2016). Therefore, our results in this region shall add further evidence to pinpoint the changes there.

For Pine Island Glacier (Fig. 10a), we observe a continuous thinning over the whole observational period since 1992 (Seasat and Geosat measurements did not cover this region). Close to the grounding line (point D) the surface elevation decreased
by -45.8±7.8 m since 1992, which means an average SEC rate of -1.80±0.31 m/yr. The time series reveals that this thinning

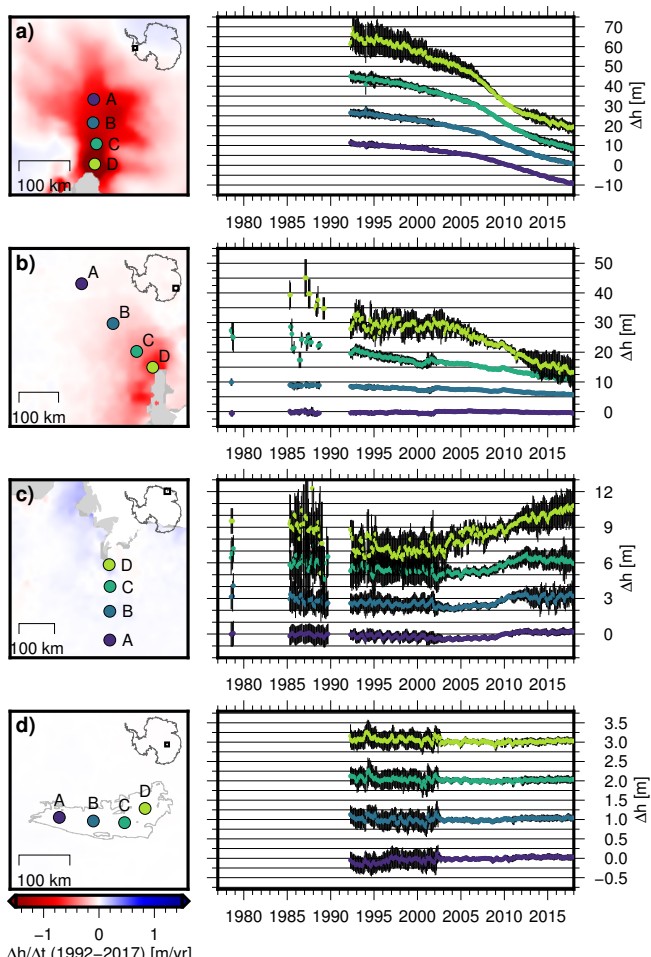

**Figure 10.** Multi-mission SEC time series in 4 selected regions (**a**) Pine Island Glacier, **b**) Totten Glacier, **c**) Shirase Glacier in Dronning Maud Land and **d**) Lake Vostok (marked by P, T, S and V in Fig. 9b). The time series of point B, C and D are shifted along $\Delta h$ for better visibility and the one $\sigma$ uncertainty range displayed in black. The maps on the left show the elevation change rate between 1992 and 2017 as in Fig. 9b (but in a different color scale).

was not constant over time, but accelerated significantly around 2006. The mean rate at D over 1992-2006 of -1.32±0.66 m/yr increased to -4.17±1.67 m/yr over 2007-2010. After 2010, the thinning rates near the grounding line decelerate again and for the period 2013-2017, the rate at D of -1.31±0.80 m/yr is very close to the rate preceding the acceleration. Also at greater distances from the grounding line (B at 80 km, A at 130 km) we observe an acceleration of the prevailing rates around 2006

5  (-0.44±0.15 m/yr over 1992-2006, -1.20±0.10 m/yr over 2006-2017 at A). In contrast to the points near the grounding line, further inland the thinning did not decelerate so far and is still at a high level. Hence, for the most recent period (2013-2017) the elevation at all points along the 130 km of the main flow line is decreasing at very similar rates. A similar acceleration of the elevation change rate near the grounding line, followed by slowdown, is observed by (Konrad et al., 2016). The onset

of this acceleration coincides with the detaching of the ice shelf from a pinning point (Rignot et al., 2014). For the time after 2009, Joughin et al. (2016) report relatively little grounding line migration, resulting in a leveling off of the ice flow velocity. This agrees with our observed slowdown of elevation changes.

Also for Totten Glacier in East Antarctica (Fig. 10b), we observe a clear negative SEC. This has been previously reported by several authors (e.g. Pritchard et al., 2009; Flament and Rémy, 2012; Zwally et al., 2015) but our data provide an unprecedented time span and temporal resolution, allowing to analyze the evolution of the elevation changes on a monthly scale over up to 40 years. At the very grounding line (point D), Totten Glacier thinned by 31.8±7.7 m between 1987 and 2017, which results in an average SEC rate of -1.03±0.25 m/yr. Seasat could not provide successful observations at the very grounding line but the time series for point C (around 60 km inland) with a rate of -0.38±0.10 m/yr between 1978 and 2017 and for point B (150 km) with a rate of -0.11±0.04 m/yr indicate that this thinning already preceded before the epoch of Geosat. At point A in a distance of 280 km, we find no significant elevation change (0.01±0.03 m/yr for 1978-2017). The temporal resolution of these data allows us to analyze the change over time. While we see a significant thinning at the grounding line between 1987 and 1994 of 16.6±9.8 m, the elevation stabilized between 1994 and 2004 to within ±1.5 m. After 2004, the ice at the grounding line thinned again by 15.4±5.5 m until 2017. Li et al. (2016) observe a similar variation in ice velocity measurements between 1989 and 2015. Combining their ice discharge estimates with surface mass balance, they obtain a relatively large mass imbalance for Totten Glacier in 1989, decreasing in the following years to a state close to equilibrium around 2000. After 2000, they observe an acceleration of ice flow, again consistent with our thinning rates. The authors attribute this high variability to variations in ocean temperature. In another study, Li et al. (2015) observe a grounding line retreat at Totten Glacier of 1 to 3 km between 1996 and 2013 using SAR Interferometry. They conclude that this indicates a thinning by 12 m, which is again consistent with our results over this period (12.0±8.8 m).

At Shirase Glacier in Dronning Maud Land (DML, Fig. 10c), we observe a relatively stable surface with a slightly negative change rate between 1978 and the early 2000s. The sub-intervals until 2002 in the elevation change maps of Fig. 9c-g confirm that this agrees with the large scale trend in this region. After 2002, however, the elevation change switches the sign. Our time series show an increasing surface elevation, which is most pronounced during the time of the two significant accumulation events in 2009 and 2011 in this region (Boening et al., 2012; Lenaerts et al., 2013). At point C, the elevation changed by 1.0±1.5 m between 2008 and 2012. Even at point A, more than 200 km inland and at an altitude of 2500 m, the elevation increased by 0.55±0.50 m during this time. At point D, an abrupt elevation increase is also observed in 2003, which corresponds to another SMB anomaly (cf. Fig. 2a in Lenaerts et al., 2013). The map in Fig. 9h shows that the coastal regions of Enderby Land (basin A'-B) already experienced elevation gains before 2007. In contrast to the 2009 and 2011 events, which affected a very large region (Fig. 9i), this earlier accumulation event is significantly more localized at the coast.

In contrast to the regions discussed so far, the elevation change on the plateau of East Antarctica is very small. The time series for four different points at Lake Vostok (Fig. 10d) show rates within uncertainties and very close to zero (point A: 5±9 mm/yr, B: -1±10 mm/yr, C: -3±9 mm/yr, D: -1±10 mm/yr between 1992 and 2017). The larger variations in the ERS time series are a result of the lower resolution of the waveform in the ice mode of the ERS satellites. These rates contradict the findings of Zwally et al. (2015). They report a surface elevation increase of 20 mm/yr over Lake Vostok, which would result in an

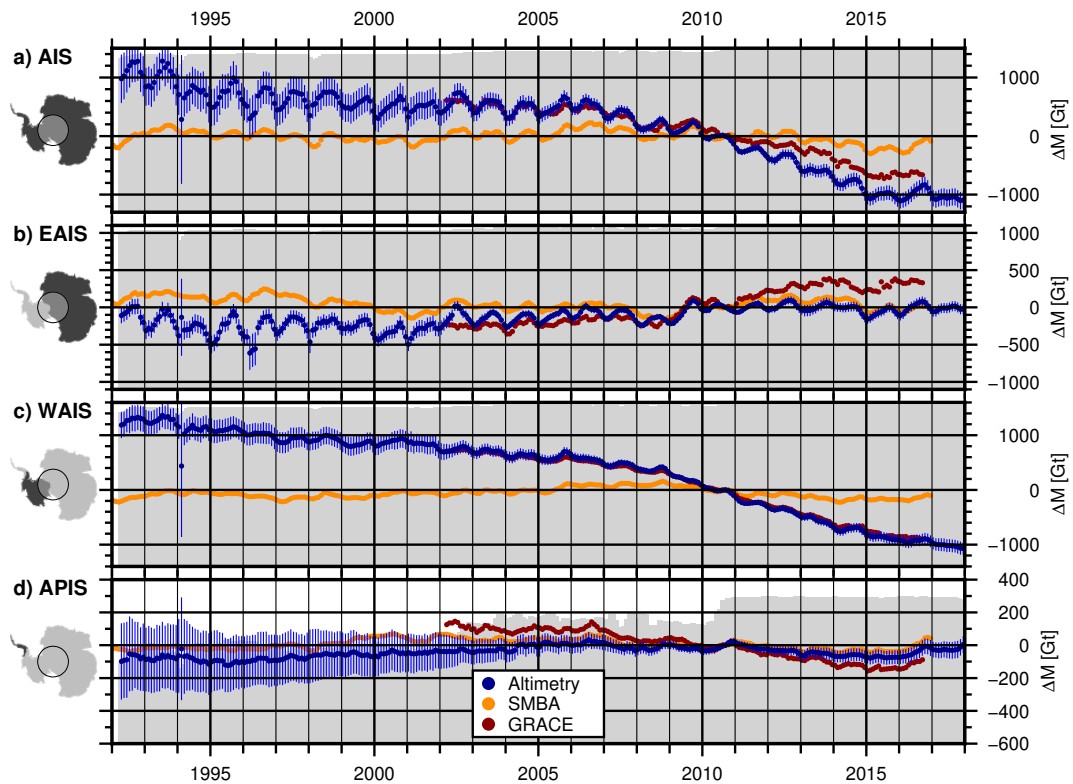

**Figure 11.** Mass change of the Antarctic Ice Sheet north of 81.5°S (**a**) and the three subregions (**b** EAIS, **c** WAIS and **d** APIS) from our combined altimetric time series (blue), GRACE (red) and SMBA (orange). The error bars show the uncertainty estimate $\sigma_\Sigma$ of the altimetry data according to Sect. F.2. The gray color in the background displays the fraction of the area covered by altimetry (up to the top means 100%).

elevation increase of 0.5 m over the period 1992-2017. Our results are confirmed by ground based static GNSS observations (Richter et al., 2008, 0.3±4.9 mm/yr), kinematic GNSS profiles measured around Vostok Station using snow mobiles (Richter et al., 2014, 1±5 mm/yr) and by GNSS profiles using traverse vehicles over the entire Lake Vostok region (Schröder et al., 2017, -1±5 mm/yr).

## 5.2 Ice sheet mass time series

The surface elevation time series are converted into ice mass changes in order to determine their effect on global sea level. In a first step, the SECs are corrected for uplift rates related to glacial isostatic adjustment (GIA) using coefficients from the IJ05_R2 model (Ivins et al., 2013). This GIA model predicts an uplift of 5 mm/yr near the Antarctic Peninsula and rates between -0.5 and +2 mm/yr in East Antarctica. Furthermore we multiplied the SEC by a scaling factor $\alpha = 1.0205$ to account for elastic solid earth rebound effects (Groh et al., 2012). The resulting ice sheet thickness changes are multiplied by each cell's area and a density according to a firn/ice mask (McMillan et al., 2014, 2016), depicted in Fig. S10, to obtain a mass

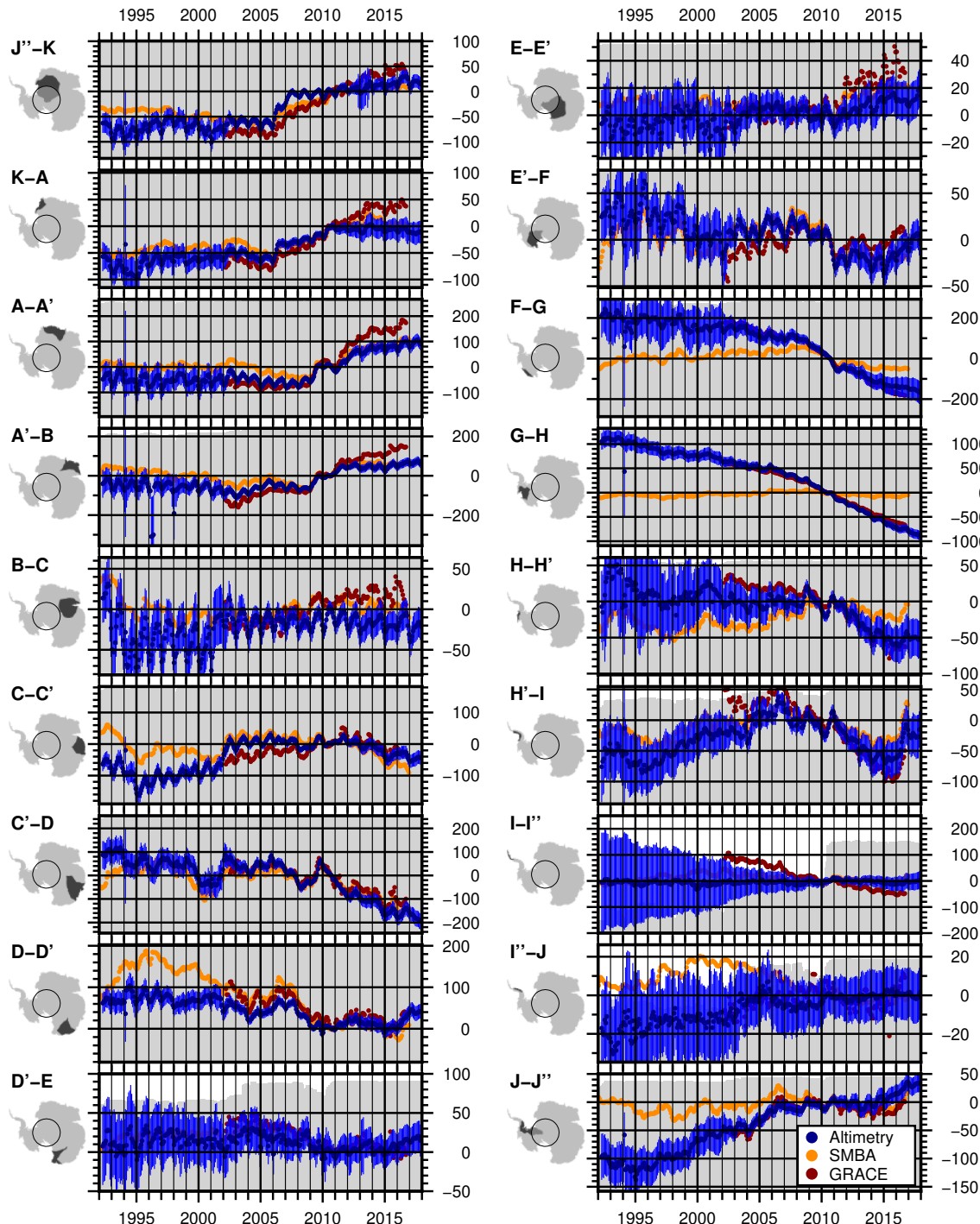

**Figure 12.** Mass change ($\Delta M\,[Gt]$) of the individual drainage basins north of 81.5°S from our combined altimetric time series (blue), GRACE (red) and SMBA (orange). The error bars show the uncertainty estimate $\sigma_\Sigma$ of the altimetry data according to Sect. F.2. The gray color in the background displays the fraction of the area covered by altimetry (up to the top means 100%).

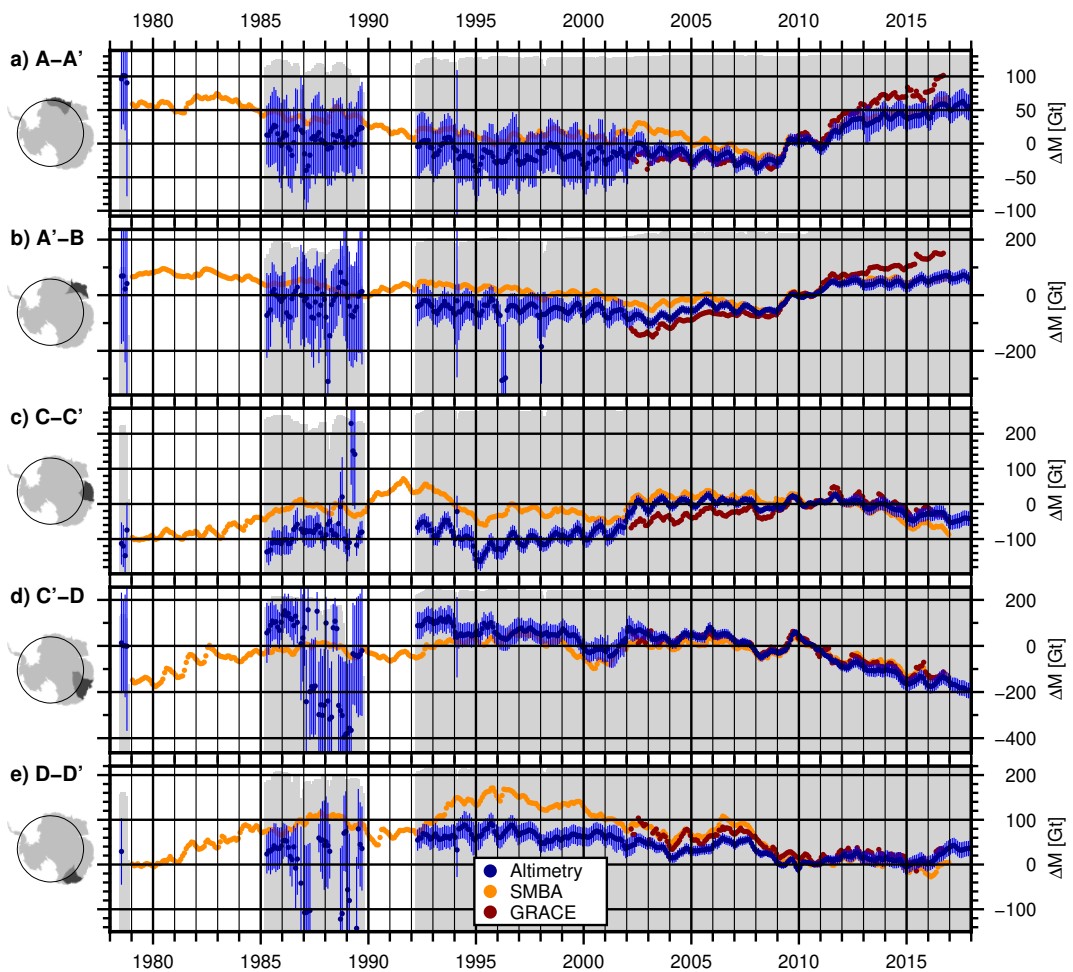

**Figure 13.** Mass change of subregions north of 72°S for several East Antarctic drainage basins from our combined altimetric time series (blue), GRACE (red) and SMBA (orange). The error bars show the uncertainty estimate $\sigma_\Sigma$ of the altimetry data according to Sect. F.2. The gray color in the background displays the fraction of the area covered by altimetry (up to the top means 100%).

change. In regions where ice dynamic processes are assumed to be dominating (e.g. in Amundsen Sea Embayment, Kamb Ice Stream or Totten Glacier), we use a density of $917\,\mathrm{kg/m^3}$. Elsewhere, we apply the density of near-surface firn as modeled by Ligtenberg et al. (2011), using annual averages of accumulation, $10\,\mathrm{m}$ wind speed and surface temperature. We have chosen this straightforward method here, instead of using the modeled impact of the temporal variations of accumulation, melting and firn compaction on the firn layer (as e.g. Zwally et al., 2015; Kallenberg et al., 2017) in the volume-to-mass conversion. This allows us to keep our altimetry time series independent from the modeled variations in SMB, which is a prerequisite for the interpretation of the comparison of both data sets.

We integrate our measurements over larger regions to calculate the cumulative mass anomalies for individual drainage basins and major Antarctic sectors (AIS, WAIS, EAIS, APIS). Our basin delineations are from Rignot et al. (2011), which have been

updated for the second Ice Sheet Mass Balance Inter-comparison Exercise (IMBIE-2, Shepherd et al., 2018). Cells that were masked out due to the predominance of rocks or that are considered unobserved after our gridding (due to the polar gap or a lack of valid observations) are not included in these sums. Uncertainty estimates are obtained by propagating the uncertainties of the SEC, the GIA and the firn density to the basin sums for each month (see Sect. F.2 for details). To account for the lack of

information due to unobserved cells, we also add a total estimate for the effect of these cells, based on trends from GRACE, to the error budget.

Figures 11a-d show time series for the entire AIS north of 81.5°S (i.e. covered by satellite altimetry since 1992), and the subregions EAIS, WAIS and the APIS. Similar time series for the single drainage basins over 1992-2017 are shown in Fig. 12. For the coastal areas of the EAIS the full time interval since 1978 is shown in Fig. 13. These four decades time series use

data north of 72°S only and, hence, provide a nearly consistent observational coverage over the whole period. To support the interpretation and evaluate the temporal evolution, we compared the respective time series to GIA-corrected cumulated mass anomalies from satellite gravimetry (GRACE, Groh and Horwath, 2016) which are products of the ESA Climate Change Initiative (CCI) Antarctic Ice Sheet project and are available for download at https://data1.geo.tu-dresden.de/ais_gmb and http://cci.esa.int/data. To reduce the effect of noise in the GRACE monthly solutions and to make the data more comparable

to our altimetry results, we applied a three-month moving average to the GRACE time series. We also compare our data to time series of cumulated surface mass balance anomaly (SMBA) from RACMO2.3p2 (van Wessem et al., 2018). To obtain these anomalies, the gridded SMB rates have been reduced by a mean rate and integrated over time. Similar to the IMAU firn model, these SMBAs contain seasonal and interannual variations due to surface processes but do not include long-term changes over the full modeled period (1979-2016). The different time series show the good agreement of the techniques in resolving

interannual variations. For example for the basin of Totten Glacier (C'-D in Fig. 12), all techniques observe a negative mass anomaly in early 2008, followed by a significant mass gain in 2009 as previously reported by Velicogna et al. (2014) and Li et al. (2016). Between 03/2008 and 10/2009, we obtain a mass difference of 116.6±27.0 Gt from altimetry, 109.4 Gt from SMBA and 113.4 Gt from GRACE. The high agreement with SMBA indicates that this mass gain at Totten is caused by snow accumulation. In most of the basins, we observe a similar high agreement in the short-term variations. A good example for a

total mass change signal which is constituted from components of SMBA and ice dynamics is the Getz and Abbot region (F-G) in West Antarctica. While all techniques observe a significant mass loss between 2009 and 2011, the SMBA does not contain the decadal trend, as observed by altimetry and GRACE. In some regions, there are also significant discrepancies between the data sets of satellite atlimetry and GRACE. Inadequate sampling by radar altimetry (such as in the northern tip of the Antarctic Peninsula (I-I") where steep regional topography and small outlet glacier size limits the recovery), leakage in the GRACE

estimate between different sectors and uncertainties in the individual measurements and in the geophysical corrections might cause these differences. In George V Land (D-D'), the agreement during the GRACE period is reasonable, while the mass gain, indicated by SMBA in the early 1990s is not revealed by the altimetry time series.

Over the last 25 years our data indicate a clearly negative mass balance of -2068±377 Gt for the AIS (Fig. 11a), which corresponds to an increase in mean sea level of 5.7±1.0 mm. This change is mainly a result of the mass loss in the WAIS over

the last decade. In contrast, the EAIS has been very stable over our observational record (120±121 Gt between 1992 and 2017).

**Table 2.** Mass change rates for different regions of the Antarctic Ice Sheet and different time intervals. The sizes of the total and observed area refer to all cells classified as ice sheet in the respective region (and, if stated, limited by the given latitude).

| region | area [$10^3$km$^2$] | | dM/dt [Gt/yr] | | | | |
|---|---|---|---|---|---|---|---|
| | total | observed | 1978-2017 | 1992-2017 | 1978-1992 | 1992-2010 | 2010-2017 |
| AIS | 11892 | 11630 | - | - | - | - | -117.5±25.5 |
| EAIS | 9620 | 9413 | - | - | - | - | 1.6±13.1 |
| WAIS | 2038 | 2008 | - | - | - | - | -114.5±19.9 |
| APIS | 232 | 208 | - | - | - | - | -4.5±8.7 |
| AIS (<81.5°S) | 9391 | 9053 | - | -84.7±15.5 | - | -58.6±20.3 | -137.0±24.9 |
| EAIS (<81.5°S) | 7764 | 7555 | - | 4.9±5.0 | - | 8.0±6.2 | 2.4±12.4 |
| WAIS (<81.5°S) | 1394 | 1358 | - | -91.7±10.3 | - | -69.4±13.1 | -134.9±19.6 |
| APIS (<81.5°S) | 232 | 142 | - | 2.1±8.9 | - | 2.8±12.3 | -4.5±8.7 |
| EAIS (<72°S) | 2779 | 2274 | 1.5±5.8 | -3.4±4.0 | 12.1±17.4 | 0.0±4.9 | -8.4±10.1 |

For the APIS (<72°S), the very sparse observations of Seasat and Geosat did not allow calculate a reliable trend.

The time series of the APIS contains large uncertainties due to many unobserved cells. Mass change rates for selected regions, obtained from the differences over a specific time interval, and their uncertainties are given in Tab. 2. We calculated separate trends for the area north of 72°S, which is covered by all satellites, the area north of 81.5°S, which is covered since ERS-1 and for the total area, which is covered since CryoSat-2, except for its 500 km polar gap. 96.4% of the cells classified as ice sheet north of 81.5°S are successfully covered by observations of ERS-1. Cells without successful observation occur mostly at the APIS, where only 61% is covered with data.

From the overall mass loss of -2068±377 Gt for the AIS (<81.5°S over 1992-2017) we obtain an average long-term rate of -84.7±15.5 Gt/yr (or a corresponding mean sea level change rate of 0.24±0.04 mm/yr). After 2010, this rate accelerated to -137±25 Gt/yr or 0.38±0.07 mm/yr of mean sea level.

# 6  Discussion

## 6.1  Surface elevation changes

Combining all the single missions consistently, our SEC time series allow to analyze the long-term changes over the full time period of satellite altimetry observations. For 79% of the area of the AIS, this means a time span of 25 years. Over 25% of the ice sheet, largely in the coastal regions of East Antarctica, the time series can be extended back 40 years. Such long-term trends are significantly less affected by short-term variations in snowfall than a trend from a single mission. Furthermore, the period of observation of a single mission is short compared to climatic oscillations as reported e.g. by Mémin et al. (2015). Our extended time series helps to separate elevation change due to climate variations from potentially accelerating volume losses.

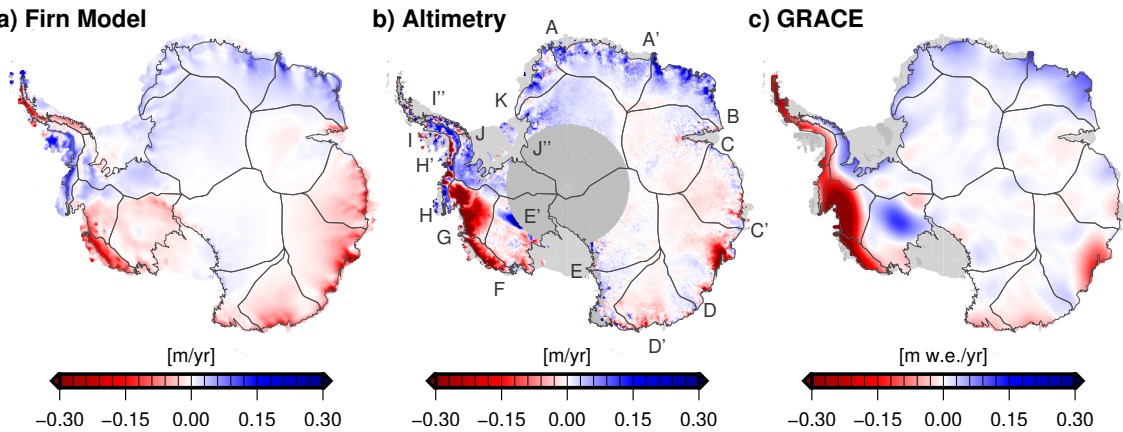

**Figure 14.** Mean rates for the time interval 2002-2016 of elevation changes from IMAU-FDM **(a)**, from the multi-mission SEC grids **(b)** and of the mass changes from GRACE **(c)**.

Also Seasat and Geosat provide important information here, despite their larger uncertainties. Due to the stability criteria in the calibration, we do not expect significant new insights on the East Antarctic plateau (even as regional variation still may be discernible as we used an ice-sheet-wide average in calibration). However, in the coastal regions of East Antarctica, with SECs of up to several meters w.r.t 2010 (see Fig. 6), also the older data can contribute significant information to study elevation
changes in a long-term context of 40 years (cf. the rates in Fig. 9 and their uncertainties in Fig. S9). Unfortunately, in coastal DML west of the ice divide A', the data of Seasat and Geosat are very noisy due to the mountain ranges just north of 72°S in these regions. They lead to many signal losses along the way across this part of the ice sheet. The same applies to the measurements at the APIS.

    The benefits of a seamless combination of the time series are demonstrated in Fig. 9. The time intervals for the elevation
changes are independent of the observational period of a single mission. This is necessary to analyze processes which occurred close to the transition between different missions. A good example of the advantage of such long time series are the elevation changes caused by the accumulation events in DML. Figure 10c clearly shows the changes in elevation, caused by the strong snowfall events in 2009 and 2011. The mission lifetime of ICESat ended in 10/2009, CryoSat-2 provided the first measurements in 07/2010. Only Envisat covered both events but here, the orbit was shifted in 10/2010, resulting in different repeat track cells
covered before and after the orbit shift. We merged all these missions as described in Sect. 3.3, which allows us to analyze the full time series. Comparing the elevation changes from altimetry with those in the FDM serves as a cross-validation of both data sets. For example at point A in Fig.10c our SEC time series observes a change of 0.55±0.50 m between 2008 and 2012 while the FDM models a very similar elevation gain of 0.48 m for this period. Figure 8 shows the degree of agreement over the entire AIS.

As these elevation change rates alone do not contain any information on their origin, additional data are needed for improved process understanding. Figure 14 shows SEC rates for the interval 2002-2016 (March-September respectively) from altimetry

and the IMAU-FDM and corresponding rates of ice mass changes from GRACE. These maps show that the elevation gains in DML and Enderby Land agree very well with the firn model, which implies that increased snow accumulation during this period is responsible for the thickening. For Princess Elizabeth Land (C-C'), the negative rates agree as well, implying that the thinning here can be related to lower than normal snow accumulation. In contrast, the strong thinning along the Amundsen Sea Embayment (G-H) or the thickening of Kamb Ice Stream (E'-F) is not present in the FDM results but does show up in the GRACE data. Due to the higher densities of the involved material, ice dynamic processes are even more pronounced in the map of mass changes, compared to the maps of elevation changes.

The inland propagation of dynamic thinning of the glaciers of the Amundsen Sea Embayment over the last decades has been described by Konrad et al. (2016). A recent onset of significant mass losses has also been reported for the adjacent glaciers along the Bellingshausen Sea (H-I, Wouters et al., 2015) and in the Getz and Abbot region (F-G, Chuter et al., 2017). Fig. 9i reveals that the largest losses along the coast of the WAIS occurred between 2007 and 2012. The period 2012-2017 (Fig. 9i) shows that only a part of these large rates is persistent. While the ice discharge of the Getz and Abbot region even increased by 6% between 2008 and 2015 (Gardner et al., 2018), the deceleration of the elevation change after 2012 indicates that also interannual variations in SMB have to be considered here (see also Chuter et al., 2017). The FDM-derived rate in Fig. 14a confirms the role of the surface mass balance in this region.

## 6.2 Ice sheet mass time series

The integrated time series of basin-wide mass changes allow to analyze the temporal evolution at a monthly resolution (Fig. 12). As described in the previous section, for most of the basins of the WAIS, they show an increase of mass loss after the mid-2000s. The acceleration of thinning at the Getz and Abbot region (F-G) started already in 2004, but experienced a further significant acceleration after 2007. In the Amundsen Sea Embayment, a small positive mass anomaly in late 2005 relates to a similar event in the SMBA time series, but after that, also here the overall mass losses accelerated. The Bellingshausen Sea basin (H-H') was relatively stable until 2009, but started to lose significant amounts of mass after that time, as reported by Wouters et al. (2015). Since 2016, however, we observe that the basins at the Bellingshausen Sea and the western part of the Peninsula regained mass. The comparison with SMBA reveals that this can be explained by a positive snowfall anomaly in this area in 2016. The shape and orientation of the Peninsula makes GRACE observations challenging with respect to leakage and GRACE error effects (Horwath and Dietrich, 2009). Nevertheless, the results of the satellite gravity mission confirm this mass anomaly.

A similar comparison of the ice sheet wide mass time series between altimetry and GRACE in Fig. 11 reveals that for the entire WAIS, both data sets agree very well, while for the APIS and the EAIS, we observe significant differences on a decadal scale of the trends. The percentage of observed area of the APIS (gray area in the background of Fig. 11d) indicates that before 2010, a significant part of the area remained unobserved. Here, conventional RA measurements very often failed due to the rugged terrain. Even for ICESat, the large across track distances and the dependence on cloud-free conditions make measurements very sparse at the Peninsula. With the weather independent, dense and small footprint measurements of CryoSat-2 in SARIn mode, up to 80% of the area are covered by observations. Compared to GRACE, however, we observe a

significantly weaker mass loss signal. Thomas et al. (2008) pointed out that RA fails to sample especially the large elevation changes in narrow valleys of outlet glaciers. This leads to an overall underestimation of the signal by altimetric observations. Furthermore, the complex terrain, especially in the APIS, also causes problems in the parameter fit. Even if enough valid measurements are available (as e.g. from ICESat or CryoSat-2), the fit of a planar surface over a diameter of $2\,\mathrm{km}$ in our repeat

altimetry processing can hardly adequately represent the real topography here. Our approach is designed to provide valid results over the majority of the AIS. Under the challenging conditions of the APIS, modifications such as a smaller diameter or more complex parametrization of the surface would surely help to improve the results. Furthermore, we did not calculate a SEC for cells that are further away than a beam-limited radar footprint from valid measurements. In order to interpolate or even extrapolate the results to unobserved cells, advanced gridding methods such as kriging, especially with the help of additional

data sets (Hurkmans et al., 2012), would be advisable.

This effect may also explain the differences of our results, compared to the results of the combination of different techniques by Shepherd et al. (2018). Their 1992-2017 rate of -109±56 $\mathrm{Gt/yr}$ agrees within error bars with our results but our rate of -84.7±15.5 $\mathrm{Gt/yr}$ is considerably smaller. Part of this disagreement might be attributed to differences in the estimates for the Antarctic Peninsula where retrieving reliable radar altimetry estimates is non-trivial. However, the extended material in

Shepherd et al. (2018) shows that there are still some discrepancies between the different techniques to determine the AIS mass balance. For the time interval 2003-2010 (Extended Data Table 4 in Shepherd et al., 2018) the Input-Output method obtains a rate of -201±82 $\mathrm{Gt/yr}$ for the AIS, while the mass balance rates from satellite gravimetry (-76±20 $\mathrm{Gt/yr}$) and from altimetry (-43±21 $\mathrm{Gt/yr}$) agree much better with our result for the AIS (<81.5°S) between 2003 and 2010 of -65±25 $\mathrm{Gt/yr}$.

Besides the Peninsula, our comparison of mass changes from altimetry and from GRACE at the EAIS (Fig. 11b) reveals

some significant differences between the time series. For the time interval 2002 to 2016 (see Sect. F.3), the mean rate at the EAIS from altimetry (9.6±6.9 $\mathrm{Gt/yr}$) is mainly dominated by the accumulation events in 2009 and 2011. In contrast, the GRACE data imply an average mass gain of 42.1 $\mathrm{Gt/yr}$ over this time interval. Especially after 2011, the differences become very prominent in the time series. The mass changes for the individual basins (Fig. 12) reveal that this difference in the signals can be attributed to DML and Enderby Land. This might be a sign for dynamic thickening. Here, all elevation changes have

been converted to mass using the density of surface firn. If a part of the positive elevation changes in this region indeed would be caused by ice dynamics, this would lead to an underestimation of mass gains from altimetry. The results of the Bayesian combined approach of Martín-Español et al. (2017) also suggest a small dynamic thickening in this region. Rignot et al. (2008) observed no significant mass changes in this region between 1992 and 2006 using the input-output-method. Gardner et al. (2018) compared present day ice flow velocities to measurements from 2008. They obtain a slightly reduced ice discharge

in DML (which would support the hypothesis of a dynamic thickening), while they observe a small increase in discharge for Enderby Land. Part of the discrepancy with the GRACE results could be also due to uncertainties in the geophysical corrections applied to the GRACE data, such as the effects of glacial isostatic adjustment. More work, similar to the Ice Sheet Mass Balance Inter-comparison Exercises (Shepherd et al., 2018) or the combination of different types of observations as in Martín-Español et al. (2016), could help identify the reasons leading to the disagreement.

# 7 Conclusions

In this paper we presented an approach to combine different satellite altimetry missions, observation modes and techniques. The reprocessing of the conventional pulse limited radar altimetry ensures that two fundamental steps in processing of radar ice altimetry, the waveform retracking and the slope correction, are performed consistently. Furthermore, we showed that the methods used here improved the overall precision by 50% over the standard data sets available from ESA and NASA. The validation with in situ and airborne measurements and the comparison with the IMAU-FDM shows that inter-mission offsets have been successfully corrected and that the uncertainty estimates for our resulting monthly multi-mission SEC grids are realistic.

We analyzed the resulting time series and found that they provide detailed insight in the evolution of the surface elevation of the Antarctic Ice Sheet. From the combined SEC time series we calculated the long-term surface elevation change over the last 25 years. Observations from the Seasat and Geosat missions extend the time series in the coastal regions of East Antarctica back to 1978. The unique data show that large parts of the East Antarctic plateau are very close to equilibrium, while changes over shorter time intervals identify interannual variations, which cannot be identified in long-term trends and are mostly associated with snowfall anomalies.

The monthly mass time series show that the AIS (excluding the polar gap within 81.5°S) lost an average amount of mass of -84.7±15.5 Gt/yr between 1992 and 2017 (equivalent to 0.24±0.04 mm/yr of mean sea level change). These losses accelerated in several regions and, hence, for 2010-2017 we obtain -137.0±24.9 Gt/yr (or 0.38±0.07 mm/yr) for the same area. The comparison of the altimetry-derived mass changes, integrated over different basins and regions of the ice sheet, with SMBA and GRACE shows high consistency of the different techniques. A correlation coefficient between the mass anomalies from altimetry and from GRACE of 0.96 (for the time interval 2002-2016, see Tab. S4) indicates the excellent agreement of the observed interannual variations. The correlation with the SMBA (0.60 for 1992-2016) is comparatively lower but still indicates a high agreement. In the APIS, differences between the mass time series of the different techniques arise mainly due to the poor spatial sampling of the altimetry data, while for the EAIS, the remaining discrepancies to mass time series from GRACE might be explained by the density mask used or uncertainties in the GRACE processing. These remaining issues and open questions should be addressed in future work in order to further reduce the uncertainty of the estimates of the mass balance of the AIS. The recently launched laser altimeter ICESat-2 promises a new milestone in ice sheet altimetry. We believe that our multi-mission combination approach can provide an important tool for including the extremely high resolution of this mission into the long-time observations of satellite altimetry spanning the past few decades.

*Data availability.* Our resulting monthly 10x10 km grids of SEC w.r.t 09/2010, accompanied by corresponding uncertainty estimates, is available for download at https://doi.pangaea.de/10.1594/PANGAEA.897390.

*Author contributions.* L. Schröder designed the study and developed the PLRA reprocessing, the repeat altimetry processing and the time series generation. V. Helm supplied the reprocessed CryoSat-2 SARIn data. Stefan Ligtenberg and Michiel van den Broeke provided the RACMO and IMAU-FDM models. All authors discussed the results and contributed to the writing and editing of the manuscript.

*Acknowledgements.* This work is supported by the Deutsche Bundesstiftung Umwelt (DBU, German Federal Environmental Foundation).
5 We thank the European Space Agency, the National Snow and Ice Data Center and the NASA Goddard Space Flight Center for providing the altimetry data products. Especially we would like to thank Jairo Santana for his support to access the GSFC data. We are very grateful for the comments from the anonymous referees, A.Shepherd and the editor E.Berthier, which significantly helped to improve and clarify the manuscript. We acknowledge support by the Open Access Publication Funds of the SLUB/TU Dresden.

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
