# Peer review of "Four decades of Antarctic surface elevation changes from multi-mission satellite altimetry"

_The Cryosphere, 2018_

## Referee Comment (RC1) · Anonymous Referee #1 · 1 Jun 2018

Summary The topic of this paper, surface elevation change in Antarctica, is extremely interesting given that altimetry is the only dataset available that can provide a continuous, continent wide record of long term ice sheet change. While elevation change measurements from individual satellites such as CryoSat-2, ICESat and Envisat already exist in the published literature, the authors present new results from previously unexplored data acquired by Seasat and Geosat in the 70's and 80's which should provide an exiting new insight. This paper fails to deliver on all counts. The conclusions state that this paper presents a method for combining multiple satellite datasets. However, the method employed is not well described either in terms of the retracking, the elevation change processing chain or the post processing data filtering, extrapolation

and smoothing. On top of this, the authors haven't taken the time to calculate an error estimate for the altimetry data, and consequently all plots and maps are presented without this essential component. The elevation change dataset is presented without any serious attempt to validate any of the results. While I acknowledge that it will be difficult to find auxiliary data to validate the earliest two satellites, there is a wealth of airborne data acquired in Antarctica over recent decades that could, but hasn't been used. Instead the authors have chosen to validate a flat, unchanging region of east Antarctica with little more than one track of data. This region is not representative of the regions experiencing the most rapid change in Antarctica, therefore don't provide any confidence in the results presented in arguably the most important regions. I have concerns about the weak attribution of the cause of change throughout the paper, and the overly simplistic nature of the analysis and discussion on this topic. The paper presents only elevation and volume change, but these signals are then attributed to ice dynamics and precipitation anomalies with no supporting evidence for the former, and largely no quantitative evidence about the relationship for the later. No previously unknown dynamic or precipitation induced elevation change signal is reported on, so we do not learn anything new from this paper about why changes in ice thickness have occurred in Antarctica. On top of these technical issues, the manuscript text is lacking numbers to back up most of the statements made. For example, the abstract and conclusions contain no key numbers on elevation change in Antarctica, despite this being the title of the paper. In addition to this quite a lot of the text is poorly written, so the text will be significantly improved if thoroughly re-edited. My criticisms of the methods and results presented in the paper are major, and will be time consuming to properly address. As it stands, ambiguity about the methods, absence of an error estimate, and lack of proper validation of the result make it entirely possible that the magnitude of the elevation change signal may not be correctly reported in this paper. As the final closing statement of the abstract and the conclusions state, the single new achievement of this paper is that it has demonstrated that Seasat and Geosat 'can provide reliable information'. This is not a scientific conclusion on how elevation change in Antarctica has

changed since the 70's as the papers title implies. If the authors can rewrite the paper to provide geophysical results and quantitative conclusions on this topic, then this forms the basis of an interesting paper, but as it stands this manuscript feels like an unfinished piece of work. The following specific edits should be addressed by the author if the paper is to be published. Specific Edits Abstract – The abstract is quite long, but not well written. About half of the text is spent discussing the percent coverage for the 25 year and 40 year epochs, however these numbers aren't key scientific results so would be better placed in the data or methods section. Additionally, the coverage stats are poorly defined here, for example is this the percent coverage that of the raw data, plane fit output at whatever grid resolution used, or the extrapolated interpolated result. Without being specific the coverage stats are open to misinterpretation. Abstract - The title of the paper states that this paper is about surface elevation change, but there are no key surface elevation change numbers stated in the abstract. Why not, if this is the main purpose of the paper? Abstract – The second paragraph of the abstract lacks any quantitative facts. For example, Pg1 L14 states that surface elevation change shows 'high coincidence' with precipitation anomalies and gravimetry; Pg1 L16 states that there is a 'high level' of agreement; and Pg1 L18 states that 'Geosat coincides very well with...'. The authors should replace these generic adjectives with quantitative statistics to back up their statements. Abstract – Pg1 L15 – 'Satellite gravimetry' is the technique, but the authors have presumably compared their elevation results against a derived product, such as mass loss. Edit wording to be precise. Abstract – Pg1 L18 – The Seasat and Geosat altimeters operated at radar frequencies and will consequently penetrate some depth into the snowpack, its therefore not a given that the elevation trend from these satellites should correlate with precipitation anomalies in snowfall. The penetration depth is spatially and temporally variable, influenced by snow density and moisture content. As the radar return originates from a scattering horizon within the snowpack, correlation between precipitation anomalies and rates of elevation change can't prove that the elevation change trends are 'reliable', so the satellite based elevation trends must be verified with a comparable elevation change

dataset. P1 L21 – Sentence wording not correct English, edit required. The wording throughout this paragraph is poor. P1 L22 – Does the author mean sequentially rather than concurrently? P2 L6 – Edit wording to say precisely what is meant. A long time series would not have prevented Wingham observing negative elevation rates in Dronning Maud Land compared with Flaments positive result for the same area, because as stated the observational time period is different. 'Help reduce the influence of such events' is factually incorrect. P2 L10 – Edit wording. Mission calibration doesn't become 'more important', it is maybe more challenging though. P2, Fig1 – Add separate colorbar for map of spatial coverage as currently hard to interpret. Just the circle outline that corresponds to bar color, but really hard to see in pole hole. P3 L1 – Edit wording to be more precise. As the raw data from both satellites was acquired in the same pulse limited imaging mode, the use of the word 'mode' to describe a processing choice could be misinterpreted. Additionally I think the two modes authors are talking about are ocean and ice retrackers, however there are actually 3 retrackers available for these missions (ice-1 and ice-2 are separate). P3 L3 – Paolo et al presents results over flat ice shelves with zero slope, whereas this paper presents results over an ice sheet, where the most rapidly changing regions are found in the most steeply sloping terrain. The logic that Paolo used to justify including data from different imaging modes therefore may not apply to this paper. The authors should use data from a single retracker which has been shown to be more reliable, or quantitatively justify why including less reliable data improves the quality of the end elevation change result. E.g. via coverage, temporal extent, or reduced error maybe. It follows that a separate error estimate should be provided for the elevation change result derived from different quality input datasets. P3 L4 – Specify the size of the bias between both modes, for both satellites. P3 L16 – The Helm et al DEM is the ice surface during the first 3 years of CryoSat, however this surface has changed significantly in many regions throughout the 40-year study period. The ice surface of the DEM should evolve temporally to reflect the known elevation change, otherwise the slope correction will not be correct. This effect will be significant in regions such as WAIS which have shown to thin at a

maximum rates of up to 9 m/yr. Have the authors done this, or if not what is the error on the slope correction that will result from not temporally evolving the ice surface for 40 years? P3 L29 – Can the authors quantitatively state how much less sensitive to noise the OCOG retracker is compared to other options, and does this affect all satellites the same way given that the spatial resolution and imaging modes are different. Some retrackers will perform better over different terrain types (sloping or flat), therefore it would be helpful for more details to be provided about the region, time period, and satellites for which this analysis was performed. P4 L3 – If the CryoSat retracking doesn't exactly replicate the methods in Helm et al, the full details should be detailed in this paper. P4 L13 – It is well known (as the authors later state) that anisotropy exists between ascending and descending tracks of radar altimetry data, and indeed many elevation change papers include a term for this in the plane fit solution to exclude any bias from this. When calculating data precision from ascending and descending tracks have the authors performed such a correction, and if so could further detail be provided on its size per mission. P5 L5 – Edit text to state quantitative statistics about the absolute or percent improvement following their slope correction. Frustrating that 'superior performance' and 'similar improvements' used when a number would be more persuasive. P5 L17 – Figure S1 does show a reduction in the anisotropy effect, but the signal is clearly still present in the data. Additionally, smoothing a result with an artefact in doesn't remove the affected data, so this error will clearly have an effect on the end result. If as the authors state recent previously published studies have designed and successfully implemented an anisotropy correction, this paper should add this step to avoid unnecessary error. If the authors chose not to do this, I would ask that the quantify what the affect of not applying the correction is to prove its not discernable. P6 L25 – If there are differences in the processing methods used for different missions as stated, this should be fully specified in the supplementary material. Use of full parameter names as found in the mission meta data will ensure that the methods and results presented are repeatable. P8 L1 – Spelling P8 L1 – Specify the thresholds and variables against which data is filtered out during the elevation change iterative

processing. Are the same values used for all missions? P8 L7 – Provide some detail on how the backscatter penetration correction is calculated and applied. E.g. over what epoch? P8 L18 – State threshold used to determine outliers. P9 Fig4 – Red colors in this plot do not print well so can't easily differentiate missions. Change color scale used. P9 L3 – The authors are simultaneously arguing that for all of the more recent missions a spatially variable offset correction must be applied as the offset is spatially variable, while stating that for Seaseat and Geosat the offset correction must be a constant because some of the difference could may be real elevation change. If the later is true, why does this not also hold for the more recent missions? The fact that a spatially variable correction can't be or hasn't been calculated doesn't remove the justification for why its needed. P10 L15 – The extrapolation and interpolation steps are not sufficient. It doesn't account for the spatially variable pattern of thinning, which increases towards the coast and is larger on fast flowing ice streams, and the method is not fully described. What is the maximum distance over which gaps are filled? In areas such as the ice sheet edge or on the Antarctic peninsula, which receive exceptionally poor coverage in earlier missions, how are these larger gaps filled? Equally, in order to state EAIS, WAIS, and continent wide thinning rates, the pole hole needs filling. P10 L31 – The signal in the 1978 to 1992 map is extremely noisy with lots of variation over short spatial scales. Although the authors assert that 'coherent signal' can be obtained from these missions, to me it looks like the differences are as great if not greater than the similarities between the later data. Generate a difference map, or statistics, to quantitatively demonstrate that the results form the early missions are 'coherent', or similar to those from later missions. P10 L32 – The authors attribute elevation change across the ice sheet to ice dynamics without providing evidence in support of this. Elevation change can be caused by dynamic ice thinning, a snowfall anomaly, change in the scattering horizon, or measurement error, so all of these factors will influence the result not just dynamic thinning alone. Which specific regions are attributed to dynamic change? If based on previous publications, please provide relevant citations. The authors should quantify how much of the elevation change is

dynamic, vs all of these other factors, in order to demonstrate that it's the largest contributor. P10 L33 – State what's classed as a short time scale, annual/ sub-decadal/ other? P11 L1 – Again maps of elevation change are not evidence of change in dynamic thinning without additional supporting data. The authors need to quantify and rule out the influence of snowfall variations, change in scattering horizon, and error, and a corresponding change in ice speed should also be observed. Without this elevation change due to long term, decadal fluctuations in snowfall, may be mischaracterized as dynamic change. The authors should also state which regions they are referring to. P11 Fig5a – Add distance markers to the flow line on one of the maps, hard to tie 5b to locations along it. For example, what distance is the limit of Seasat and Geoset data? P11 Fig5b – Add error bars to this plot. P12 Fig6 – Provide a spatially variable error map for each of these epochs. So far the results have been presented without any error method described, or measurements included in the plots. P12 L4 – Edit text. The discussions aren't controversial, there is just are just different approaches each of which have advantages and limitations. P13 Fig7 – Add error bars to all lines on this plot. P13 Fig7 – Clarify how the % coverage has been calculated, and add coverage labeling to an axis. For Antarctica and the LPZ, I don't see how it can be 100% during the 1990's when the pole hole is not observed. Additionally, in the methods its stated that the raw data was originally gridded at 1km resolution, which will result in data gaps of several kilometers between tracks before the CryoSat's precessing orbit comes online. So again I struggle to see how such complete observational coverage is achieved, unless extrapolated and interpolated data is classed as an observation, which of course it isn't. Could the authors clarify? P14 Fig8 - Add error bars to all lines on this plot. P14 Fig8 - Add % coverage axis label to plot. Same comment applies about how the coverage calculation is done. P14 Fig8 – Add a table in the SOM with the areas of the drainage basin sub-regions used to generate this plot. P15 Fig9 – I don't understand the figure caption, please rewrite more clearly. P16 L1 to 14 – These results sections are very poorly written as no actual results are described! The authors just state what some of the figures show, and leave the reader to do all the hard work

of reading off numbers and key statistics, comparing this with numbers they have read in previous studies. Re write the results section to present some actual results. I don't think a single elevation change number has been presented in the text yet, despite that being the title of the paper! P16 L9 to 13 – The authors have described what this plot is, but haven't explained why it matters or what the key scientific result is. Either remove figure 9 or explain why its an important addition. P16 F10 – Label y axis of b, presumably count. Edit figure caption to state time period data validated over. P16 L15 – The validation performed in this paper is completely insufficient, and I would argue it leaves the result presented essentially unvalidated. Use of only 19 GNSS profiles, in a region of no known change, over a limited time period and spatial extent, and on unchallenging flat terrain, does not inform the reader about the validity of these results. At a minimum the authors must use a more comprehensive independent dataset, e.g. ice bridge. P17 L2 – This 'validation' cannot be interpreted as an error. A formal error budget based on the altimetry data itself must be documented and added to the plots in this paper. Validation and error estimation are separate things. P17 L14 – The authors don't need to limit their validation data to in situ measurements, much more spatially and temporally extensive airborne data is available and this should be used. P17 L21 – State the number, don't leave the reader to guess how much elevation change you have measured! Presumably it is different for the peninsula and east Antarctica, so again please present your result. P17 L22 – Add figure number. P17 L23 – Add figure number, or label somewhere. Location of ice streams mentioned hasn't been identified on any plots in this paper. State quantitatively how your numbers compare with this thinning rates presented by Rignot (2006), the time period is different so there should be something new to say. P18 L2 to 6 – Use statistics to show the agreement, or disagreement between the elevation change and precipitation anomaly. State with numbers what 'significant difference' is that allows ice dynamics to be determined. State how far inland the thinning was in earlier decades vs how far inland it reaches now. P18 L14 – State the threshold used to determine a strong snowfall anomaly. It looks like it varies just as much at different times around other regions in

Antarctica. P18 L15 – what distance away from the grounding line were Seasat and Geosat typically able to observe. P18 L20 – The 12 m/yr thinning suggested by Li et al was due to grounding line retreat between '96 and 2013, however the 12 m thinning present in this paper is for 1985 – 2010. Given that the measurements presented in this paper start approximately a decade earlier, if as this paper sats the glacier was already thinning in the 80's then the magnitude and rate are not in agreement with Li et al. Please clarify. P18 L 23 – The authors need to take more care before attributing elevation change to dynamic ice mass loss. There are many signals present in their continent wide maps that may well not be attributed to dynamic ice loss. There is not consensus in the published literature that all Antarctic peninsula elevation change is dominated by ice dynamics, and the authors themselves later attribute a different elevation change signal on the peninsula to precipitation anomalies without providing any more or less evidence that a different process could be responsible (P18 L30). The authors must present quantitative evidence to support their claims either way. P18 L30 – GRACE data cannot disentangle whether elevation change is caused by snowfall anomaly or ice dynamics, as ice mass is lost in both instances. Only velocity data can demonstrate whether ice was exported from the catchment at an increased rate, proving ice dynamics. Both dynamic thinning and snowfall anomalies result in mass loss, but gravimetry mass loss measurements don't show which of these two different processes might be the cause. P18 L35 – Quantify very well. P19 L10 – In Fig S7c I can't see any 2002 step in the precipitation time series so its not clear to me that there is good agreement. Again please provide quantitative stats to back this up, rather than just making unsupported qualitative statements. Cite Lenaerts et al 2013 with respect to the 2009 and 2011 precipitation anomaly results as not a new result from this paper. P19 L16 – While this appears to be true for 2008/10, there is a more significant accumulation gain in the 1990's that is not visible in the elevation change result at all. This is in part because the authors are comparing different things, snow mass anomaly, vs elevation change. Direct comparison not possible unless elevation change converted to mass change. P20 Table2 – There is negligible data coverage outside of East Antarctica prior to 1992, so not valid to include an Antarctic wide number for the '78 to 2017 period in row 3. Remove this number as misleading. P21 Conclusions – There are no key results from this paper presented in the conclusions. Add a few key numbers. SOM A.1 – State threshold used to determine if noise is too high. SOM A.1 – State the start and end date for each satellite dataset used. SOM A.2 – State which retracker the elevation measurements were derived from. SOM A.2 – State the specific name of the metadata flag used to filter out data, and if a threshold was used, state the number that this was set at. SOM A.2 – Adjust Figure 1 to reflect the actual time period of ERS-1 data used. (same applies for all missions) SOM A.3 – State which retracker the elevation measurements were derived from. SOM A.3 – State specifically which measurement confidence flags were used, and again if a threshold was used, state the number that this was set at. SOM A.4 – State specifically which measurement confidence flags were used, and again if a threshold was used, state the number that this was set at. SOM A.5 – State which LRM retracker the elevation measurements were derived from. SOM A.5 – State specifically which measurement confidence flags were used to filter data, and again if a threshold was used, state the number that this was set at. SOM B – Edit title and section text to be more specific as its unclear specifically what the authors have reprocessed? Is it that the elevation measurements have been retracked? Read as a stand alone section I don't know what SOM E S6 and S7 - Add error bars to all lines on this plot. Add % coverage axis label to plot. Same comment applies about how the coverage calculation is done.

---

## Short Comment (SC1) · 5 Jun 2018

I am a fan of all things altimetry, but this paper falls a long way short of the standard that publications on this topic need to meet today.

1. Title. The title is misleading; a minority (25%) of the data set spans 4 decades. It should be modified to explain this or address the majority data set

2. Error budget. The authors use the variance of single cycle crossover differences as a measure of error, and conclude that the reduced variance offered by their preferred retracker indicates a de-facto improvement in error. This is misleading, as their

conclusion is entirely related to their choice of error metric and is therefore subjective. To conclude an improvement the authors should evaluate each retracker against independent observations of greater and known precision.

3. Methods. The authors discuss that a variety of approaches have been used to derive continental scale elevation change measurements, leading to apparently large differences in solutions, and yet they present only one solution. The reader is unable to assess whether the presented solution is optimal. The authors should show how the choice of power correction, firn correction, retracker, elevation change solver, spatial and temporal sampling, spatial and temporal interpolation, and mission cross calibration, influence the final product.

4. Validation. Great efforts have been made by others to acquire independent elevation change measurements in Antarctica, for example NASA Icebrige. The authors should make use of these measurements to evaluate their satellite product, and their estimated error budget, in support of their claims that it offers improved accuracy and is optimal.

5. Comparison to GRACE and ERA. I don't understand why the authors have compared altimeter volume changes to mass changes and precipitation anomalies derived from GRACE and from ERA Interim. These are not equivalent, and so a side-by-side comparison has no meaning. There is potential value in contrasting these measurements, if they are each worked up to a common unit such as mass, but that requires more work.

---

## Referee Comment (RC2) · Anonymous Referee #2 · 27 Jun 2018

This study presents the first complete time series of Antarctic land ice elevation changes obtained by merging all radar and laser altimetry data since the late 1970s. Prior to merging the data from different missions, they reprocessed the radar altimetry data using an OCOG threshold retracker and POCA relocation method of the laser altimetry footprint. The merged data set is spatiotemporally interpolated to produce monthly ice sheet elevation changes with a 10 km resolution for 07/1978-12/2017. The results are presented in various ways, including volume change time series of major drainage basins and annual elevation difference grids and compared those with mass changes from GRACE and precipitation anomaly related mass changes from ERA-Interim.

[Figure]

I really wanted to like this manuscript. The study aimed to solve an ambitious goal by generating a four-decade-long time series of ice sheet changes in Antarctica. However, the authors did not have specific, well-defined objectives, beyond to "identify rapid changes associated, e.g., to snowfall events as well as long-term changes as, e.g. due to changing ice dynamics over nearly four decades". Unfortunately, the manuscript failed to convince me about the successful merging the different altimetry data sets. Also, beyond presenting the results of the new reconstruction, it did not provide any new insight into the behavior of the Antarctic ice sheets. Moreover, the authors did not place their methodology and results in the context of previous work. Multidecadal time series were developed from different radar missions by other studies, such as Fricker et al., 2011; Paolo et al., 2015. While most previous results were limited on ice shelves, where the smooth, flat topography makes change detection easier, lessons learned from those studies should have been summarized and advantages of the new approach should have been presented.

Numerous assumptions and simplifications are employed during the processing. However, the assessment of their impact is missing as no error models, and error estimates are presented. Error estimates could have been derived from rigorous error propagation or by a comprehensive comparison between the multisensor satellite altimetry time series developed in this study and repeat observations for example from OIB airborne altimetry. The only validation example from repeat from repeat kinematic GNSS is over the smooth, high elevation region of the EAIS between 2001 and 2015. Thus, the derived error estimates do not apply for the rugged, undulating coastal regions, where more significant changes occur or for the missions before 2000 (Seasat, Geosat, ERS-1). The ice sheet thickness change reconstructed in the low precipitation zone (LPZ), defined as the area where the average annual precipitation is less than 20mm/yr water equivalent from ERA-Interim indicates a good agreement with the ERA-Interim and small seasonal changes after 2003 (Fig 7.e). However, very large average thickness changes, up to 0.1 m in half a year (!!), with a seasonal pattern suggest unremoved systematic errors before 2003. This large seasonal variation is clearly an artifact that

should be investigated as it can shed light on the errors and their distribution of ERS-1 and ERS-2 altimetry.

The proposed approach includes a series of assumptions, most notably the assumption of linear temporal changes, that is violated over several rapidly changing regions. No attempt is made to assess the impact of these assumptions. The issue of non-linear change is relevant to several steps. (1) Reconstruction of elevation change time series. Equation (1-3) works well for linear temporal trends, for surfaces that can be approximated with planar surfaces within the search radius and when a linear relationship exists between the backscattered power and the corresponding elevation correction. In this case, $res_i$ will provide a measure of random errors. In all other cases, e.g., non-linear temporal trend or non-planar surface shape or non-linear backscatter correction, $res_i$ will include both measurement and modeling errors. Moreover, the non-linear component of the temporal change is determined from $res_i$. Therefore, the detection of outliers is not an easy task and should be solved in an adaptive fashion. The manuscript fails to explain this critical step in sufficient detail. (2) The step of merging the time series from different missions also assumes a linear trend, a linear trend of temporal elevation changes. It is not clear if the linear temporal trend is assumed to be the same for all missions, same for different subsets of missions (e.g., ENVISat, ICESat, CryoSat-2, page 9, lines 24-35, page 10, lines 1-2) or could be different for each mission.

An evident deficiency of the study is that the elevation change is not converted into mass change. Solutions for estimating mass changes from thickness changes have been applied by several authors (see, for example, Shepherd et al., 2012 and 2018, and references therein). Presenting the results as mass changes would enable quantitative comparison of the results with other studies, and thus validating them against those. The comparison of measured thickness/volume changes with modeled precipitation from ERA-Interim (expressed as mass) and mass changes from GRACE gravity have insufficient value because of the complexity of the thickness/mass change relationship.

[Figure]

As mentioned above, the comparison of the ice sheet thickness change time series with ERA interim precipitation anomalies has major caveats, due to the complicated relationship between the two geophysical parameters. However, the availability of the two detailed time series would easily lend itself to a statistical analysis of their relationship, ranging from regression analysis to more sophisticated investigations. Instead of pursuing a statistical analysis, the authors relied on visual comparison. The interpretation and discussion section lists some similarities between reconstructed thickness changes and precipitation anomalies in a somewhat ad hoc fashion. Most of these patterns have already been described by other authors and the manuscript fails to provide a new insight into the surface processes and ice dynamics acting on the Antarctic Ice Sheets.

The results presented as "grids of surface elevation change (SEC) with respect to 09/2010 for each month observed and at a 10 km spatial resolution" (page 10, line 10) and as "elevation changes from year to year" (page 16, line 1). These geophysical parameters do not appear to be carefully designed, they are not explained properly and have misleading names. For example, the SEC is ice thickness (GIA corrected elevation change) rather than ice surface elevation, relative to the reference time, which usually (e.g., for linear trend) changes its sign at the reference time. Thus, regions characterized by thinning have positive values before the reference time and negative values after the reference time (see, for example, Fig. 5.a). As for the elevation changes from year to year, they appear to be differences between average annual ice thickness values of consecutive years. Therefore, they could be called as annual ice thickness rate change. The difference between the ice thickness at the end of two consecutive the balance years might be a better parameter to estimate annual ice thickness change rates as often used by other studies.

The description and interpretation of the change patterns in section 5 (Interpretation and Discussion, pages 18-20) is very difficult to follow. A few simple change would help, such as including an index map that shows the drainage basins (with letters), the

LPZ, and the major geographic names over a background of thickness changes (e.g., Fig. 6.b) and labeling the panels in Fig. 8 and Fig. S6-S7 by the drainage basin/glacier names. The description of the changes should be better organized, both in space and time.

Detailed comments: Throughout the paper the reconstructed changes are described as ice sheet elevation changes. However, changes due to vertical crustal deformations (GIA) have been removed from the reconstructed elevation change rates (page 10, lines 32-34). Therefore, it would be more appropriate to call the parameter ice thickness change.

Abbreviations should be spelled out when they appear first, e.g., ESA, SARIn.

Page 2, lines 15-16: use release numbers instead of "most recent".

Page 3, lines 10-11: add beam limited, i.e., approximately 20 km "beam limited" footprint; lines 14-18: more details are needed to explain on how to find the POCA; line 24: ICE-1 and ICE-2 methods need to be described; line 25: remarkably higher precision than what?

Page 4, line 22: spell out CFI retracker, include reference.

Pages 6, 7: it would work better to explain first why the planar surface approximations are different for the different missions, followed by the equations.

Page 7: outlier detection procedure should be explained in detail.

Page 9: explain the use and effect of the moving median filter.

Page 10, lines 2-4: provide more details on the spatiotemporal smoothing, why was it performed and how effective was it? Line 10: explain the definition of "each month observed". Is there a minimum number of observations or spatial coverage? Line 15: how are the surface elevation change rates determined? Are these average rates determined by straight line fitting in temporal domain?

Page 16, lines 10-12: the error of the trend (slope of the linear fit) is not the standard deviation from the linear fit and can easily be estimated from the data.

Page 18, line 4-6: the long-term trends over Kamb Ice Stream and Totten Glacier have been detected earlier, for example by Zwally et al., 2015.

Page 18, lines 29-34: it is not clear what this statement refers to: "Around kilometer 600 where the profile bends into the main flowline of Totten Glacier, we see a significantly rising elevation. The profiles at different epochs reveal that this is not a continuous change but that there is a distinct jump in the early 2000s." Maybe a different representation and a more detailed explanation would help.

Table 1: $\sigma_{constant}$ is a misleading parameter name – $\sigma_{flat}$ or $\sigma_{noslope}$ might be better. Figure caption should include the type of retracker used, i.e., 10%-threshold retracker from this study. Better yet, a comparison of the performance of the different retrackers (from Fig. 2, Fig. S2) could be compared in this table.

Figure 1. The southern extents of the different radar altimetry missions are not clearly presented in the left panel.

Figure 2. ICE-2 retracker is mentioned in this figure caption only, not in text. Needs more explanation.

Figure 4. Time axis labels should be fixed. Describe vertical axis. Should show the combined time series.

Figure 5. Define the yearly mean surface elevation change.

---

## Editor Comment (EC1) · E. Berthier (Editor) · 5 Jul 2018

Dear authors,

The online discussion of your paper 'tc-2018-49' is now closed. Your TCD manuscript has been thoroughly review by two external referees (+ one short comment) and I want first to acknowledge them for the time they spent on evaluating comprehensively your study.

Both referees and A. Shepherd raised some major issues on the methodology and the significance of your results. In particular, they converge on the fact that your study

lacks a clear description of the methodology, an error estimate and they all regret the unconvincing validation. The general opinion is well summarized by the statement of Anonymous Reviewer #2 "Unfortunately, the manuscript failed to convince me about the successful merging of the different altimetry data sets. Also, beyond presenting the results of the new reconstruction, it did not provide any new insight into the behavior of the Antarctic ice sheets".

In this context, I ask you to answer first to these major criticisms and explain how you would proceed if you were asked to provide a revised manuscript.

Then, and only then, I will decide whether we can move forward with the review process and if a revised paper and a detailed point by point answer is needed.

Best regards,

Etienne Berthier

---

## Author Comment (AC1) · 10 Jul 2018

Dear Etienne Berthier,

Thank you for your comment. We appreciate the thorough and insightful reviews. We think that the comments will contribute to a major improvement of the manuscript. We believe that we will be able, by a major revision of the manuscript, to address all raised concerns. In some more detail, we plan to address the concerns as follows.

Clear description of the methodology: We agree that the methodology has to be described in more detail. This is a straightforward task because naturally all the details

[Figure]

are at our hand. We will give comprehensive details about the parameters, outlier criteria and error propagation of each step in the processing of the SEC grid. However, with regard to readability of the main manuscript, we decided to include this as a section in the supplementary material.

Uncertainty characterisation: Together with the detailed description of the methodology, will describe how we derive uncertainty estimates. We will outline which additional uncertainty is introduced due to the different processing steps, e.g. due to the intermission offset correction or interpolation, and support this estimate by validation with independent data.

Validation: We will follow the reviewers' suggestion to include ICEBridge data into our validation. In fact, comparison with ICEBridge is already implemented in our suite of analysis tools. We note, however, that validation will always remain incomplete. The first flights contributing to ICEBridge level 4 (dH/dt) data are from 2002. We do not have any older data for ground truthing. As another validation approach, we will improve the way we are comparing altimetry results with atmospheric modeling results: We will use results of a firn densification model (FDM) driven by modeled surface mass balance (SMB). We will provide statistical evidence on the degree of agreement of interannual variations from altimetry and FDM, as an upper bound on the uncertainty of either approach (altimetry and modeling) to capture SMB-driven interannual variations. We note that this approach has limited value for validating long-term trends induced by dynamic imbalance or by long-term SMB trends. However, this approach allows us some validation back to the 1970's.

Furthermore, we will follow the suggestion to include a conversion from volume to mass, which allows direct validation against GRACE and SMB modeling results, even though with the inclusion of the additional uncertainty of the volume-to-mass conversion. We decided to include the mass change results in the revised version.

Lack of new insights on ice sheet change: Even though we think that all the potential

insights into ice sheet processes cannot be exploited in this single manuscript, we will illustrate the new insights from our new dataset for two case studies: one addressing the long-term change in Wilkes Land (elaborating on the results shown in Figure 5b) and one on the SMB-driven interannual variations represented on a time series level.

We hope, that these changes will satisfy the reviewers and that they find the revised paper suitable for publication.

Best regards, on behalf of all co-authors, Ludwig Schröder

---

## Editor Comment (EC2) · E. Berthier (Editor) · 17 Jul 2018

Dear authors,

Thanks a lot for your general responses to the referee's comments. Your letter suggests that you are confident in your ability to address their comments in a revised manuscript.

I would like to ask you to submit a point-by-point response to all comments (including the Short Comment) and a revised version (both track-changed and cleaned) of the manuscript. I will send it back to the referees, only if all points have been addressed carefully.

[Figure]

Best regards,

Etienne Berthier
* * *

---

## Author Response (AR1)

**Authors response to the comments of referee #1**

We thank referee #1 for the thorough and insightful review of the manuscript. The reviewer criticized our description of the methodology, the missing information concerning our error estimates, the validation by kinematic GNSS profiles only and the lack of 'numbers to back up most of the statements made'.

We have put much effort in rewriting the respective sections and think that this contributed to a much clearer presentation of the methodology and the results now. In order to describe the methodology in more detail, we have added significantly more details in the supplementary. We think that this is a good compromise between keeping the manuscript itself relatively short for the majority of the readers but having all technical details available for everybody who is interested. Error estimates have always been part of our processing but we agree, they have to be given their space in the manuscript as well. Therefore, we added respectively short descriptions in the manuscript as well as further details in the supplement.

Also the validation with IceBridge has been included now. However, we would like to stress that we are comparing the absolute elevation differences between two epochs. The magnitude of the difference is not important for this validation. Nevertheless, we agree that IceBridge contributes significantly more validation data in areas with complex topographies. Hence, this validation indeed provided important information. We included this validation but added a slope dependency to account for the topography effects.

To provide more 'quantitative facts' we added several correlation coefficients and key numbers. However, one of the major benefits of this work is the high temporal sampling of processes. Single numbers as linear trends, or even acceleration rates, can not always describe the underlying processes adequately.

Besides addressing these key issues, a major change in the revised version is that we have converted the volume changes into mass changes. With respect to some of the specific comments below, this indeed makes the comparison with the external data sets more meaningful, compared to the earlier version of the manuscript. We have decided for a rather straightforward and robust density mask approach, allowing us to still compare the SMBs as independent data set.

In the following we will respond the specific comments one by one.

*Comment 1: Abstract – The abstract is quite long, but not well written. About half of the text is spent discussing the percent coverage for the 25 year and 40 year epochs, however these numbers aren't key scientific results so would be better placed in the data or methods section. Additionally, the coverage stats are poorly defined here, for example is this the percent coverage that of the raw data, plane fit output at whatever grid resolution used, or the extrapolated interpolated result. Without being specific the coverage stats are open to misinterpretation.*
The abstract has been completely redesigned, being more focused. Besides the abstract, more details on our definition of 'observed' cells have been added in Sect. 3.3.3 and 5.2.

*Comment 2: Abstract - The title of the paper states that this paper is about surface elevation change, but there are no key surface elevation change numbers stated in the abstract. Why not, if this is the main purpose of the paper?*
The main focus of this paper are the time series of surface elevation change from multi-mission altimetry. We show that surface elevation changes are much more complex than just a single rate or even an acceleration. These variation are shown on a spatial and a temporal scale in the results section. In our revised version, we added a key value for the total mass change of the AIS since 1992 but also discuss the importance of the time series.

*Comment 3: Abstract – The second paragraph of the abstract lacks any quantitative facts. For example, Pg1 L14 states that surface elevation change shows 'high coincidence' with precipitation anomalies and gravimetry; Pg1 L16 states that there is a 'high level' of agreement; and Pg1 L18 states that 'Geosat coincides very well with. . .'. The authors should replace these generic adjectives with quantitative statistics to back up their statements.*
Respective correlation coefficients have been included in the document, but the respective sentence is no longer part of the abstract.

*Comment 4: Abstract – Pg1 L15 – 'Satellite gravimetry' is the technique, but the authors have presumably compared their elevation results against a derived product, such as mass loss. Edit wording to be precise.*

We agree, but in the revised version, this is no longer part of the abstract.

*Comment 5: Abstract – Pg1 L18 – The Seasat and Geosat altimeters operated at radar frequencies and will consequently penetrate some depth into the snowpack, its therefore not a given that the elevation trend from these satellites should correlate with precipitation anomalies in snowfall. The penetration depth is spatially and temporally variable, influenced by snow density and moisture content. As the radar return originates from a scattering horizon within the snowpack, correlation between precipitation anomalies and rates of elevation change can't prove that the elevation change trends are 'reliable', so the satellite based elevation trends must be verified with a comparable elevation change dataset.*

The penetration depth variations have been accounted for by the backscatter correction in Eq. (2) (except for Seasat, where the time period is too short). The offset corrections align the scattering horizons of the different missions. Section D in our revised supplement shows that the trend-corrected anomalies of the FDM and of the SEC differ by $0.12\pm0.21$ cm for Geosat and $0.26\pm0.32$ cm for Seasat (including one year without observations). Considering that the model is prone to some uncertainties as well, we think that this agreement is remarkable.

*Comment 6: P1 L21 – Sentence wording not correct English, edit required. The wording throughout this paragraph is poor.*

The introduction has been completely rewritten.

*Comment 7: P1 L22 – Does the author mean sequentially rather than concurrently?*

The introduction has been completely rewritten.

*Comment 8: P2 L6 – Edit wording to say precisely what is meant. A long time series would not have prevented Wingham observing negative elevation rates in Dronning Maud Land compared with Flaments positive result for the same area, because as stated the observational time period is different. 'Help reduce the influence of such events' is factually incorrect.*

The introduction has been completely rewritten.

*Comment 9: P2 L10 – Edit wording. Mission calibration doesn't become 'more important', it is maybe more challenging though.*

Changed.

*Comment 10: P2, Fig1 – Add separate colorbar for map of spatial coverage as currently hard to interpret. Just the circle out- line that corresponds to bar color, but really hard to see in pole hole.*

Changed.

*Comment 11: P3 L1 – Edit wording to be more precise. As the raw data from both satellites was acquired in the same pulse limited imaging mode, the use of the word 'mode' to describe a processing choice could be misinterpreted. Additionally I think the two modes authors are talking about are ocean and ice retrackers, however there are actually 3 retrackers available for these missions (ice-1 and ice-2 are separate).*

There seems to be a misunderstanding. The measurements of ERS have been switched between 'ice' and 'ocean' mode. See Paolo et al. (2016): 'To improve performance over the ice sheets, ERS-1 and ERS-2 operated in both a standard 'ocean mode' and a specialized 'ice mode', with mode switching based on an ocean-ice mask. For ice mode, the 64-bin range window (the segment of return echo that is recorded) was four times wider than for ocean mode (116.48 m vs 29.12 m), increasing the chances of capturing return signals over rough topographic surfaces.' We changed the wording to give a more self-contained explanation.

*Comment 12: P3 L3 – Paolo et al presents results over flat ice shelves with zero slope, whereas this paper presents results over an ice sheet, where the most rapidly changing regions are found in the most steeply sloping terrain. The logic that Paolo used to justify including data from different imaging modes therefore may not apply to this paper. The authors should use data from a single retracker which has been shown to be more reliable, or quantitatively justify why including less reliable data improves the quality of the end elevation change result. E.g. via coverage, temporal extent, or reduced error maybe. It follows that a separate error estimate should be provided for the*

*elevation change result derived from different quality input datasets.*

In this comment 'observation mode' and 'retracker' seems to be mixed up again. The retracker is a post-processing step, applied to the observed waveform. In contrast, one of the main differences between the observation modes is the onboard sampling of this waveform (see answer to comment 11). This question obviously refers to the observation modes. Both modes have been treated as independent data sets. For ERS-1 and ERS-2, we used individual a priori uncertainty estimates in the repeat altimetry processing (Eq. 2), estimated individual mission parameters and applied individual offsets. For each epoch where both modes exist, the monthly mean values have their individual error estimate. Hence, in the PLRA averaging step (Sect. 3.3.1), which combines the data, a poor quality of any mode would be reflected by a high RMS and, hence, a low weight in the average. We see no reason why we should remove one observation mode completely. A further discussion to the respective modes has been added in C.2.

*Comment 13: P3 L4 – Specify the size of the bias between both modes, for both satellites.*
This is done later in Sect. 3.3.1. The bias can be seen at Fig. S3.

*Comment 14: P3 L16 – The Helm et al DEM is the ice surface during the first 3 years of CryoSat, however this surface has changed significantly in many regions throughout the 40-year study period. The ice surface of the DEM should evolve temporally to reflect the known elevation change, otherwise the slope correction will not be correct. This effect will be significant in regions such as WAIS which have shown thin at a maximum rates of up to 9 m/yr. Have the authors done this, or if not what is the error on the slope correction that will result from not temporally evolving the ice surface for 40 years?*
The absolute elevation of the topography has only a negligible influence on the location of the POCA. The POCA mainly depends on the relative topography, that is, the variations of topography w.r.t. the mean, over the footprint. If any changes of this relative topography occur, their rates are significantly lower than the rates of the mean absolute topography change (at least in regions that can be observed by pulse limited radar altimetry at all). Hence, we agree that such an effect exists, but its influence on the location of the POCA is negligible.

*Comment 15: P3 L29 – Can the authors quantitatively state how much less sensitive to noise the OCOG retracker is compared to other options, and does this affect all satellites the same way given that the spatial resolution and imaging modes are different. Some retrackers will perform better over different terrain types (sloping or flat), therefore it would be helpful for more details to be provided about the region, time period, and satellites for which this analysis was performed.*
Details on the comparison between functional fit and threshold retrackers have been added to Tab.1. The noise of different retrackers has been discussed in Schröder et al. (2017) as well, to which we refer at the respective locations here. Concerning the comparison between a waveform maximum and a OCOG threshold retracker, Bamber (1994) explain that the single 'maximum bin' is significantly more affected by noise as the squared mean over all bins. However, we did not apply this option, so we cannot provide any numbers.

*Comment 16: P4 L3 – If the CryoSat retracking doesn't exactly replicate the methods in Helm et al, the full details should be detailed in this paper.*
The retracking of the CryoSat-2 SARIn data was the method of Helm et al. (2014). We changed the wording to make this more clear.

*Comment 17: P4 L13 – It is well known (as the authors later state) that anisotropy exists between ascending and descending tracks of radar altimetry data, and indeed many elevation change papers include a term for this in the plane fit solution to exclude any bias from this. When calculating data precision from ascending and descending tracks have the authors performed such a correction, and if so could further detail be provided on its size per mission.*
We did not apply such a correction, but showed how the A-D bias is reduced by our low threshold retracker. Consequently, the precision from ascending-descending crossover differences includes the effect of the A-D bias as well. Table 1 and Fig. S2 show how the precision (including the effect of the A-D bias) is improved for each mission.

*Comment 18: P5 L5 – Edit text to state quantitative statistics about the absolute or percent improvement following their slope correction. Frustrating that 'superior performance' and 'similar improvements' used when a number would be more persuasive.*

Numbers added and text modified.

*Comment 19: P5 L17 – Figure S1 does show a reduction in the anisotropy effect, but the signal is clearly still present in the data. Additionally, smoothing a result with an artefact in doesn't remove the affected data, so this error will clearly have an effect on the end result. If as the authors state recent previously published studies have designed and successfully implemented an anisotropy correction, this paper should add this step to avoid unnecessary error. If the authors chose not to do this, I would ask that the quantify what the affect of not applying the correction is to prove its not discernable.*

Firstly, the mentioned studies use data which have been retracked using a functional fit retracker, where the effect of the A-D bias if much larger (see Fig. S2). Secondly, we do not simply smooth data with a systematic offset. When averaging ascending with descending data (which are both affected by the A-D bias, but with opposite signs), the result will not contain a bias any more. The bias will only affect the resulting uncertainty estimate (see answer to comment 17). We average ascending and descending tracks (with typical cross-track distances of less than 20 km) over 60 km. Hence, we usually average 3x3 ascending and 3x3 descending tracks (over 3 months). This perfect constellation will not always be true but, however, also the alternative, the application of a A-D bias correction might contain some issues. As this discussion belongs to the repeat track processing, we have moved it to C.1 and discussed this point there.

*Comment 20: P6 L25 – If there are differences in the processing methods used for different missions as stated, this should be fully specified in the supplementary material. Use of full parameter names as found in the mission meta data will ensure that the methods and results presented are repeatable.*

These 'parameters' refer to the parameter fit (Eq. 1). We have rewritten the whole paragraph.

*Comment :21 P8 L1 – Spelling*

Obsolete due to edits.

*Comment 22: P8 L1 – Specify the thresholds and variables against which data is filtered out during the elevation change iterative processing. Are the same values used for all missions?*

More details added, moved to suppl. C.1.

*Comment 23: P8 L7 – Provide some detail on how the backscatter penetration correction is calculated and applied. E.g. over what epoch?*

The backscatter penetration correction is applied according to Eq. (2). For each repeat cell and each mission (except Seasat and ICESat) a parameter dBS was estimated. By not including $dBS$ in Eq. (3), the resulting time series are backscatter corrected. This has been explained in more detail in Sect. 3.2 and a further discussion was added to C.1.

*Comment 24: P8 L18 – State threshold used to determine outliers.*

Each processing step is described in much more detail now in the supplement.

*Comment 25: P9 Fig4 – Red colors in this plot do not print well so can't easily differentiate missions. Change color scale used.*

Done.

*Comment 26: P9 L3 – The authors are simultaneously arguing that for all of the more recent missions a spatially variable offset correction must be applied as the offset is spatially variable, while stating that for Seaseat and Geosat the offset correction must be a constant because some of the difference could may be real elevation change. If the later is true, why does this not also hold for the more recent missions? The fact that a spatially variable correction can't be or hasn't been calculated doesn't remove the justification for why its needed.*

For the recent missions we use overlapping epochs. The differences, used to calibrate these missions, refer to the same time (within one month), hence, real elevation changes do not play a significant role here. We believe that the true offsets between Seasat/Geosat and Envisat are spatially variable, just as the offsets between ERS-1/ERS-2/CryoSat-2 LRM and Envisat. However, in contrast to ERS-1/ERS-2/CryoSat-2 LRM, we are not able to estimate the spatially variable offsets for Seasat and Geosat over their region of coverage. This is because Seasat and Geosat have no temporal overlap to the later missions, so that actual elevation changes between the mission times are an additional source of error in the offset estimation. Therefore, our final estimate of the Seasat and Geosat offset is constant in space. The assessed spatial variability of the offsets is in turn included

in the uncertainty estimate and makes the estimate much more uncertain than for ERS-1/ERS-2/CryoSat-2 LRM. It is legitimate to adapt the offset estimation to what is possible, as long as the uncertainties are adapted accordingly. We clarified this in our explanation of the Geosat/Seasat offset estimation.

*Comment 27: P10 L15 – The extrapolation and interpolation steps are not sufficient. It doesn't account for the spatially variable pattern of thinning, which increases towards the coast and is larger on fast flowing ice streams, and the method is not fully described. What is the maximum distance over which gaps are filled? In areas such as the ice sheet edge or on the Antarctic peninsula, which receive exceptionally poor coverage in earlier missions, how are these larger gaps filled? Equally, in order to state EAIS, WAIS, and continent wide thinning rates, the pole hole needs filling.*
We explicitly do not perform an extrapolation to unobserved regions (not at the margins and not in the polar gap as well). This was achieved due to the criterion of different sectors around the cell, which need to contain data. Only cells which are surrounded by observations were filled. During the revision of the manuscript, we modified this criterion to be even more strict. In our final grid, now, we calculate a value only for 10x10 km cells that are within a beam-limited radar footprint of repeat altimetry results. A more detailed description has been added to Sect. 3.3.3 and C.4.

**The 'Results' and the 'Discussion' section has been completely rewritten, so many of the following comments have been considered but do not apply directly to the revised version.**

*Comment 28: P10 L31 – The signal in the 1978 to 1992 map is extremely noisy with lots of variation over short spatial scales. Although the authors assert that 'coherent signal' can be obtained from these missions, to me it looks like the differences are as great if not greater than the similarities between the later data. Generate a difference map, or statistics, to quantitatively demonstrate that the results form the early missions are 'coherent', or similar to those from later missions.*
We agree that this point should have been explained in more detail. Differences to the results over later periods logically arise due to interannual variations. However, as the whole section has been redesigned, this does not apply to the revised version any more.

*Comment 29: P10 L32 – The authors attribute elevation change across the ice sheet to ice dynamics without providing evidence in support of this. Elevation change can be caused by dynamic ice thinning, a snowfall anomaly, change in the scattering horizon, or measurement error, so all of these factors will influence the result not just dynamic thinning alone. Which specific regions are attributed to dynamic change? If based on previous publications, please provide relevant citations. The authors should quantify how much of the elevation change is dynamic, vs all of these other factors, in order to demonstrate that it's the largest contributor.*
In the revised version of this section, this very important remark has been taken into consideration carefully.

*Comment 30: P10 L33 – State what's classed as a short time scale, annual/ sub-decadal/ other?*
This does not apply directly to the revised version any more but has been taken into consideration in the wording.

*Comment 31: P11 L1 – Again maps of elevation change are not evidence of change in dynamic thinning without additional supporting data. The authors need to quantify and rule out the influence of snowfall variations, change in scattering horizon, and error, and a corresponding change in ice speed should also be observed. Without this elevation change due to long term, decadal fluctuations in snowfall, may be mischaracterized as dynamic change. The authors should also state which regions they are referring to.*
The passage has been rewritten.

*Comment 32: P11 Fig5a – Add distance markers to the flow line on one of the maps, hard to tie 5b to locations along it. For example, what distance is the limit of Seasat and Geoset data?*
With regard to this and the following comment, this figure has been replaced (now Fig. 9). Both comments would have been very difficult to apply while still keeping the figures readable.

*Comment 33: P11 Fig5b – Add error bars to this plot.*
See answer above. Figure 9 now contains error bars.

*Comment 34: P12 Fig6 – Provide a spatially variable error map for each of these epochs. So far the results have been presented without any error method described, or measurements included in the plots.*

Respective maps have been included in the supplement.

*Comment 35: P12 L4 – Edit text. The discussions aren't controversial, there is just are just different approaches each of which have advantages and limitations.*

Changed.

*Comment 36: P13 Fig7 – Add error bars to all lines on this plot.*

We added error bars to our altimetry results. Error bars for all data would be hard to identify in the plot.

*Comment 37: P13 Fig7 – Clarify how the % coverage has been calculated, and add coverage labeling to an axis. For Antarctica and the LPZ, I don't see how it can be 100during the 1990's when the pole hole is not observed. Additionally, in the methods its stated that the raw data was originally gridded at 1km resolution, which will result in data gaps of several kilometers between tracks before the CryoSat's precessing orbit comes online. So again I struggle to see how such complete observational coverage is achieved, unless extrapolated and interpolated data is classed as an observation, which of course it isn't. Could the authors clarify?*

The gridding is described in more detail now. The percentage of coverage serves as ancillary information for the interpretation, only. We think a label is not necessary to see when the coverage was almost 100% and when it was only 80%. Instead, we want to keep the plot itself as large as possible.

Concerning the polar gap: The caption says 'of the Antarctic Ice Sheet north of 81.5°S', which means *excluding the polar gap*.

Concerning the distinction between 'observed' and 'unobserved' we would like to stress that the majority of the 'raw' data in fact has beam-limited footprints of 20 km. We process the data at each kilometer but within overlapping circles of 2 km in the parameter fit. We calculate a value for our final gridded result (with a resolution of 10 km) only if the closest data is less than 20 km away (in the TCD version, the applied criteria to decide, whether we calculate a value or not was different, but the effect was similar). Hence, as we do not extrapolate to regions which are not close (in terms of a footprint) to data, we call our final result 'observed'.

*Comment 38: P14 Fig8 - Add error bars to all lines on this plot.*

See response above.

*Comment 39: P14 Fig8 - Add % coverage axis label to plot. Same comment applies about how the coverage calculation is done.*

See response above.

*Comment 40: P14 Fig8 – Add a table in the SOM with the areas of the drainage basin sub-regions used to generate this plot.*

Done.

*Comment 41: P15 Fig9 – I don't understand the figure caption, please rewrite more clearly.*

Removed during revision.

*Comment 42: P16 L1 to 14 – These results sections are very poorly written as no actual results are described! The authors just state what some of the figures show, and leave the reader to do all the hard work of reading off numbers and key statistics, comparing this with numbers they have read in previous studies. Rewrite the results section to present some actual results. I don't think a single elevation change number has been presented in the text yet, despite that being the title of the paper!*

Completely rewritten.

*Comment 43: P16 L9 to 13 – The authors have described what this plot is, but haven't explained why it matters or what the key scientific result is. Either remove figure 9 or explain why its an important addition.*

This is obsolete now.

*Comment 44: P16 F10 – Label y axis of b, presumably count. Edit figure caption to state time period data validated over.*

The validation has been completely revised, this is obsolete.

*Comment 45: P16 L15 – The validation performed in this paper is completely insufficient, and I would argue it leaves the result presented essentially unvalidated. Use of only 19 GNSS profiles, in a region of no known change, over a limited time period and spatial extent, and on unchallenging flat terrain, does not inform the reader about the validity of these results. At a minimum the authors must use a more comprehensive independent dataset, e.g. ice bridge.*

We agree, that a comparison with IceBridge could be interesting and included it.
Nevertheless, we do not think, that the previous validation leaves the results 'essentially unvalidated'. The elevation change is not important here. Our validation analyzes if both data sets see the same elevation change between two epochs. It doesn't matter if this is 1 cm or 20 m. The temporal coverage of IceBridge (2002-2016) is practically the same as for the GNSS profiles (2001-2015). However, we agree that the coverage of more challenging terrain by IceBridge is also interesting. For this reason, we now validate with both data sets and made our validation now slope dependent.

*Comment 46: P17 L2 – This 'validation' cannot be interpreted as an error. A formal error budget based on the altimetry data itself must be documented and added to the plots in this paper. Validation and error estimation are separate things.*

Changed.

*Comment 47: P17 L14 – The authors don't need to limit their validation data to in situ measurements, much more spatially and temporally extensive airborne data is available and this should be used.*

From a satellite point of view, we consider also airborne data as in situ. However, the sentence this comment refers to is about 'the earlier missions'. We would be happy about any suggestions on pre-2000 validation data with more than 'a limited time period and spatial extent'.

*Comment 48: P17 L21 – State the number, don't leave the reader to guess how much elevation change you have measured! Presumably it is different for the peninsula and east Antarctica, so again please present your result.*

Numbers and some more details added.

*Comment 49: P17 L22 – Add figure number.*

Location of the reference changed.

*Comment 50: P17 L23 – Add figure number, or label somewhere. Location of ice streams mentioned hasn't been identified on any plots in this paper. State quantitatively how your numbers compare with this thinning rates presented by Rignot (2006), the time period is different so there should be something new to say.*

The location of the glaciers have been marked in Fig. 10b now. The Rignot (2006) paper uses the input-output method. They map the ice velocity and use these values to obtain ice mass balances. Quantitative numbers for thinning are not given there. Anyways, we added our maximum thinning rates for the respective glaciers to allow for such a comparison in future.

*Comment 51: P18 L2 to 6 – Use statistics to show the agreement, or disagreement between the elevation change and precipitation anomaly. State with numbers what 'significant difference' is that allows ice dynamics to be determined. State how far inland the thinning was in earlier decades vs how far inland it reaches now.*

This passage is largely edited. We now set the plots to be compared (Fig. 14 now) side by side. We also added a whole section concerning the comparison between SEC from a firn model and from our altimetry data (Sect. 4.2). Correlations for the basin mass time series can be found in F.3. The inland thinning was well reported by Konrad et al. (2016), which is cited here.

*Comment 52: P18 L14 – State the threshold used to determine a strong snowfall anomaly. It looks like it varies just as much at different times around other regions in Antarctica.*

Obsolete in the revised version.

*Comment 53: P18 L15 – what distance away from the grounding line were Seasat and Geosat*

*typically able to observe.*

This strongly depends on the topography, the state of the tracking window loop and the orbit direction. It is now discussed concerning Fig. 9 and can also be seen in Fig. S3.

*Comment 54: P18 L20 – The 12 m/yr thinning suggested by Li et al was due to grounding line retreat between '96 and 2013, however the 12 m thinning present in this paper is for 1985 – 2010. Given that the measurements presented in this paper start approximately a decade earlier, if as this paper sats the glacier was already thinning in the 80's then the magnitude and rate are not in agreement with Li et al. Please clarify.*

This section has been completely rewritten.

*Comment 55: P18 L 23 – The authors need to take more care before attributing elevation change to dynamic ice mass loss. There are many signals present in their continent wide maps that may well not be attributed to dynamic ice loss. There is not consensus in the published literature that all Antarctic peninsula elevation change is dominated by ice dynamics, and the authors themselves later attribute a different elevation change signal on the peninsula to precipitation anomalies without providing any more or less evidence that a different process could be responsible (P18 L30). The authors must present quantitative evidence to support their claims either way.*

The wording has been changed to be more precise.

*Comment 56: P18 L30 – GRACE data cannot disentangle whether elevation change is caused by snowfall anomaly or ice dynamics, as ice mass is lost in both instances. Only velocity data can demonstrate whether ice was exported from the catchment at an increased rate, proving ice dynamics. Both dynamic thinning and snowfall anomalies result in mass loss, but gravimetry mass loss measurements don't show which of these two different processes might be the cause.*

This was a misunderstanding. We were arguing that we see an anomaly and that ERA-Interim sees the same, hence it is very likely a snowfall anomaly. GRACE just confirms the anomaly, not the origin. We changed the wording to be more precise.

*Comment 57: P18 L35 – Quantify very well.*

Obsolete.

*Comment 58: P19 L10 – In Fig S7c I can't see any 2002 step in the precipitation time series so its not clear to me that there is good agreement. Again please provide quantitative stats to back this up, rather than just making unsupported qualitative statements. Cite Lenaerts et al 2013 with respect to the 2009 and 2011 precipitation anomaly results as not a new result from this paper.*

Edited.

*Comment 59: P19 L16 – While this appears to be true for 2008/10, there is a more significant accumulation gain in the 1990's that is not visible in the elevation change result at all. This is in part because the authors are comparing different things, snow mass anomaly, vs elevation change. Direct comparison not possible unless elevation change converted to mass change.*

The volume to mass conversion has been included in the revised manuscript. The respective difference in the 1990's, however, is still present. The correlation in Fig. 8a shows that there are regions where the interannual variation of the FDM and of the altimetry do not agree very well. A more detailed analysis what causes this specific disagreement would be very interesting but is beyond the scope of this paper.

*Comment 60: P20 Table2 – There is negligible data coverage outside of East Antarctica prior to 1992, so not valid to include an Antarctic wide number for the '78 to 2017 period in row 3. Remove this number as misleading.*

Done.

*Comment 61: P21 Conclusions – There are no key results from this paper presented in the conclusions. Add a few key numbers.*

Done.

*Comment 62: SOM A.1 – State threshold used to determine if noise is too high.*

This is a flag, contained in the data from GSFC.

We have listed all the flags and criteria for data editing in a descriptive way (which also applies to the following data). For some of the datasets (as from GSFC), the data does not contain

fixed names. The documentation does only contain a description to the parameters. The ERS data comes with auxiliary files containing additional flags, not included in the binary data files. Furthermore, the ERS data contains outlier in the time tags ('time jumps') as reported by the RA L2 Validation Report. Some of them are flagged but we found several outliers in timing also in the remaining data. All those details are very technical and cannot simply be listed as 'flags and thresholds'. We think the commonly used descriptive text is sufficient as it is done by a range of other publications (Smith et al., 2009; Pritchard et al., 2012; Fricker and Padman, 2012; Sørensen et al., 2015; Paolo et al., 2016) (while many others don't mention data editing at all).

*Comment 63: SOM A.1 – State the start and end date for each satellite dataset used.*
Very good point. Table added.

*Comment 64: SOM A.2 – State which retracker the elevation measurements were derived from.*
Our own, see Sect. 2.1.

*Comment 65: SOM A.2 – State the specific name of the metadata flag used to filter out data, and if a threshold was used, state the number that this was set at.*
See above.

*Comment 66: SOM A.2 – Adjust Figure 1 to reflect the actual time period of ERS-1 data used. (same applies for all missions)*
Done.

*Comment 67: SOM A.3 – State which retracker the elevation measurements were derived from.*
Our own, see Sect. 2.1.

*Comment 68: SOM A.3 – State specifically which measurement confidence flags were used, and again if a threshold was used, state the number that this was set at.*
See above.

*Comment 69: SOM A.4 – State specifically which measurement confidence flags were used, and again if a threshold was used, state the number that this was set at.*
See above.

*Comment 70: SOM A.5 – State which LRM retracker the elevation measurements were derived from.*
See above.

*Comment 71: SOM A.5 – State specifically which measurement confidence flags were used to filter data, and again if a threshold was used, state the number that this was set at.*
See above.

*Comment 72: SOM B – Edit title and section text to be more specific as its unclear specifically what the authors have reprocessed? Is it that the elevation measurements have been retracked? Read as a stand alone section I don't know what*
The section has been edited accordingly.

*Comment 73: SOM E S6 and S7 - Add error bars to all lines on this plot. Add % coverage axis label to plot. Same comment applies about how the coverage calculation is done.*
Error bars added to altimetry. Concerning the %-label, we refer to our answer to comment 37.

We have put much effort in rewriting the respective sections and think that this contributed to a much clearer presentation of the methodology and the results now. The revised 'Results' section starts with some examples for the surface elevation changes (SEC) at some selected locations, then presents the spatial pattern of the results over different time intervals and finally (after a conversion to mass) provides different time series of basin scale mass changes. The objective of this manuscript is to show which points have to be considered when combining the different satellite altimetry missions, to prove that our approach did successfully deal with these points and to give some examples for the application of the results. As stated in the revised 'Introduction', it is not possible to fully exploit the whole potential of this data set in this paper. Nevertheless, we think that after the revision, the objectives are much clearer now and, also by including significantly more quantitative results, it provides several new insights. The reviewer argues that 'most of the patterns have already been described by other authors'. We think that the point that our measurements see similar patterns and effects as previously reported by other authors (with different data and methods) is not a weakness but, instead, proves the reliability of our results.

To provide further evidence for the successful merging of the data sets, we added the IceBridge data as well as a comparison of the anomalies with a firn model. In order to better discuss this work in the context of previous work, we have completely rewritten the introduction, which now also gives an overview over different previously published approaches to multi-mission altimetry processing.

We totally agree that the inclusion of our uncertainty estimates contains very important information. Hence, we added several maps, error bars and (mainly in the supplement) a detailed description how we obtain these uncertainty estimates.

The reviewer mentions two 'assumptions and simplifications', which might influence the result. The first (1) is the impact of unmodeled effects on our time series. We have added Sect. C.1 to the supplement which discusses our choice of parameters and the impact of these choices. This section, furthermore, contains more detailed information about our outlier detection.
The second (2) is the *stable linear trend* criterion when calibrating Seasat and Geosat. We have revised this criterion. We now introduce additional information about the data gaps between the missions using a firn model. We have shown now that after the offset correction, the anomalies differ by 0.26±0.32 m for Seasat and 0.12±0.31 m for Geosat, which agrees very well within the respective uncertainty.

In the following we will respond to the specific comments one by one.

*Comment 1: Throughout the paper the reconstructed changes are described as ice sheet elevation changes. However, changes due to vertical crustal deformations (GIA) have been removed from the reconstructed elevation change rates (page 10, lines 32-34). Therefore, it would be more appropriate to call the parameter ice thickness change.*
The mentioned section described how the basin scale ice volume change time series (Fig. 7 and 8 in the TCD manuscript) were generated. Earlier results, which we called 'surface elevation changes (SEC)', were not corrected for GIA. We have changed the structure of the results section, which makes it clearer that the GIA correction was only applied to the 'Ice Sheet mass time series'.

*Comment 2: Abbreviations should be spelled out when they appear first, e.g., ESA, SARIn.*
We have spelled out SARIn. However, the 'The Cryosphere - English guidelines and house standards' say that abbreviation do not need to be defined when they 'are better known than their written-out form (e.g. NASA, GPS, GIS, MODIS).' In our opinion, this applies to ESA as well.

*Comment 3: Page 2, lines 15-16: use release numbers instead of "most recent".*

As mentioned at the end of this paragraph, all details, including the release numbers, are located in the supplement. We modified this paragraph, so this information appears directly after the data center listing now.

*Comment 4: Page 3, lines 10-11: add beam limited, i.e., approximately 20 km "beam limited" footprint;*
Done.

*Comment 5: lines 14-18: more details are needed to explain on how to find the POCA*
Text edited for easier understanding.

*Comment 6: line 24: ICE-1 and ICE-2 methods need to be described;*
The description of ICE-1 has been modified for more clarity. A reference for ICE-2 has been added

*Comment 7: line 25: remarkably higher precision than what?*
Edited.

*Comment 8: Page 4, line 22: spell out CFI retracker, include reference.*
CFI means 'Customer Furnished Item'. However, the written out form is widely unused. Instead, we added a reference.

*Comment 9: Pages 6, 7: it would work better to explain first why the planar surface approximations are different for the different missions, followed by the equations.*
Done.

*Comment 10: Page 7: outlier detection procedure should be explained in detail.*
Explained now in detail in the supplement C.2.

*Comment 11: Page 9: explain the use and effect of the moving median filter.*
Explained now in detail in the supplement C.3.1.

*Comment 12: Page 10, lines 2-4: provide more details on the spatiotemporal smoothing, why was it performed and how effective was it?*
Explained now in detail in the supplement C.4.

*Comment 13: Line 10: explain the definition of "each month observed". Is there a minimum number of observations or spatial coverage?*
Explained in C.4 now as well.

*Comment 14: Line 15: how are the surface elevation change rates determined? Are these average rates determined by straight line fitting in temporal domain?*
This has been explained in detail now in Sect. 5.1.

*Comment 15: Page 16, lines 10-12: the error of the trend (slope of the linear fit) is not the standard deviation from the linear fit and can easily be estimated from the data.*
We are not sure how the reviewer defines 'standard deviation from the linear fit'. This was the formal error of the trend parameter obtained by the fit. However, in the revised version we changed the way how we calculate trends. Instead of a fit, we now use the differences between epochs. This is discussed in Sect. 5.1. This also applies to the rates from the time series. The respective uncertainty is discussed in Sect. F.2.

*Comment 16: Page 18, line 4-6: the long-term trends over Kamb Ice Stream and Totten Glacier have been detected earlier, for example by Zwally et al., 2015.*
For this reason, the sentence continued with 'which was already reported by a range of previous publications (e.g. Wingham et al., 2006; Flament and Rémy, 2012; Helm et al., 2014; Zwally et al., 2015).'

*Comment 17: Page 18, lines 29-34: it is not clear what this statement refers to: "Around kilometer 600 where the profile bends into the main flowline of Totten Glacier, we see a significantly rising elevation. The profiles at different epochs reveal that this is not a continuous change but that there is a distinct jump in the early 2000s." Maybe a different representation and a more detailed explanation would help.*
The whole paragraph has been completely revised.

*Comment 18: Table 1: $\sigma_{constant}$ is a misleading parameter name – $\sigma_{flat}$ or $\sigma_{noslope}$ might be better.*
Changed to $\sigma_{noise}$.

*Comment 19: Figure caption should include the type of retracker used, i.e., 10%-threshold retracker from this study. Better yet, a comparison of the performance of the different retrackers (from Fig. 2, Fig. S2) could be compared in this table.*
Table has been modified accordingly.

*Comment 20: Figure 1. The southern extents of the different radar altimetry missions are not clearly presented in the left panel.*
Figure + caption modified.

*Comment 21: Figure 2. ICE-2 retracker is mentioned in this figure caption only, not in text. Needs more explanation.*
Done.

*Comment 22: Figure 4. Time axis labels should be fixed. Describe vertical axis. Should show the combined time series.*
In the submitted pdf, the time axis was complete. This issue occurred when the journal header was added for the Discussion Paper. The comments concerning the vertical axis and the additional final result have been adapted accordingly.

*Comment 23: Figure 5. Define the yearly mean surface elevation change.*
This is obsolete as the results have been presented in a entirely different way now.

**Authors response to the comments of A. Shepherd.**

We would like to thank A. Shepherd for the very helpful and insightful comments. We have made a major revision of the manuscript, added a detailed description of our uncertainty estimates, a validation with IceBridge data and converted our final results from volume to mass.

*Comment 1: Title. The title is misleading; a minority (25%) of the data set spans 4 decades. It should be modified to explain this or address the majority data set*
The title has been modified so it doesn't imply four decades for the 'whole' Ice Sheet any more.

*Comment 2: Error budget. The authors use the variance of single cycle crossover differences as a measure of error, and conclude that the reduced variance offered by their preferred retracker indicates a de-facto improvement in error. This is misleading, as their conclusion is entirely related to their choice of error metric and is therefore subjective. To conclude an improvement the authors should evaluate each retracker against independent observations of greater and known precision.*
We did this by validating our retracked data as in Schröder et al. (2017). This shows similar improvements. However, as explained in detail in the manuscript, this is a measure for accuracy, not precision. Such a validation imposes systematic errors due to the different sampling of topography, which has to be considered when absolute elevations are important. With respect to elevation change detection, we chose the precision (or 'repeatability') as a measure for uncertainty. This is discussed in Sect. 2.3.

*Comment 3: Methods. The authors discuss that a variety of approaches have been used to derive continental scale elevation change measurements, leading to apparently large differences in solutions, and yet they present only one solution. The reader is unable to assess whether the presented solution is optimal. The authors should show how the choice of power correction, firn correction, retracker, elevation change solver, spatial and temporal sampling, spatial and temporal interpolation, and mission cross calibration, influence the final product.*
A description for our uncertainty estimates has been added which assesses the uncertainty of the respective data. However, we would like to stress that the method of repeat track parameter fit is well established. The choices we made are based one the results of previous publications as cited at the respective places in the manuscript.

*Comment 4: Validation. Great efforts have been made by others to acquire independent elevation change measurements in Antarctica, for example NASA Icebrige. The authors should make use of these measurements to evaluate their satellite product, and their estimated error budget, in support of their claims that it offers improved accuracy and is optimal.*
The validation with IceBridge has been included.

*Comment 5: Comparison to GRACE and ERA. I don't understand why the authors have compared altimeter volume changes to mass changes and precipitation anomalies derived from GRACE and from ERA Interim. These are not equivalent, and so a side-by-side comparison has no meaning. There is potential value in contrasting these measurements, if they are each worked up to a common unit such as mass, but that requires more work.*
The volume-to-mass conversion has been included in the revised version. Instead of ERA, we now use RACMO and the respective FDM.

**References**

[revised manuscript text omitted]
 (09/2010). They represent three-month temporal averages sampled every month and an effective spatial resolution of about 20 km sampled to a 10 km grid. We evaluate our results and their estimated uncertainties by a comparison with independent in situ data sets, results from satellite gravimetry and results from regional atmospheric climate modeling. We illustrate that these time series of surface elevation change (SEC) allow to study geometry changes and derived mass changes of the AIS in unprecedented detail. For some examples as Pine Island Glacier, Totten Glacier, Shirase Glacier (Dronning Maud Land) and Lake Vostok, we demonstrate the benefits of the long time series. Finally, we calculate ice sheet mass balances from these data for the respectively covered regions. A comparison with independent data indicates a high consistency of the different data sets but reveals also remaining discrepancies.

While this paper gives some examples for new insights obtained from the presented multi-mission altimetry analysis, it can not fully exploit all potential applications. This will be the scope of future work with this data set.

**2  Data**

**2.1  Altimetry data used**

[revised manuscript text omitted]

25   ## 3.2 Single-mission time series

After fitting all parameters according to the multi-mission model (Eq. 2), we regain elevation time series by recombining the parameters $a_0$ and $dh/dh$ with monthly averages of the residuals $(\overline{res})$.  For each month $j$ and each mission $M$, the time series is

30   constructed as

$$
h_{j,M} = a_{0,M} + dh/dt(t_j - t_0) + \overline{res_{j,M}}.
\tag{3}
$$

 The elevations $h_{j,M}$ all relate to the cells center and are corrected for

time-variable penetration, as the parameters of the topography slope and backscatter correction are omitted in this recombination. 
[revised manuscript text omitted]
 datasets shows that these estimates seem reasonable. In the comparison with the GNSS profiles, the relatively low differences, even in regions which imply a higher uncertainty, are likely just incidental for the small sample of $\delta\Delta h$ along the GNSS profiles.

In conclusion, this validation shows that remaining systematic biases (originating from satellite altimetry or the validation data) are on a centimeter level only and that our uncertainty estimate is realistic. However, we have to stress that only altimetric SEC within the interval 2001-2016 can be validated in this way. For the earlier missions, no spatially extensive high precision in situ data are available to us.

**4.2 Firn model**

Another data set which covers almost the identical spatial and temporal range as the altimetric data is the firn thickness data set of the IMAU Firn Densification Model (FDM Ligtenberg et al., 2011), forced at the upper boundary by accumulation and temperature of the Regional Atmospheric Climate Model, version 2.3p2 (van Wessem et al., 2018). Before we can compare this model to our SEC results, however, it is important to mention that the FDM only contains elevation anomalies. A long-term elevation trend over 1979-2016, e.g. due to changes in precipitation on longer time scales (Thomas et al., 2015) would not be included in the model. Furthermore, due to the nature of the model, it cannot give information about ice dynamic thinning/ thickening. Hence, to compare the FDM and the SEC from altimetry, we first remove a linear trend. This is performed for the period 1992-2016. The trends are only calculated from epochs where both data sets have data, i.e. in the polar gap this comparison is limited to 2003-2016 or 2010-2016, depending on the first altimetry mission providing data here. After the detrending, the anomalies are used to calculate correlation coefficients for each cell, depicted in Fig. 8a. Figure 8b shows the RMS of these anomalies from the altimetry data, representing the magnitude of the seasonal and interannual variations. Comparing the two maps shows that the correlation is around 0.5 or higher, except in regions where the magnitude of the anomalies is small, i.e. where the signal-to-noise ratio of the altimetric data is low. This relationship is depicted in Fig. 8c, where we see that for the vast majority of cells the correlation is positive. For anomalies with a RMS>0.5 m, the average correlation is between 0.3 and 0.6.

Anomalies against the simultaneously observed long-term trend (1992-2016) can also be computed for earlier epochs. Assuming no significant changes in ice dynamics here, these anomalies allow a comparison of Geosat and Seasat with the FDM. The median difference between the anomalies according to Geosat and the anomalies according to the FDM amounts to 0.12±0.21 m (see Fig. S6). Considering that this difference is very sensitive to extrapolating the respective long-term trends, this is a remarkable agreement. With a median of 0.26±0.32 m, the difference between anomalies from Seasat and from the FDM is larger, but this comparison is also more vulnerable to potential errors due to the extrapolation. As the FDM starts in

[Figure]

**Figure 8. a)** Correlation coefficient between the SEC anomalies of the altimetry grids and the FDM over 1992-2016 after detrending. **b)** RMS of the detrended anomalies of the 1992-2016 altimetry time series. **c)** Correlation coefficient plotted against the RMS. The point density is color coded from yellow to blue. The black dots show the binned mean values.

1979 while Seasat operated in 1978, we compare the Seasat data with the FDM anomalies from the respective months of 1979, which might impose additional differences. Finally, the FDM model has its own inherent errors and uncertainties. Therefore, only part of the differences originates from errors in the altimetry results.

**5 Results**

**5.1 Surface elevation changes**

Some examples for elevation change time series in the resulting multi-mission SEC grids are shown in Fig. 9 (coordinates in Tab. S2). For Pine Island Glacier (PIG, Fig. 9a), we observe a continuous thinning over the whole observational period since 1992 (Seasat and Geosat measurements do not cover this region). Close to the front (point D) the surface elevation decreased by -45.8±7.8 m since 1992, which means an average SEC rate of -1.80±0.31 m/yr. The time series reveals that this thinning was not constant over time, but accelerated near the grounding line (point D and C at a distance of 40 km  ) around 2006. Also the points at greater distances from the grounding line (B at 80 km, A at 130 km) show an acceleration around 2006. After 2010, the thinning rates at near front decelerate again. For the period 2013-2017, the rate of -1.3±0.8 m/yr is very close to the rate preceding the acceleration. In contrast, further inland the thinning did not decelerate so far and is still at a level of about -1.2 m/yr. Hence, for the most recent period (2013-2017) the elevation at all points along the 130 km of the main flow line is decreasing at very similar rates. A similar acceleration near the grounding line, followed by slowdown, is observed by (Konrad et al., 2016). The onset of this acceleration coincides with the detaching of the ice shelf from a pinning point (Rignot et al., 2014). After that speedup terminated around 2009, the grounding line position was relatively stable (Joughin et al., 2016), which agrees with the elevation changes in our observations.

[revised manuscript text omitted]

5   In order to determine the effect of the SEC on global sea level, they are converted to ice mass changes. In a first step, all time series are corrected for glacial isostatic adjustment (GIA) using the IJ05_R2  model (Ivins et al., 2013). This GIA model predicts an uplift of 5 mm/yr near the Antarctic Peninsula and rates between -0.5 and +2 mm/yr in East Antarctica.

10   Furthermore we applied a scaling factor $\alpha = 1.0205$ to account for elastic solid earth rebound effects (Groh et al., 2012). We multiply the resulting ice sheet thickness changes by each cell's area and apply a density according to a firn/ice mask (McMillan et al., 2014, 2016), depicted in Fig. S8, to obtain a mass change. In regions where ice dynamic processes are assumed to be dominating (e.g. in Amundsen Sea Embayment, Kamb Ice Stream or Totten Glacier), we use a density of 917  kg/m³. Elsewhere, we apply the density of near-surface firn, obtained from firn modeling using atmospheric forcing (Ligtenberg et al., 2011). We have chosen this straightforward and robust method here, instead of using modeled temporal variations of the firn layer

Volume change of subregions north of 72°S for several East Antarctic drainage basins from our combined altimetric time series (blue). The respective time series of mass change from GRACE (red) and precipitation anomaly (pink) refer to the scale at the right. The gray color in the background displays the fraction of the area covered by observations.

[Figure]

**Figure 12.** Mass change ($\Delta M [Gt]$) of the individual drainage basins north of 81.5°S from our combined altimetric time series (blue), GRACE (red) and SMB (orange). The error bars show the uncertainty estimate $\sigma_\Sigma$ of the altimetry data according to Sect. F.2. The gray color in the background displays the fraction of the area covered by altimetry (up to the top means 100%).

[Figure]

**Figure 13.** Mass change of subregions north of 72°S for several East Antarctic drainage basins from our combined altimetric time series (blue), GRACE (red) and SMB (orange). The error bars show the uncertainty estimate $\sigma_\Sigma$ of the altimetry data according to Sect. F.2. The gray color in the background displays the fraction of the area covered by altimetry (up to the top means 100%).

5 (as e.g. Zwally et al., 2015; Kallenberg et al., 2017) in the volume-to mass conversion. This allows us to compare the time series from altimetry with time series from SMB modeling.

Cumulated mass anomalies over larger regions such as drainage basins or even the total AIS are obtained by summing up the results accordingly. Therefore, we used the basin definitions by Rignot et al. (2011) (updated for Shepherd et al. (2018) , see Figs.  10a and 14b). Cells containing no valid data after the gridding (as e.g. where not enough observations were available, in the polar gap or where rocks are predominant) are not considered here. Uncertainty estimates were obtained by propagating the respective uncertainties of the SEC, the  GIA and the
5

[revised manuscript text omitted]

10 2008 and 10/2009, we obtain a mass difference of 116.6±27.0 Gt from altimetry, 109.4 Gt from SMB and 113.4  Gt from GRACE. The high agreement with the SMB indicates that this mass gain is caused by snow accumulation. In most of the basins, we observe similar high agreement in the short-term variations. A good example for the different components of the total mass change signal is the Getz and Abbot region (F-G). While all techniques observe a significant mass loss between 2009

15 and 2011, the  SMB does not contain a long term trend, as observed by altimetry and GRACE. In some regions, however, there are also significant discrepancies between the different data sets. The poor sampling of the northernmost APIS (I-I") by altimetry is a good example for the limitations of this technique. In George V Land (D-D'), the agreement during the GRACE period is

5 reasonable, while the mass gain, indicated by the SMB in the early 1990s is not revealed by the altimetry time series.

Over the last 25 years our data indicate a clearly negative mass balance of -2068±377 Gt for the AIS (Fig.

10 11a). This is mainly a result of the mass loss in the WAIS over the last decade. In contrast, the EAIS has been very stable over our observational record (120±121 Gt between 1992 and 2017). The time series of the APIS contains large uncertainties due to many unobserved cells. Mass change rates for selected regions, obtained from the differences over a specific time interval, and their respective uncertainties are given in Tab. 2. We calculated separate trends for the area north of 72°S, which is covered

15 by all satellites, north of 81.5°S which is covered since ERS-1 and for the total area, which is (except the 500 km diameter polar gap) covered since CryoSat-2. The observed area shows that 96.4% of the cells, classified as ice sheet north of 81.5°S, are successfully covered by observations of ERS-1. Cells without successful observation occur mostly at the APIS, where only 61% is covered with data.

20

From the overall mass loss of -2068±377

25

Table 2.  Mass change  rates for  different regions of the  Antarctic Ice Sheet  and  and the Antarctic Peninsula (APIS) different time intervals. The  sizes of the  observed area  refer to  all cells classified as ice sheet in the respective region (, if stated, limited by the  given latitude).

| region | area [$10^3$km$^2$] | | dM/dt [Gt/yr] | | | |
|---|---|---|---|---|---|---|
| | total | observed | 1978-2017 | 1992-2017 | 1978-1992 | 1992-2010 |
|  AIS | 11892 |  11630 | - | - | - |  |
|  EAIS |  9620 |  9413 | - |  4.8 |  4.9 |  |
|  WAIS |  2038 |  2008 |  1.4 |  1.8 |  2.6 |  |
|  APIS |  232 |  208 | - | - | - |  |
|  AIS (<81.5°S) |  9391 |  9053 | - |  -84.7±3.0 15.5 |  |  20.3   - |
|  EAIS (<81.5°S) |  7764 |  7555 | - |  4.9±0.7 5.0 |  | 8.0±0.6 6.2 |
|  WAIS (<81.5°S) |  1394 |  1358 | - |  -91.7±10.3 | - |  -69.4±7.1 13.1 |
|  APIS (<81.5°S) |  232 |  142 | - |  2.1±2.5 8.9 |  | 2.8±3.8 12.3 |
| EAIS (<72°S) | 2779 |  2274 |  1.5±1.4 5.8 |  -3.4±1.7 4.0 |  12.1±2.5 17.4 |  0.0±3.9 4.9 |

For the APIS (<72°S), the very sparse observations of Seasat and Geosat did not allow calculate a reliable trend.

~~measurements. Here, we only work with elevation differences w.r.t. a reference epoch. As we want to analyze the temporal variability, we compare elevation changes over the same time observed by both techniques . For each crossover difference between kinematic profiles from different years we compare the differences of the respective altimetric SEC epochs in this location ($\delta\Delta h = \Delta h_{KIN} - \Delta h_{ALT}$). Figure 7 shows the results of this validation. We obtain an overall agreement of 6±16or an RMS of 17.7. As both techniques should observe the same elevation change,difference $\delta\Delta h$ can be interpreted as an error measure. It can be expressed as~~

$$\delta\Delta h^2 = 2 \cdot RMS_{ALT}^2 + 2 \cdot RMS_{KIN}^2$$

 extended material in Shepherd et al. (2018) shows that there are still some discrepancies between the different techniques to determine the AIS mass balance. For the  time interval 2003-2010 (Extended Data Table 4 in Shepherd the Input-Output method obtains a rate of -201±82

10 ~~that no significant additional uncertainty has been added due to the combination. Small positive or negative biases in some regions of the map could be attributed to remaining antenna height offset errors in one of the GNSS profiles, but might also originate from remaining penetration effects of the radar signals. The very high agreement between the two datasets shows that our processing successfully eliminated the biases between the different satellite altimetry missions and thus provides reliable results~~ Gt/yr for the AIS, while the mass balance rates, aggregated from satellite gravimetry (-76±20 Gt/yr) and from

15 altimetry (-43±21 Gt/yr) agree much better with our result for the AIS (<81.5°S) between 2003 and 2010 of -64.7±24.9 Gt/yr.

**6 Discussion**

**6.1 Multi-mission SEC time series**

The single-mission time series, obtained in Sect. 3.2, contain satellite-specific calibration biases as well as offsets due to the

20 specific sampling characteristics of different sensor types. In order to form a consistent SEC time series, these biases needed to be determined and corrected. A comparison with in situ data showed that there are no significant offsets between elevation changes from our multi-mission altimetry data and the validation datasets. This comparison, however, could only validate our data in the interval  2001-2016. A quality control for the whole time span was performed by a comparison with a firn model.

25 The correlation of the detrended data sets shows that especially for regions where the interannual variation is large (compared to the measurement noise of the altimeters) both time series agree very well. This comparison even provided independent estimates for the error of the early missions. The average differences between the detrended time series of the FDM and the SEC show that the observations of Geosat and even of Seasat agree with the model results within a few decimeters. For SECs of up to several meters w.r.t 2010 (see Fig. 6), this means that also the older data can be used to calculate elevation change rates

30 with an accuracy better than a centimeter per year (see Fig. S7a). Unfortunately, in coastal DML west of the ice divide A', the data of Seasat and Geosat are very noisy. This due to the mountain ranges just north of 72°S, which lead to many losses of lock of the measurements all the way across this part of the ice sheet. The same applies to the measurements at the APIS.

**7**

**6.1 Surface elevation changes**

The  mean rates of elevation change in Fig. 10 show the regions which experience a significant thinning (Amundsen Sea Embayment, Totten Glacier) or thickening (Kamb Ice Stream) which was already

5

 reported by previous publications (e.g. Wingham et al., 2006b; Flament and Rémy, 2012; F

. By combining all the single missions consistently we analyze long-term changes over the full time period covered. For 79% of the area of the AIS, this means a time span of 25 years. For 25 %, mainly the coastal regions of East Antarctica, even 40 years are covered. We assume that these long-term

10  trends are significantly less affected by short-term variations in snowfall than a trend from a single mission.

The benefits of a seamless combination of the time series are demonstrated in Fig. 10. The time intervals for the elevation changes are independent of the

15 observational period of a single mission. This is necessary to analyze processes which occurred close to the transition between different missions. A good example of the advantage of such long time series are the elevation changes caused by the accumulation

20 events in DML. Figure

~~The different sub-intervals in Fig.10c-e show that the spatial pattern of elevation changes in East Antarctica is not constant over time. During 1978-1992 the East Antarctic sector west of the Amery Ice Shelf (A-B) was losing volume while in the eastern sector (C-D') the rates were mainly positive. For the interval 1992-2010 we see positive rates in Coats Land and western Dronning Maud Land (J''-A')in Princess Elizabeth Land (C-C') while elsewhere, the rates are very close to zero (except for the dynamically thinning~~

5  2011. The mission lifetime of ICESat ended in 10/2009, CryoSat-2 provided the first measurements in 07/2010. Only Envisat covered both events but here, the orbit was shifted in 10/2010, resulting in different repeat track cells covered before and after the orbit shift. We merged all these missions as described in Sect. 3.3, which allows us to analyze the full time series. Comparing the elevation change from altimetry e.g. at point A in Fig.9c of 0.55±0.50 m with the change modeled using the

10 FDM (0.48 m between 2008 and 2012) is a good example of successfully cross-validating these two data sets. Figure 8 shows the degree of agreement over the entire AIS.

As these elevation change rates alone do not contain any information on their origin, additional data are needed for improved process understanding. Figure 14 shows SEC rates for the interval 2002-2016 (March-September respectively) from altimetry and the FDM and respective rates of ice mass changes from GRACE. These maps show that the elevation gains in DML

15 and Enderby Land agree very well with the firn model, which implies that increased snow accumulation during this period is responsible for the thickening. For Princess Elizabeth Land (C-C')~~. In the supplementary material Fig. S5 we calculated similar rates for the precipitation anomalies. These precipitation induced rates show a very consistent pattern compared to the elevation changes in East Antarctica. For the interval 2010-2017 they show that the strong mass losses in Princess Elizabeth Land can be related to a similar decline in precipitation. Also in Dronning Maud Land or Wilkes Land, the rates over the~~

[Figure]

**a) Firn Model**  **b) Altimetry**  **c) GRACE**

[m/yr]  [m/yr]  [m w.e./yr]

−0.30 −0.15 0.00 0.15 0.30   −0.30 −0.15 0.00 0.15 0.30   −0.30 −0.15 0.00 0.15 0.30

[revised manuscript text omitted]
 for ICESat this is true in this case, as cloudy conditions are not unusual in this region. But even when enough valid measurements would have been available, the fit of a planar surface over a diameter of 2 km would have been very challenging in the initial repeat altimetry processing here. Our approach is designed to provide valid observations over the majority of the AIS. Under the challenging conditions of the APIS, modifications such as a smaller diameter or more complex parametrization of the surface would surely help to improve the results. Furthermore, we did not calculate a SEC for cells that are further away than a beam-limited radar footprint from valid measurements. In order to interpolate or even extrapolate the results to unobserved cells, advanced gridding methods such as kriging, especially with the help of additional data sets (Hurkmans et al., 2012), would be advisable. In contrast, here we concentrate on the observed cells only.

The differences of the yearly averages of SEC in Fig.**??** and the respective variations of precipitation in Fig. S8 reveal the spatial signature of the interannual variations. Even for the early missions Seasat and Geosat they show a very consistent picture and hence again demonstrate that these missions can provide important information to extend the observed time interval to a maximum.Between Seast and Geosat, we see elevations gains along the East Antarctic coast between 0°E and 70°E while west of the Amery Ice Shelf (70°E-150°E) the differences are negative. This is confirmed by the precipitation data for most of the area, even if the data here do not include 1978. Elevation gains in coastal western Enderby Land (40°E-50°E), which are very prominent in the precipitation data, are not well resolved by the early altimetry missions due to the distinct topography. Nevertheless, we see some positive elevation changes there too.In 1986, we observe a positive anomaly in Princess Elizabeth Land and Wilkes Landwhich coincides with an increased snowfall. A very similar spatial pattern repeats again in 1996, 2001, in 2004/2005 and in 2009. Mémin et al. (2015) already observed such periodic elevation anomalies using Envisat and calculated a frequency of 4.7 years. They identify the Antarctic Circumpolar Wave as a main driver for this periodic signal but also note that it is superimposed by ENSO. This might be the reason why in 2013 another occurrence, which would by expected by the periodicity, is unusually weak.

In contrast to the highly variable coastal regions, the interior of the EAIS is very stable. As it can be seen from Fig. 11e the surface in the LPZ varied between 5 and -8For the EAIS (Fig. during the last 25 years. A very slightly positive rate is consistent with the precipitation anomalies over this period. Zwally et al. (2015) observed a positive rate in this region as well and explain it by a continuing dynamic response to increased accumulation since the early Holocene. However, their estimated rate of about 211b) we see significant differences between the time series of mass changes from altimetry and from GRACE. For the time interval 2002 to 2016 (see Sect. would result in an elevation increase of 0.5F.3), the mean rate from altimetry (9.6±6.9 over the past 25 years, which clearly contradicts our results. Furthermore, kinematic GPS measurements on snowmobiles around Vostok station (Richter et al., 2014) yield an elevation change rate of 0.1Gt/yr) is mainly dominated by the accumulation events in 2009 and 2011. In contrast, the GRACE data imply an average mass gain of 42.1 0.5Gt/yr over this time interval. Especially after 2011, the differences become very prominent in the time series. The respective mass changes for the individual basins (Fig. for the period 2001 to 2013 which is in every good agreement with our altimetric time series12) reveal that this difference in the signals can be attributed to DML and Enderby Land. This might be a sign for dynamic thickening. Here, all elevation changes have been converted to mass using the density of surface firn. If a part of the positive elevation changes in this region indeed would be caused by ice dynamics, this would lead to an underestimation of mass gains from altimetry compared to gravimetric measurements. The results of the Bayesian combined approach of Martín-Español et al. (2017) also suggest a small dynamic thickening in this region. Rignot et al. (2008) observed no significant mass changes in this region between 1992 and 2006 using the input-output-method. Gardner et al. (2018) compared present day ice flow velocities to measurements from 2008. They obtain a slightly reduced ice discharge in DML (which would support the hypothesis of a dynamic thickening), while they observe a small increase in discharge for Enderby Land. Part of this misfit might also be explained by remaining processing issues in the GRACE processing (e.g. the GIA correction). Hence, we conclude that further work is needed to identify the origin of this discrepancy.

**7 Conclusions**

In this paper we presented an approach to combine different satellite altimetry missions, observation modes and techniques. The reprocessing of the conventional pulse limited radar altimetry  ensures that two fundamental steps in processing of radar ice altimetry, the waveform retracking and the slope correction, are  performed consistently. Furthermore, we showed that the advanced methods, used in this processing, improved the precision by more than 50%, compared to the widely used standard products. The validation with in situ measurements and the comparison with the IMAU-FDM shows that inter-mission offsets have been successfully corrected and that the uncertainty estimates for our resulting monthly multi-mission SEC grids are realistic.

 We analyzed the resulting time series and found that they provide detailed insight in the evolution of the surface elevation of the Antarctic Ice Sheet. From the combined SEC time series we calculated the long-term surface elevation change  over the last 25 years.  Due to Seasat and Geosat, observations in the coastal EAIS date back until 1978 , covering four decades. The unique data show that large parts of the East Antarctic plateau are very close to equilibrium, while changes over shorter time intervals identify interannual variations, which cannot be identified in long-term trends and are mostly associated with snowfall anomalies.

~~Besides the linear rates we were able to create monthly multi-mission time series which allow to study the evolution of the ice sheet volume over the full time span covered by any of the measurements. Peaks and steps in the altimetric SEC time series agree very well with the corresponding cumulated precipitation anomalies from ERA-Interim as well as the mass balance time series from GRACE. Note that we do not expect a perfect physical correlation between the three records. Making them fully comparable would require a sophisticated separation between surface and ice dynamic processes and the consideration of firn compaction (Ligtenberg et al., 2011; Li and Zwally, 2011). Both are beyond the scope of this paper. Nevertheless, the comparison of the different datasets shows, that our methodology successfully eliminated the biases between the different missions and observation techniques. We do not see any jumps in the time series, which could be interpreted as a calibration bias.~~

The monthly mass time series show that the AIS (excluding the polar gap within 81.5°S) lost an average amount of mass of -84.7±15.5 Gt/yr between 1992 and 2017. These losses accelerated in several regions and, hence, for 2010-2017 we obtain -137.0±24.9 Gt/yr for the same area. The comparison of the altimetry-derived mass changes, integrated over different basins and regions of the ice sheet, with SMB and GRACE shows high consistency of the different techniques. A correlation coefficient between the

mass anomalies from altimetry and from GRACE of 0.96 (for the time interval 2002-2016, see Tab. S4) indicates the excellent agreement of the observed interannual variations. The respective correlation with the SMB anomalies (0.60 for 1992-2016) is

20 comparatively lower but still indicates a high agreement. In the APIS, differences between the mass time series of the different techniques arise mainly due to the poor spatial sampling of the altimetry data, while for the EAIS, the remaining discrepancies to mass time series from GRACE might be explained by the density mask used. These remaining issues and open questions should be addressed in future work in order to further reduce the uncertainty of the estimates of the mass balance of the AIS. The recently launched laser altimeter ICESat-2 promises a new milestone in ice sheet altimetry. We believe that our

25 multi-mission combination approach can provide an important tool to combine the extremely high resolution of this mission with the long time period, covered by the previous missions.

[revised manuscript text omitted]

---

## Referee Report (RR1)

**Overview**

Schröder et al. (2018) use a suite of radar and laser altimetry measurements to calculate the volume and mass change of the Antarctic ice sheet. The paper works to improve our understanding of changes over multi-decadal time-scales that puts in context measurements from other missions, such as the GRACE mass change time series. The work presented by the authors falls within the scope of *The Cryosphere* and could make an interesting contribution to Antarctic volume and mass balance studies. The changes made to the manuscript are substantial and greatly improve the presentation of the results. However, there are a number of issues that should be resolved before I can recommend the publication of this manuscript.

**General comments**

- Combining laser altimetry estimates with radar altimetry estimates is a non-trivial task (e.g. Fricker and Padman, 2012). Not simply due to the differences in footprint diameter but by potentially measuring different surfaces. The comparisons with NASA Operation IceBridge in West Antarctica and the Peninsula are helpful for this but it was difficult to discern the local discrepancies between the measurements as presented.

- The paper can be still be difficult to follow at times. The manuscript could be reorganized and restructured to not repeat information between sections.

- The data and methods sections could also still be more detailed. Are the GRACE basin time series results simply an available product? If so, it could include a citation for the data repository. If not, should detail the GRACE processing scheme as you start to diagnose specific uncertainties in time-variable gravity measurements.

**Line-by-line comments**

**Abstract** This is much improved from the previous versions.

**P1, L14–15** I would cite Shepherd et al. (2012) as a reference the time-variability of Antarctic surface elevation change.

**P12, L9** Remove "Also" from the start of this sentence.

**P2, L18** "Finally, we merge all time series from both radar and laser altimetry"

**P2, L22** What do you mean by "three-month temporal averages sampled every month"? A moving average of the elevation data with a three month sliding window?

**P2, L24** "independent in-situ and airborne datasets, satellite gravimetry estimates, and regional climate model outputs".

**P2, L25** I wouldn't use SEC as an acronym here and elsewhere

**P2, L26–27** I would phrase it something ilke "The recent elevation changes of Pine Island Glacier in West Antarctica, Totten Glacier in East Antarctica, and Shirase Glacier of Dronning Maud Land in East Antarctica are put in context with the extended time series from radar altimetry"

**P2, L30–31** I would remove these two sentences ("While this paper...").

**P3, L3–4** I would place the release numbers of each dataset in the main text.

**P3, L8** Should at least include citations for the pre-processing steps used to remove corrupted ICESat returns within this section.

**P3, L14–20** Were the ocean and ice modes both used everywhere or was the ocean mode used only in the interior plateau? If different regions were used, a map would be beneficial.

**P4, L4–6** Could include detail on the advantages and disadvantages of the "relocation" and "direct" methods. Is the "relocation" method valid over the entire timespan of measurements? The ice sheet has changed surface shape over this time period and a single DEM, particularly one from the latter period of observations, may not be accurate.

**P4, L11–12** Is this sentence about optimization necessary? Could simply state the differences from prior work.

**P4, L12–13** Will there always be a unique determination of the POCA location with this methodology?

**P5, L1–8** This is a good review of the effects of spatial aliasing. However, the penetration of the radar signal into the surface is another bias term that would potentially affect rate estimates. Could add a caveat.

**P7, L9–11** At least for laser altimetry, 1km may be too coarse of resolution for the assumption of planar surfaces in coastal regions (Markus et al., 2017). Could include additional higher order surface shape terms.

**P9, L1–2** Rewrite to something similar to "Over regions of flat topography, such as the interior of East Antarctica, the weights between PLRA and ICESat are comparable".

**P10, L6** The use of FDM is an interesting approach. I would add the caveat that changes in ice dynamics would not be captured by this metric.

**P11, L5–10** Is applying a uniform single bias just forcing the linearity condition to regions where it might not be viable? The distributions of biases in the supplementary material appears to be not random but fairly spatially correlated.

**P11, L24** These instruments could be detecting different surfaces beyond the topographic sampling discrepancy.

**P12, L1–2** Could possibly use an interpolation scheme with tension or inherent smoothing.

**Figure 7** Difficult to discerns the differences from GNSS and Operation IceBridge with these maps. Lots of dots area overlapping. Is the color scale in a) and d) the same as in b) and e)?

**P14, L2–3** Could add possible explanations of the difference between the airborne altimetry measurements and the combined estimate.

**P14, L4–8** Could expand to note the potential error sources between GNSS and altimetry. This seems like a fairly large amount of variability for East Antarctica. Could potentially use other datasets for validation as well.

**P14, L10** Remove "we have to stress that"

**P14, L12** I would change it to "publicly available" versus "available to us".

**P14, L15–17** Remove "Before we can compare this model to our SEC results, however, it is important to mention that"

**Figure 8** The RMS with the FDM seems surprisingly high.

**P16, L3** In terms of spatial resolution?

**P17, L15** "rates within uncertainties and very close to zero"

**L17, L17–22** The discrepancy with Zwally et al. (2015) is an important finding.

**Figure 10** Would the early mission data over the Antarctic Peninsula be at all viable? The Peninsula is particularly difficult to measure with radar altimetry (Shepherd et al., 2012).

**P17, L31–P18, L4** This paragraph repeats a lot of information provided previously about individual glaciers. Could be merged.

**P17, L 34** Remove "too"

**P19, L7** "The surface elevation time series is converted into ice mass changes in order to determine their effect on global sea level."

**P19, L7–8** "The elevation time series are corrected for uplift rates related to glacial isostatic adjustment (GIA) using coefficients from the IJ05_R2 model (Ivins et al., 2013)."

**P19, L9–10** The elastic effects would likely be much more localized than the outputs from a GIA model, and would be related to the modern-day change rather than the unloading since the LGM.

**Figure 11,12** As the area of observation changes with time, would it be more appropriate to calculate the average surface mass density (kg/m$^2$) change rather than the total mass change of these sectors (i.e. divide the resultant mass by the total area)?

**P21, L7–8** "We integrate our measurements over larger regions to calculate the cumulative mass anomalies for individual drainage basins and major Antarctic sectors (AIS, WAIS, EAIS, APIS). Our basin delineations are from Rignot et al. (2011a), which have been updated for the second Ice Sheet Mass Balance Intercomparison Exercise (IMBIE-2, Shepherd et al., 2018)."

**Table 2** Is the area of observation constant for these volume rates?

**P22, L7–9** Same GIA model outputs used to correct the GRACE data?

**P11, L4** What is the reference period used to calculate SMB anomalies? 1979–2017?

**P22, L17** Variations and attribution of Cp-D mass change shown here are supported by Velicogna et al. (2014) and Li et al. (2016).

**P22, L18–19** "There is less agreement in regions where surface mass balance change may not be dominant, such as of the Getz and Abbot regions (F-G) in West Antarctica."

**P22, L20–21** I'd suggest something like "In some regions, there are also significant discrepancies between the different atlimetry and GRACE data sets. These differences can be due to inadequate sampling by radar altimetry, such as in the northern tip of the Antarctic Peninsula where steep regional topography and small outlet glacier size limits the recovery, leakage in the GRACE estimate between different sectors, and uncertainties in the individual measurements and geophysical corrections."

**P22, L34–35** Could add that some of the disagreement with Shepherd et al. (2018) is due to differences in the estimates for the Antarctic Peninsula where retrieving reliable radar altimetry estimates is non-trivial

**P23, L17** "better than a centimeter per year in some regions". This wouldn't be expected in some coastal and mountainous regions as you explain in the following sentences.

**P24, L8** "Over 25% of the ice sheet, largely in the coastal regions of East Antarctica, the time series can be extended back 40 years."

**P24, L8–9** Could reference **?** to indicate that the period of observation for single instruments is short compared to climatic oscillations. Having the extended time series helps separate elevation change due to climate variations, with potential accelerating volume losses.

**P24, L13–14** Remove "e.g. at point A in Fig.9c of 0.55±0.50 m with the change modeled using the FDM (0.48 m between 2008 and 2012)". It makes the sentence hard to read.

**P25, L8–9** Alternatively it means the altimetry measurements are sensitive to variations in low density material, such as changes in accumulation.

**P26, L1–9** This section about radar altimetry in the Peninsula is important as it provides context for the results.

**P26, L22–23** "Part of the discrepancy with the GRACE results could be due to uncertainties in the geophysical corrections applied, such as the effects of glacial isostatic adjustment. More work, similar to the Ice Sheet Mass Balance Inter-comparison Exercises, could help identify the key processes leading to the disagreement."

**P26, L27–29** "We showed that the methods used here improved the overall precision by 50% over the standard datasets available from ESA and NASA."

**P26, L29–31** Mention the comparison with airborne altimetry.

**P27, L1** "Observations from the Seasat and Geosat missions extend the time series in the coastal regions of East Antarctica back to 1978."

**P27, L13** "might be explained by the density mask used or uncertainties in the GRACE processing"

**P27, L16–18** I would use something like "We believe that our multi-mission combination approach can provide an important tool for including and providing context for the ICESat-2 data with observations spanning the past few decades."

**Supplement** What is the reference frame of each of these datasets? What ITRF was used for the final combined product?

**References**

H. A. Fricker and L. Padman. Thirty years of elevation change on Antarctic Peninsula ice shelves from multimission satellite radar altimetry. *Journal of Geophysical Research: Oceans*, 117(C2), Feb. 2012. doi: 10.1029/2011JC007126. C02026.

E. R. Ivins, T. S. James, J. Wahr, E. J. O Schrama, F. W. Landerer, and K. M. Simon. Antarctic contribution to sea level rise observed by GRACE with improved GIA correction. *Journal of Geophysical Research: Solid Earth*, 118(6):3126–3141, June 2013. ISSN 2169-9356. doi: 10.1002/jgrb.50208.

X. Li, E. J. Rignot, J. Mouginot, and B. Scheuchl. Ice flow dynamics and mass loss of Totten Glacier, East Antarctica from 1989 to 2015. *Geophysical Research Letters*, 43(12):6366–6373, 2016. ISSN 1944-8007. doi: 10.1002/2016GL069173. 2016GL069173.

T. Markus, T. Neumann, A. Martino, W. Abdalati, K. Brunt, B. Csatho, S. Farrell, H. Fricker, A. Gardner, D. Harding, M. Jasinski, R. Kwok, L. Magruder, D. Lubin, S. Luthcke, J. Morison, R. Nelson, A. Neuenschwander, S. Palm, S. Popescu, C. Shum, B. E. Schutz, B. Smith, Y. Yang, and J. Zwally. The Ice, Cloud, and land Elevation Satellite-2 (ICESat-2): Science requirements, concept, and implementation. *Remote Sensing of Environment*, 190:260–273, Mar. 2017. ISSN 0034-4257. doi: 10.1016/j.rse.2016.12.029.

E. J. Rignot, J. Mouginot, and B. Scheuchl. Ice Flow of the Antarctic Ice Sheet. *Science*, 333(6048):1427–1430, Jan. 2011a. doi: 10.1126/science.1208336.

E. J. Rignot, I. Velicogna, M. R. van den Broeke, A. J. Monaghan, and J. T. M. Lenaerts. Acceleration of the contribution of the Greenland and Antarctic ice sheets to sea level rise. *Geophysical Research Letters*, 38(5), 2011b. ISSN 1944-8007. doi: 10.1029/2011GL046583. L05503.

L. Schröder, M. Horwath, R. Dietrich, and V. Helm. Four decades of surface elevation change of the Antarctic Ice Sheet. *The Cryosphere Discussions*, 2018:1–25, 2018. doi: 10.5194/tc-2018-49.

A. Shepherd, E. R. Ivins, G. A, V. R. Barletta, M. J. Bentley, S. Bettadpur, K. H. Briggs, D. H. Bromwich, R. Forsberg, N. Galin, M. Horwath, S. Jacobs, I. R. Joughin, M. A. King, J. T. M. Lenaerts, J. Li, S. R. M. Ligtenberg, A. Luckman, S. B. Luthcke, M. McMillan, R. Meister, G. A. Milne, J. Mouginot, A. Muir, J. P. Nicolas, J. D. Paden, A. J. Payne, H. D. Pritchard, E. J. Rignot, H. Rott, L. S. Sørensen, T. A. Scambos, B. Scheuchl, E. Schrama, B. E. Smith, A. V. Sundal, J. H. van Angelen, W. J. van de Berg, M. R. van den Broeke, D. G. Vaughan, I. Velicogna, J. Wahr, P. L. Whitehouse, D. Wingham, D. Yi, D. A. Young, and H. J. Zwally. A Reconciled Estimate of Ice-Sheet Mass Balance. *Science*, 338(6111):1183–1189, Nov. 2012. doi: 10.1126/science.1228102.

A. Shepherd, E. Ivins, E. Rignot, B. Smith, M. van den Broeke, I. Velicogna, P. Whitehouse, K. Briggs, I. Joughin, G. Krinner, S. Nowicki, T. Payne, T. Scambos, N. Schlegel, G. A, C. Agosta, A. Ahlstrøm, G. Babonis, V. Barletta, A. Blazquez, J. Bonin, B. Csatho, R. Cullather, D. Felikson, X. Fettweis, R. Forsberg, H. Gallee, A. Gardner, L. Gilbert, A. Groh, B. Gunter, E. Hanna, C. Harig, V. Helm, A. Horvath, M. Horwath, S. Khan, K. Kjeldsen, H. Konrad, P. Langen, B. Lecavalier, B. Loomis, S. Luthcke, M. McMillan, D. Melini, S. Mernild, Y. Mohajerani, P. Moore, J. Mouginot, G. Moyano, A. Muir, T. Nagler, G. Nield, J. Nilsson, B. Noel, I. Otosaka, M. P. Pattle, W. R. Peltier, P. Nadege, R. Rietbroek, H. Rott, L. Sandberg-Sørensen, I. Sasgen, H. Save, E. Schrama, L. Schröder, K.-W. Seo, S. Simonsen, T. Slater, G. Spada, T. Sutterley, M. Talpe, L. Tarasov, W. J. van de Berg, W. van der Wal, M. van Wessem, B. D. Vishwakarma, D. Wiese, and B. Wouters. Mass balance of the Antarctic Ice Sheet from 1992 to 2017. *Nature*, 558(7709):219–222, 2018. ISSN 1476-4687. doi: 10.1038/s41586-018-0179-y.

I. Velicogna, T. C. Sutterley, and M. R. van den Broeke. Regional acceleration in ice mass loss from Greenland and Antarctica using GRACE time-variable gravity data. *Geophysical Research Letters*, 41(22):8130–8137, 2014. ISSN 1944-8007. doi: 10.1002/2014GL061052.

H. J. Zwally, J. Li, J. W. Robbins, J. L. Saba, D. Yi, and A. C. Brenner. Mass gains of the Antarctic ice sheet exceed losses. *Journal of Glaciology*, 61(230):1019–1036, 2015. ISSN 0022-1430. doi: 10.3189/2015JoG15J071.

---

## Referee Report (RR2)

**Overview**

The changes made to the Schröder et al. (2018) improve the description and presentation of the work. The additions help contextualize the challenges and unresolved issues in resolving both the volume and mass balances of the Antarctic ice sheet. There are a couple of issues that could be resolved before the publication of this manuscript.

**General comments**

- The comparisons with GNSS and NASA Operation IceBridge in the supplementary material are helpful. Could use in the main text to separate the OIB data by region and replace the "zoomed out" figure.

- The paper is still difficult to follow at times. The paper could be reworked for conciseness and to better separate the Results and Discussion sections.

- Significant figures (e.g. $1.8 \pm 0.3$ instead of $1.80 \pm 0.31$)

- Could list some of the mass changes observed in terms of sea level equivalents.

**Line-by-line comments**

**P2, L1** Awkwardly constructed sentence. "As a consequence, results derived from a single mission and mean linear rates reported from a single mission have limited significance in characterizing the long-term evolution of the ice sheet"

**P2, L5** could use "have been" instead of "were"

**P2, L10–11** I'd recommend something like "Paolo et al. (2016) cross-calibrated ERS-1, ERS-2 and Envisat on each grid cell using overlapping epochs. We use a very similar approach for these missions and data from the low-resolution mode of CryoSat-2." Could also reference Adusumilli et al. (2018) here.

**P3, L20** remove "as well"

**P4, L22** "the coarse POCA location"

**P4, L31–33** possibly move "especially with a low threshold of 10%" to the end of the sentence.

**P4, L34** possibly add a comma after amplitude and remove the comma after level.

**P5, L7** use "and GNSS profiles" instead of "or GNSS profiles"

**P5, L28** possibly "which helps confirm the findings of"

**P6, L4** remove "already"

**P7, L29** remove "furthermore"

**P8, L2** remove "like the first group of authors did"

**P8, L8–9** possibly "prefer the simplest viable model in order to keep the number of parameters small as compared to the number of observations"

**P8, L16** change to "(i.e. PLRA, SARin and laser altimetry), the effective surface slope may also differ"

**P9, L13** possibly "to exclude any $res_i$ that exceed five times"

**P12, L17** remove "thus"

**P13, L7–8** Awkwardly constructed sentence

**P13, L10–13** Awkwardly constructed sentence. Also remove "hence"

**P13, L15** PPP processing of the GNSS data?

**P13, L16–19** possibly "The ground-based GNSS profiles were completed between 2001 and 2015 on traverse vehicles of the Russian Antarctic Expedition. Most of the profiles cover more than 1000km"

**P14, L1** possibly "been used due to poorly determined antenna height offsets."

**P14, L7** remove "nevertheless"

**P14, L8** add a comma after terrain

**Figure 7** results appear to be saturated in the Operation IceBridge portion of the plot. Could possibly rasterize the OIB data to be comparable spatially with the merged product. The discrepancies compared with OIB in uncrevassed areas is disconcerting.

**P15, L13** The planes used in the Level-4 ATM product are ∼100m across. While the data is affected by slope errors, the larger sources of uncertainty are roughness and crevassing.

**P15, L14** How often are the vehicle track depths measured when acquiring the GNSS data?

**Section 4.2** Could reference Helsen et al. (2008) in this section.

**P15, L27** Remove "furthermore"

**P16, L1** could add something like "and where large accelerations in ice velocity are observed, such as Pine Island Glacier"

**P18, L8–9** perhaps "For Totten and Denman Glaciers, the 40-year rates at a distance of approximately 100km inland from the grounding line are similar to the rates over the 1992–2017 interval, which indicates a persistent rate of thinning."

**P18, L9–10** possibly "With our merged time series, elevation change rates can be derived for any sub-interval in time for which there is data available, as shown in Figures 9c-j."

**P18, L12–13** Somewhat awkwardly constructed sentence

**P18, L14** for some regions

**P18, L15–29** why were these regions chosen?

**P25, L9** This sentence could be rewritten"Also Seasat and Geosat provide important information here"

**P27, L6–14** Paragraph seems disconnected and difficult to follow.

**P27, L13** remove "nevertheless"

**P27, L19** could use "in the peninsula" versus "here"

**References**

S. Adusumilli, H. A. Fricker, M. R. Siegfried, L. Padman, F. S. Paolo, and S. R. M. Ligtenberg. Variable Basal Melt Rates of Antarctic Peninsula Ice Shelves, 1994–2016. *Geophysical Research Letters*, 45(9): 4086–4095, Mar. 2018. ISSN 0094-8276. doi: 10.1002/2017GL076652.

M. M. Helsen, M. R. van den Broeke, R. S. W. van de Wal, W. J. van de Berg, E. van Meijgaard, C. H. Davis, Y. Li, and I. Goodwin. Elevation Changes in Antarctica Mainly Determined by Accumulation Variability. *Science*, 320(5883):1626–1629, June 2008. doi: 10.1126/science.1153894.

F. S. Paolo, H. A. Fricker, and L. Padman. Constructing improved decadal records of Antarctic ice shelf height change from multiple satellite radar altimeters. *Remote Sensing of Environment*, 177:192–205, May 2016. doi: 10.1016/j.rse.2016.01.026.

L. Schröder, M. Horwath, R. Dietrich, and V. Helm. Four decades of surface elevation change of the Antarctic Ice Sheet. *The Cryosphere Discussions*, 2018:1–25, 2018. doi: 10.5194/tc-2018-49.

---

## Editor Decision (ED1)

[revised manuscript text omitted]

20 sensors.

    We arrive at consistent and seamless time series of gridded surface elevation differences with respect to a reference epoch (09/2010). They represent three-month temporal averages sampled every month and an effective spatial resolution of about $20\,\mathrm{km}$ sampled to a $10\,\mathrm{km}$ grid. We evaluate our results and their estimated uncertainties by a comparison with independent in situ data sets, results from satellite gravimetry and results from regional atmospheric climate modeling. We illustrate that

25 these time series of surface elevation change (SEC) allow to study geometry changes and derived mass changes of the AIS in unprecedented detail. For some examples as Pine Island Glacier, Totten Glacier, Shirase Glacier (Dronning Maud Land) and Lake Vostok, we demonstrate the benefits of the long time series. Finally, we calculate ice sheet mass balances from these data for the respectively covered regions. A comparison with independent data indicates a high consistency of the different data sets but reveals also remaining discrepancies.

30     While this paper gives some examples for new insights obtained from the presented multi-mission altimetry analysis, it can not fully exploit all potential applications. This will be the scope of future work with this data set.

[revised manuscript text omitted]

**3.2 Single-mission time series**

After fitting all parameters according to the multi-mission model (Eq. 2), we regain elevation time series by recombining the parameters $a_0$ and $dh/dh$ with monthly averages of the residuals ($\overline{res}$). For each month $j$ and each mission $M$, the time series is constructed as

$$h_{j,M} = a_{0,M} + dh/dt(t_j - t_0) + \overline{res_{j,M}}. \tag{3}$$

The elevations $h_{j,M}$ all relate to the cells center and are corrected for time-variable penetration, as the parameters of the topography slope and backscatter correction are omitted in this recombination. 
[revised manuscript text omitted]
 datasets shows that these estimates seem reasonable. In the comparison with the GNSS profiles, the relatively low differences, even in regions which imply a higher uncertainty, are likely just incidental for the small sample of $\delta\Delta h$ along the GNSS profiles.

In conclusion, this validation shows that remaining systematic biases (originating from satellite altimetry or the validation data) are on a centimeter level only and that our uncertainty estimate is realistic. However, we have to stress that only altimetric SEC within the interval 2001-2016 can be validated in this way. For the earlier missions, no spatially extensive high precision in situ data are available to us.

**4.2   Firn model**

Another data set which covers almost the identical spatial and temporal range as the altimetric data is the firn thickness data set of the IMAU Firn Densification Model (FDM Ligtenberg et al., 2011), forced at the upper boundary by accumulation and temperature of the Regional Atmospheric Climate Model, version 2.3p2 (van Wessem et al., 2018). Before we can compare this model to our SEC results, however, it is important to mention that the FDM only contains elevation anomalies. A long-term elevation trend over 1979-2016, e.g. due to changes in precipitation on longer time scales (Thomas et al., 2015) would not be included in the model. Furthermore, due to the nature of the model, it cannot give information about ice dynamic thinning/thickening. Hence, to compare the FDM and the SEC from altimetry, we first remove a linear trend. This is performed for the period 1992-2016. The trends are only calculated from epochs where both data sets have data, i.e. in the polar gap this comparison is limited to 2003-2016 or 2010-2016, depending on the first altimetry mission providing data here. After the detrending, the anomalies are used to calculate correlation coefficients for each cell, depicted in Fig. 8a. Figure 8b shows the RMS of these anomalies from the altimetry data, representing the magnitude of the seasonal and interannual variations. Comparing the two maps shows that the correlation is around 0.5 or higher, except in regions where the magnitude of the anomalies is small, i.e. where the signal-to-noise ratio of the altimetric data is low. This relationship is depicted in Fig. 8c, where we see that for the vast majority of cells the correlation is positive. For anomalies with a RMS>0.5 m, the average correlation is between 0.3 and 0.6.

Anomalies against the simultaneously observed long-term trend (1992-2016) can also be computed for earlier epochs. Assuming no significant changes in ice dynamics here, these anomalies allow a comparison of Geosat and Seasat with the FDM. The median difference between the anomalies according to Geosat and the anomalies according to the FDM amounts to 0.12±0.21 m (see Fig. S6). Considering that this difference is very sensitive to extrapolating the respective long-term trends, this is a remarkable agreement. With a median of 0.26±0.32 m, the difference between anomalies from Seasat and from the FDM is larger, but this comparison is also more vulnerable to potential errors due to the extrapolation. As the FDM starts in

[Figure]

**Figure 8. a)** Correlation coefficient between the SEC anomalies of the altimetry grids and the FDM over 1992-2016 after detrending. **b)** RMS of the detrended anomalies of the 1992-2016 altimetry time series. **c)** Correlation coefficient plotted against the RMS. The point density is color coded from yellow to blue. The black dots show the binned mean values.

1979 while Seasat operated in 1978, we compare the Seasat data with the FDM anomalies from the respective months of 1979, which might impose additional differences. Finally, the FDM model has its own inherent errors and uncertainties. Therefore, only part of the differences originates from errors in the altimetry results.

**5 Results**

**5.1 Surface elevation changes**

Some examples for elevation change time series in the resulting multi-mission SEC grids are shown in Fig. 9 (coordinates in Tab. S2). For Pine Island Glacier (PIG, Fig. 9a), we observe a continuous thinning over the whole observational period since 1992 (Seasat and Geosat measurements do not cover this region). Close to the front (point D) the surface elevation decreased by -45.8±7.8 m since 1992, which means an average SEC rate of -1.80±0.31 m/yr. The time series reveals that this thinning was not constant over time, but accelerated near the grounding line (point D and C at a distance of 40 km) around 2006. Also the points at greater distances from the grounding line (B at 80 km, A at 130 km) show an acceleration around 2006. After 2010, the thinning rates at near front decelerate again. For the period 2013-2017, the rate of -1.3±0.8 m/yr is very close to the rate preceding the acceleration. In contrast, further inland the thinning did not decelerate so far and is still at a level of about -1.2 m/yr. Hence, for the most recent period (2013-2017) the elevation at all points along the 130 km of the main flow line is decreasing at very similar rates. A similar acceleration near the grounding line, followed by slowdown, is observed by (Konrad et al., 2016). The onset of this acceleration coincides with the detaching of the ice shelf from a pinning point (Rignot et al., 2014). After that speedup terminated around 2009, the grounding line position was relatively stable (Joughin et al., 2016), which agrees with the elevation changes in our observations.

[revised manuscript text omitted]

5    Comparing the long-term elevation changes over 40 years (Fig. 10a) with those over 25 years shows the limitations of the early observations, but also the additional information they provide. There were only relatively few successful observations at the very margins but e.g. for Totten or Denman Glacier, they show similar rates at a distance of about 100 km from the grounding line. In DML and Enderby Land (basins A-B in Fig. 10a), the 40 yr interval shows less positive rates, compared to 1992-2017. Until 2002, a large part of this region even experienced significant thinning (see time series in Fig. 9c and the

[Figure]

**Figure 11.** Mass change of the Antarctic Ice Sheet north of 81.5°S (**a**) and the three subregions (**b** EAIS, **c** WAIS and **d** APIS) from our combined altimetric time series (blue), GRACE (red) and SMB (orange). The error bars show the uncertainty estimate $\sigma_\Sigma$ of the altimetry data according to Sect. F.2. The gray color in the background displays the fraction of the area covered by altimetry (up to the top means 100%).

maps for the sub-intervals in Fig. 10c-g). After that time, especially over the period 2007-2012 (Fig. 10i), this region shows a huge increase in elevation, which relates mainly to the accumulation events in 2009 and 2011. The sub-intervals in Fig. 10c-j demonstrate the effect of interannual snowfall variability on the elevation change rates over shorter time intervals. They show similar variations also in other regions, pointing out that accumulation events have a strong influence on interannual elevation changes over all parts of Antarctica (Horwath et al., 2012; Mémin et al., 2015).

**5.2 Ice sheet mass time series**

In order to determine the effect of the SEC on global sea level, they are converted to ice mass changes. In a first step, all time series are corrected for glacial isostatic adjustment (GIA) using the IJ05_R2 model (Ivins et al., 2013). This GIA model predicts an uplift of 5 mm/yr near the Antarctic Peninsula and rates between -0.5 and +2 mm/yr in East Antarctica. Furthermore we applied a scaling factor $\alpha = 1.0205$ to account for elastic solid earth rebound effects (Groh et al., 2012). We multiply the resulting ice sheet thickness changes by each cell's area and apply a density according to a firn/ice mask (McMillan et al., 2014,

[Figure]

**Figure 12.** Mass change ($\Delta M\,[Gt]$) of the individual drainage basins north of 81.5°S from our combined altimetric time series (blue), GRACE (red) and SMB (orange). The error bars show the uncertainty estimate $\sigma_\Sigma$ of the altimetry data according to Sect. F.2. The gray color in the background displays the fraction of the area covered by altimetry (up to the top means 100%).

[Figure]

**Figure 13.** Mass change of subregions north of 72°S for several East Antarctic drainage basins from our combined altimetric time series (blue), GRACE (red) and SMB (orange). The error bars show the uncertainty estimate $\sigma_\Sigma$ of the altimetry data according to Sect. F.2. The gray color in the background displays the fraction of the area covered by altimetry (up to the top means 100%).

2016), depicted in Fig. S8, to obtain a mass change. In regions where ice dynamic processes are assumed to be dominating (e.g. in Amundsen Sea Embayment, Kamb Ice Stream or Totten Glacier), we use a density of $917 \, \mathrm{kg/m^3}$. Elsewhere, we apply the density of near-surface firn, obtained from firn modeling using atmospheric forcing (Ligtenberg et al., 2011). We have chosen this straightforward and robust method here, instead of using modeled temporal variations of the firn layer (as e.g. Zwally et al.,

5   2015; Kallenberg et al., 2017) in the volume-to mass conversion. This allows us to compare the time series from altimetry with time series from SMB modeling.

    Cumulated mass anomalies over larger regions such as drainage basins or even the total AIS are obtained by summing up the results accordingly. Therefore, we used the basin definitions by Rignot et al. (2011) (updated for Shepherd et al. (2018), see Figs. 10a and 14b). Cells containing no valid data after the gridding (as e.g. where not enough observations were available,

in the polar gap or where rocks are predominant) are not considered here. Uncertainty estimates were obtained by propagating the respective uncertainties of the SEC, the GIA and the firn density to the basin sums for each month (see Sect. F.2 for details). We also include an estimate for the effect of unobserved cells in the error budget.

Figures 11a-d show time series for the entire AIS north of 81.5°S (i.e. covered by satellite altimetry since 1992), and the
5  respective subregions EAIS, WAIS and the APIS. Similar time series for the single drainage basins over 1992-2017 are shown in Fig. 12. The full four decade time interval for the coastal areas of the EAIS is shown in Fig. 13. These time series use the data north of 72°S only and, hence, provide a nearly consistent observational coverage since 1978. To support the interpretation and evaluate the temporal evolution, we compared the respective time series to GIA-corrected cumulated mass anomalies from satellite gravimetry (GRACE, Groh and Horwath, 2016). To reduce the effect of noise in the GRACE monthly solutions and to
10  make the data more comparable to our altimetry results, we applied a three-month moving average to the GRACE time series. We also compare our data to time series of cumulated surface mass balance anomalies from RACMO2.3p2 (SMB, van Wessem et al., 2018). Similar to the firn model, the SMB contains seasonal and interannual variations due to surface processes. However, it assumes an equilibrium over the modeled period and, hence, does not include long-term changes. The different time series show the good agreement of the techniques in resolving interannual variations. For example for the basin of Totten Glacier
15  (C'-D in Fig. 12), all techniques observe a negative mass anomaly in early 2008, followed by a significant mass gain in 2009. Between 03/2008 and 10/2009, we obtain a mass difference of 116.6±27.0 Gt from altimetry, 109.4 Gt from SMB and 113.4 Gt from GRACE. The high agreement with the SMB indicates that this mass gain is caused by snow accumulation. In most of the basins, we observe similar high agreement in the short-term variations. A good example for the different components of the total mass change signal is the Getz and Abbot region (F-G). While all techniques observe a significant mass loss between
20  2009 and 2011, the SMB does not contain a long term trend, as observed by altimetry and GRACE. In some regions, however, there are also significant discrepancies between the different data sets. The poor sampling of the northernmost APIS (I-I") by altimetry is a good example for the limitations of this technique. In George V Land (D-D'), the agreement during the GRACE period is reasonable, while the mass gain, indicated by the SMB in the early 1990s is not revealed by the altimetry time series.

Over the last 25 years our data indicate a clearly negative mass balance of -2068±377 Gt for the AIS (Fig. 11a). This is mainly
25  a result of the mass loss in the WAIS over the last decade. In contrast, the EAIS has been very stable over our observational record (120±121 Gt between 1992 and 2017). The time series of the APIS contains large uncertainties due to many unobserved cells. Mass change rates for selected regions, obtained from the differences over a specific time interval, and their respective uncertainties are given in Tab. 2. We calculated separate trends for the area north of 72°S, which is covered by all satellites, north of 81.5°S which is covered since ERS-1 and for the total area, which is (except the 500 km diameter polar gap) covered
30  since CryoSat-2. The observed area shows that 96.4% of the cells, classified as ice sheet north of 81.5°S, are successfully covered by observations of ERS-1. Cells without successful observation occur mostly at the APIS, where only 61% is covered with data.

From the overall mass loss of -2068±377 Gt for the AIS (<81.5°S over 1992-2017) we obtain an average long-term rate of -84.7±15.5 Gt/yr. This rate agrees within error bars but is considerably smaller than the results of Shepherd et al. (2018)
35  of -109±56 Gt/yr. However, the extended material in Shepherd et al. (2018) shows that there are still some discrepancies

**Table 2.** Mass change rates for different regions of the Antarctic Ice Sheet and different time intervals. The sizes of the total and observed area refer to all cells classified as ice sheet in the respective region (and, if stated, limited by the given latitude).

| region | area [$10^3$km$^2$] | | dM/dt [Gt/yr] | | | | |
|---|---|---|---|---|---|---|---|
| | total | observed | 1978-2017 | 1992-2017 | 1978-1992 | 1992-2010 | 2010-2017 |
| AIS | 11892 | 11630 | - | - | - | - | -117.5±25.5 |
| EAIS | 9620 | 9413 | - | - | - | - | 1.6±13.1 |
| WAIS | 2038 | 2008 | - | - | - | - | -114.5±19.9 |
| APIS | 232 | 208 | - | - | - | - | -4.5±8.7 |
| AIS (<81.5°S) | 9391 | 9053 | - | -84.7±15.5 | - | -58.6±20.3 | -137.0±24.9 |
| EAIS (<81.5°S) | 7764 | 7555 | - | 4.9±5.0 | - | 8.0±6.2 | 2.4±12.4 |
| WAIS (<81.5°S) | 1394 | 1358 | - | -91.7±10.3 | - | -69.4±13.1 | -134.9±19.6 |
| APIS (<81.5°S) | 232 | 142 | - | 2.1±8.9 | - | 2.8±12.3 | -4.5±8.7 |
| EAIS (<72°S) | 2779 | 2274 | 1.5±5.8 | -3.4±4.0 | 12.1±17.4 | 0.0±4.9 | -8.4±10.1 |

For the APIS (<72°S), the very sparse observations of Seasat and Geosat did not allow calculate a reliable trend.

between the different techniques to determine the AIS mass balance. For the time interval 2003-2010 (Extended Data Table 4 in Shepherd et al., 2018) the Input-Output method obtains a rate of -201±82 Gt/yr for the AIS, while the mass balance rates, aggregated from satellite gravimetry (-76±20 Gt/yr) and from altimetry (-43±21 Gt/yr) agree much better with our result for the AIS (<81.5°S) between 2003 and 2010 of -64.7±24.9 Gt/yr.

**6   Discussion**

**6.1   Multi-mission SEC time series**

The single-mission time series, obtained in Sect. 3.2, contain satellite-specific calibration biases as well as offsets due to the specific sampling characteristics of different sensor types. In order to form a consistent SEC time series, these biases needed to be determined and corrected. A comparison with in situ data showed that there are no significant offsets between elevation changes from our multi-mission altimetry data and the validation datasets. This comparison, however, could only validate our data in the interval 2001-2016. A quality control for the whole time span was performed by a comparison with a firn model. The correlation of the detrended data sets shows that especially for regions where the interannual variation is large (compared to the measurement noise of the altimeters) both time series agree very well. This comparison even provided independent estimates for the error of the early missions. The average differences between the detrended time series of the FDM and the SEC show that the observations of Geosat and even of Seasat agree with the model results within a few decimeters. For SECs of up to several meters w.r.t 2010 (see Fig. 6), this means that also the older data can be used to calculate elevation change rates with an accuracy better than a centimeter per year (see Fig. S7a). Unfortunately, in coastal DML west of the ice divide A', the data of

[Figure]

**Figure 14.** Mean rates for the time interval 2002-2016 of elevation changes from IMAU-FDM **(a)**, from the multi-mission SEC grids **(b)** and of the mass changes from GRACE **(c)**.

Seasat and Geosat are very noisy. This due to the mountain ranges just north of 72°S, which lead to many losses of lock of the measurements all the way across this part of the ice sheet. The same applies to the measurements at the APIS.

**6.2 Surface elevation changes**

The mean rates of elevation change in Fig. 10 show the regions which experience a significant thinning (Amundsen Sea Embayment, Totten Glacier) or thickening (Kamb Ice Stream) which was already reported by previous publications (e.g. Wingham et al., 2006b; Flament and Rémy, 2012; Helm et al., 2014; Zwally et al., 2015). By combining all the single missions consistently we analyze long-term changes over the full time period covered. For 79% of the area of the AIS, this means a time span of 25 years. For 25%, mainly the coastal regions of East Antarctica, even 40 years are covered. We assume that these long-term trends are significantly less affected by short-term variations in snowfall than a trend from a single mission.

[revised manuscript text omitted]

---

## Author Response (AR2)

**Author's Response**
for the second review of
"**Four decades of Antarctic surface elevation changes from multi-mission satellite altimetry**"

We are grateful for the constructive and helpful comments of both referees and the editor himself. We think that the very specific hints helped to significantly clarify the manuscript in the description of the methods, the results and in the interpretation as well. We have rearranged some sections, also between the supplement and the main part, and think that now, the text reads more fluidly. The description of several additional data sets, as the validation data sets, the FDM, the firn density grid, the SMB and GRACE has been clarified.

On the following pages, we respond to all referee reports in detail, followed by a version of the manuscript containing all changes made since the reviewed version.

**Authors response to referee report #1**

*General comments*

• *Combining laser altimetry estimates with radar altimetry estimates is a non-trivial task (e.g. Fricker and Padman, 2012). Not simply due to the differences in footprint diameter but by potentially measuring different surfaces. The comparisons with NASA Operation IceBridge in West Antarctica and the Peninsula are helpful for this but it was difficult to discern the local discrepancies between the measurements as presented.*

Besides the different topography sampling, we agree that also the firn penetration is a challenge in this combination of different techniques. This penetration might even vary in time as the surface properties of the upper firn layers change, e.g. due to the depositioning of fresh snow. Such temporal variations in the penetration depth of the radar signal, however, are significantly reduced by our low threshold retracking as it reduces the overall influence of snowpack penetration. The remaining effect of changes in firn properties is accounted for by our backscatter correction. Hence, we consider the remaining penetration bias of radar altimetry w.r.t. the surface as invariant in time. This constant bias between radar and laser is removed as we reduce each technique's time series by their respective elevation at the reference epoch 09/2010. This reduces any constant offset between the time series, may it be due to the topography sampling, the firn penetration or remaining calibration issues.

We added these points to Sect. 3.2 and 3.3.2 to make the reader more aware of the fact that besides the spatial sampling also other factors may be responsible for these biases. Nevertheless, any constant biases between the time series, may they originate from instrumental calibration, sampling issues or firn penetration, are removed in this step. Concerning the validation with ICEBridge, we added some more detailed plots to the supplement as indicated in the line-by-line comment to highlight the level of agreements as well as the noise on smaller scales.

• *The paper can be still be difficult to follow at times. The manuscript could be reorganized and restructured to not repeat information between sections.*

We changed the order of Figs. 9 and 10 and the related discussions. This also helped to eliminate some repetitions there. Furthermore, we removed the former Sect. 6.1 and integrated the most important content elsewhere. We hope that these and the other changes helped to make it easier to follow now.

• *The data and methods sections could also still be more detailed. Are the GRACE basin time series results simply an available product? If so, it could include a citation for the data repository. If not, should detail the GRACE processing scheme as you start to diagnose specific uncertainties in time-variable gravity measurements.*

We used the GRACE product of the Antarctic Ice Sheet CCI project. This information and links to the data repositories has been added.

*Line-by-line comments*

*Abstract This is much improved from the previous versions.*
Good to hear.

*P1, L14–15 I would cite Shepherd et al. (2012) as a reference the time-variability of Antarctic surface elevation change.*
Done.

*P2, L9 Remove "Also" from the start of this sentence.*
Done.

*P2, L18 "Finally, we merge all time series from both radar and laser altimetry"*
Changed.

*P2, L22 What do you mean by "three-month temporal averages sampled every month"? A moving average of the elevation data with a three month sliding window?*
Exactly. Changed the sentence to make this more clear.

*P2, L24 "independent in-situ and airborne datasets, satellite gravimetry estimates, and regional climate model outputs".*

Done.

In contrast to dh/dt, which describes the rate of surface elevation change, the surface elevation change "SEC" explains our result very well: the elevation change w.r.t a reference epoch. We cannot see any reason why this acronym shouldn't be used.

Done.

Done.

Modified accordingly.

The outlier screening is an important step, but we think that if we would present more details concerning ICESat here, the reader would wonder why no details concerning the other missions are presented here. Instead, in order to keep the main text short we refer to the supplement, where all these technical details for each mission can be found.

We did not employ any regional selection. If the data screening and flags indicates a measurement as "valid", it is used. However, there is also an iterative outlier screening in the repeat altimetry processing. The spatial distribution of the data can already be seen in Fig. S3.

The text has been modified accordingly. Concerning the temporal changes in the DEM, we would like to mention that for the POCA, only the relative topography within the BLF is relevant. For regions, that are observable by radar altimetry (i.e. flat enough) we do not expect a significant effect of changes in the relative topography within the BLF on the location of the POCA.

We think it is necessary as the original method of Roemer et al. (2007) is more precise, while we use a simplification. For each shot, the DEM-to-satellite distances are calculated for every cell within the beam limited footprint (BLF). After that, Roemer et al. (2007) used a 2x2km moving window to find the closest average pulse limited footprint location. Instead, we do not use this moving window but smooth the DEM instead. This saves a lot of computational time, compared to calculating all the moving averages for each shot in each BLF again. We modified the text to make this difference more clear.

Surely, cases might exist where two local maxima might have exactly the same distance to the sensor. However, in such a case, the other techniques would not correctly handle this situation as well. Instead, they would use some "mean" slope, which is just a trade off and would in fact assign the measurement to somewhere completely different. Instead, this approach would refer the measurement to one of the two maxima, which is definitely the preferred method.

*P5, L1–8 This is a good review of the effects of spatial aliasing. However, the penetration of the radar signal into the surface is another bias term that would potentially affect rate estimates. Could add a caveat.*

For this reason, Lake Vostok is a great spot to study penetration effects in the absence of topographic features (see Schröder et al., 2017). The text has been modified accordingly .

*P7, L9–11 At least for laser altimetry, 1km may be too coarse of resolution for the assumption of planar surfaces in coastal regions (Markus et al., 2017). Could include additional higher order surface shape terms.*
This might be true for several cases. This point has already been discussed later in Sect. 6.3 with regard to the APIS. However, we prefer to fit as few parameters as possible for the sake of robustness. As discussed by Ewert et al. (2012, Fig. 2d), an unfortunate distribution of tracks within the repeat track box might lead to large ambiguities between elevation differences due to topography and due to real elevation changes. This becomes even more critical as more parameters are fitted.

*P9, L1–2 Rewrite to something similar to "Over regions of flat topography, such as the interior of East Antarctica, the weights between PLRA and ICESat are comparable".*
Done.

*P10, L6 The use of FDM is an interesting approach. I would add the caveat that changes in ice dynamics would not be captured by this metric.*
Done.

*P11, L5–10 Is applying a uniform single bias just forcing the linearity condition to regions where it might not be viable? The distributions of biases in the supplementary material appears to be not random but fairly spatially correlated.*
The linearity condition only selects the regions which are proven to be very likely linear. From these regions, we calculate the average bias. In any other region, where we assume a non-linear signal, the bias does not force the data sets to become linear. It just corrects for the most likely bias, observed over other regions which are close to linearity.

We are aware of the fact that the spatial pattern of this bias is a big issue in the calibration of Seasat and Geosat. Therefore, we use the significantly larger standard deviation of 85cm for Seasat and 61cm for Geosat in our error propagation everywhere, instead of the much lower values for the other missions in Fig.S4. However, as we do not fully understand the origins of these patterns, we do not want to extrapolate the spatial pattern to the coastal data.

*P11, L24 These instruments could be detecting different surfaces beyond the topographic sampling discrepancy.*
Text edited, see first bullet of the general comments.

*P12, L1–2 Could possibly use an interpolation scheme with tension or inherent smoothing.*
We decided against an interpolation scheme with tension in order to avoid extrapolating values to regions where no real measurements exist. Instead, we limited our smoothing to regions within a footprint, hence to the measured locations only. This is already discussed in Sect. 3.3.3 and in Sect. 6.3 with regard to the APIS.

*Figure 7 Difficult to discerns the differences from GNSS and Operation IceBridge with these maps. Lots of dots area overlapping. Is the color scale in a) and d) the same as in b) and e)?*
We added some more detailed maps of key regions in the supplement (Fig. S6). A comment concerning the color scale has been added as well.

*P14, L2–3 Could add possible explanations of the difference between the airborne altimetry measurements and the combined estimate.*
Done.

*P14, L4–8 Could expand to note the potential error sources between GNSS and altimetry. This seems like a fairly large amount of variability for East Antarctica. Could potentially use other datasets for validation as well.*
Explanations for potential error sources have been added. We compared our results to elevation changes from OIB, kinematic GNSS and (as a correlation coefficient) to the FDM. We conclude the section with a statement about the issue that we could not validate the earlier data due to the lack of appropriate validation data sets.

*P14, L10 Remove "we have to stress that"*
Done.

*P14, L12 I would change it to "publicly available" versus "available to us".*
Done.

*P14, L15–17 Remove "Before we can compare this model to our SEC results, however, it is important to mention that"*
Done.

*Figure 8 The RMS with the FDM seems surprisingly high.*
This RMS did not mean an error, it is the squared mean of the anomalies. Figure 8b should give an impression where we can expect a (non-linear) signal above the noise level of the altimetry, hence, where we can expect a correlation at all. We changed the label to make this more clear and modified the text as well.

*P16, L3 In terms of spatial resolution?*
"temporal" added.

*P17, L15 "rates within uncertainties and very close to zero"*
Changed.

*L17, L17–22 The discrepancy with Zwally et al. (2015) is an important finding.*
We think so too, however, this has already been discussed several times, e.g. by Scambos and Shuman (2016), Richter et al. (2016) or at different points in Schröder et al. (2017).

*Figure 10 Would the early mission data over the Antarctic Peninsula be at all viable? The Peninsula is particularly difficult to measure with radar altimetry (Shepherd et al., 2012).*
In Sect. 6.1, we discuss that the APIS is one of the regions where the early data is very noisy due to difficult topography. This is the reason why we do not calculate a basin mass trend here (see footnote for Tab. 2).

*P17, L31–P18, L4 This paragraph repeats a lot of information provided previously about individual glaciers. Could be merged.*
The presentation of the results at Fig. 9 and Fig. 10 have been switched and in this context, also some repetitions have been eliminated. We think that this makes the whole results section now easier to follow as we now first give a general overview before we discuss the details.

*P17, L 34 Remove "too"*
Done.

*P19, L7 "The surface elevation time series is converted into ice mass changes in order to determine their effect on global sea level."*
Edited.

*P19, L7–8 "The elevation time series are corrected for uplift rates related to glacial isostatic adjustment (GIA) using coefficients from the IJ05_R2 model (Ivins et al., 2013)."*
Edited.

*P19, L9–10 The elastic effects would likely be much more localized than the outputs from a GIA model, and would be related to the modern-day change rather than the unloading since the LGM.*
We agree that the formulation was misleading here. The scaling factor is applied to the SEC time series, not to the GIA model. We edited this sentence to make this more clear.

*Figure 11,12 As the area of observation changes with time, would it be more appropriate to calculate the average surface mass density (kg/m 2 ) change rather than the total mass change of these sectors (i.e. divide the resultant mass by the total area)?*
As the elevation changes are not uniformly distributed in a basin, we think that such an average surface mass density would have the same issues. As the largest changes typically occur in difficult terrain for radar altimetry, an observational gap in an area with large changes would also significantly affect an average mass density. However, for most of the basins, there are no significant changes in the observed area.

*P21, L7–8 "We integrate our measurements over larger regions to calculate the cumulative mass anomalies for individual drainage basins and major Antarctic sectors (AIS, WAIS, EAIS, APIS). Our basin delineations are from Rignot et al. (2011a), which have been updated for the second Ice Sheet Mass Balance Inter-comparison Exercise (IMBIE-2, Shepherd et al., 2018)."*
Changed.

*Table 2 Is the area of observation constant for these volume rates?*
The rates were obtained from epoch differences of the time series and the observed area in Tab. 2 refers to the minimum area of any of those epochs. For the <81.5°S sector, they refer to the time series displayed in Fig. 11. We agree that it causes an issue if the area is not the same at both epochs, however, as it can be seem from Fig.11, significant differences in the observed area occur only at the APIS. To account for the errors which might be imposed by unobserved areas, we apply an estimate for the magnitude of the effect to the uncertainty as explained in the supplement at F.2.

*P22, L7–9 Same GIA model outputs used to correct the GRACE data?*
Yes. More details concerning the GRACE data have been added to the manuscript as described at bullet 3 of the general comments.

*P11, L4 What is the reference period used to calculate SMB anomalies? 1979–2017?*
We are not sure to which exact location this comment refers. However, we added the modeled period (1979-2016, which is the same as for the FDM) to the text in Sect. 5.2 once more.

*P22, L17 Variations and attribution of Cp-D mass change shown here are supported by Velicogna et al. (2014) and Li et al. (2016).*
Good hint, we added the references.

*P22, L18–19 "There is less agreement in regions where surface mass balance change may not be dominant, such as of the Getz and Abbot regions (F-G) in West Antarctica."*
We modified the passage but we would like to stress the different components of the signal more. In the short term variations, all data sets agree, while only the long term changes are not contained in the SMB model, hence we attribute them to ice dynamics.

*P22, L20–21 I'd suggest something like "In some regions, there are also significant discrepancies between the different atlimetry and GRACE data sets. These differences can be due to inadequate sampling by radar altimetry, such as in the northern tip of the Antarctic Peninsula where steep regional topography and small outlet glacier size limits the recovery, leakage in the GRACE estimate between different sectors, and uncertainties in the individual measurements and geophysical corrections."*
The passage has been modified accordingly.

*P22, L34–35 Could add that some of the disagreement with Shepherd et al. (2018) is due to differences in the estimates for the Antarctic Peninsula where retrieving reliable radar altimetry estimates is non-trivial*
Done.

*P23, L17 "better than a centimeter per year in some regions". This wouldn't be expected in some coastal and mountainous regions as you explain in the following sentences.*
This phrase has been modified.

*P24, L8 "Over 25% of the ice sheet, largely in the coastal regions of East Antarctica, the time series can be extended back 40 years."*
Changed.

*P24, L8–9 Could reference ? to indicate that the period of observation for single instruments is short compared to climatic oscillations. Having the extended time series helps separate elevation change due to climate variations, with potential accelerating volume losses.*
Very good point. We added this argument.

*P24, L13–14 Remove "e.g. at point A in Fig.9c of 0.55 ± 0.50 m with the change modeled using the FDM (0.48m between 2008 and 2012)". It makes the sentence hard to read.*
Put this to a separate sentence to make it easier to read.

*P25, L8–9 Alternatively it means the altimetry measurements are sensitive to variations in low density material, such as changes in accumulation.*

They are sensitive to variations in low density materials as in high density materials as well. The point here is that we want to explain, why the ice dynamic changes (as in the Amundsen Sea basins) show up in darker red in the GRACE mass time series in comparison to altimetry, while the mass gains in DML are less pronounced.

*P26, L1–9 This section about radar altimetry in the Peninsula is important as it provides context for the results.*

Great, thanks.

*P26, L22–23 "Part of the discrepancy with the GRACE results could be due to uncertainties in the geophysical corrections applied, such as the effects of glacial isostatic adjustment. More work, similar to the Ice Sheet Mass Balance Inter-comparison Exercises, could help identify the key processes leading to the disagreement."*

Changed accordingly.

*P26, L27–29 "We showed that the methods used here improved the overall precision by 50% over the standard datasets available from ESA and NASA."*

Done.

*P26, L29–31 Mention the comparison with airborne altimetry.*

Done.

*P27, L1 "Observations from the Seasat and Geosat missions extend the time series in the coastal regions of East Antarctica back to 1978."*

Done.

*P27, L13 "might be explained by the density mask used or uncertainties in the GRACE processing"*

Done.

*P27, L16–18 I would use something like "We believe that our multi-mission combination approach can provide an important tool for including and providing context for the ICESat-2 data with observations spanning the past few decades."*

Modified accordingly.

*Supplement What is the reference frame of each of these datasets? What ITRF was used for the final combined product?*

All missions referred to WGS84, except ICESat, which originally refers to a T/P ellipsoid. However, also ICESat data come with corrections for WGS84, which were applied as mentioned in A.4. We stated more clearly now that all other missions already refer to WGS84. The final product does not contain absolute elevations, only elevation changes, hence, it does not refer to any ellipsoid.

**Authors response to referee report #2**

*The manuscript has improved significantly due to the changes addressing the issues raised by the reviewers and posted comments.*

*The main goal of the manuscript is to introduce a new method for deriving multi-mission elevation changes and mass change time series of altimetry data. The new results are used for investigating climate and ice dynamics related changes of the Antarctic Ice Sheets.*
*Part of the methodology is described in the supplementary information and some of the arguments of selecting certain models, eg., planar surface fit vs. higher order polynomial, are also explained in the supplement only. The supplement is well written and it is easy to follow. However, the main manuscript is fragmented and unclear in some places, making it difficult to follow the discussion as it moves from the main paper to the supplement and back. The manuscript should explain all main points of the study indicating where the reader should consult the supplement and references for details.*

The intention of this separation was to keep the (already quite long) document short by providing only the main points in the manuscript while providing more details (as the calculation of the uncertainties or several maps for each mission) for the interested reader in the supplement. However, we agree that the choice of repeat altimetry parameters is too important to appear in the supplement only. Therefor, the former Section C.1 has now been completely integrated into the main manuscript.

In section 3.3, we changed the reference to the supplement instead. We believe that it will be very hard for a reader to follow the main idea if we discuss all the details of each merging step as the processing parameters, outlier checks and the specific arguments for these decisions in the main part.

*Also, the grammar needs some improvement. For example, the words respective or respectively are used a total of more than 60 times (paper and supplement), in my view, often unnecessarily.*

We agree and eliminated or replaced many occurrences of this word. Many other changes have been made with regard to all detailed comments.

*Following recommendations from the review of the original manuscript, the authors included a conversion of the elevation changes to mass changes. The applied method uses an ice density mask defined by a priori knowledge of the spatial distribution of changes due to ice dynamics and surface processes for the conversion. To my best knowledge, this technique has not been used before for long periods with significant changes in ice dynamics. There is ample evidence of changing Antarctic ice dynamics in the literature and in the manuscript itself. The authors describe the technique as "straightforward and robust", but they don't provide any assessment of its performance. Considering the changes in ice dynamics and the complex variations of firn-compaction and density during the 40 years spanned by the altimetry record, I expect large errors. These errors, only partially accounted in the error budget, could be responsible for some of the significant discrepancies between the mass balance derived from GRACE and altimetry in the study.*

We agree that ice dynamic processes might also be involved in other regions as indicated by the density mask. This is discussed as a likely cause for the mentioned discrepancy between altimetry and GRACE in Sect. 6.3. At least for some of the regions, the preferred density should be chosen according to the combination of ice dynamics and surface processes. Zwally et al. (2015) discuss, that such a pseudo-density could even be out of the range between firn and ice density if both processes have opposite signs. This is clearly a caveat of this approach. However, without additional information, the selection of a more appropriate density is not possible. Such a detailed analysis is far beyond the scope of this paper. Instead, the main intention of the mass time series here was to make the altimetry data comparable to alternative data sets of mass changes as SMB or GRACE. This comparison can, in fact, be interpreted as the assessment recommended. If the FDM/SMB data sets were already used in the volume to mass conversion, such a comparison would not be possible.

*Also, the manuscript and the references do not describe the method with sufficient details. Was the density an average density for a given time period? What was this time period and how did the authors account for the different length of the altimetry record?*

This surface density grid is described in Ligtenberg et al. (2011) (see also Fig.7A therein). It is also publicly available via the Quantarctica (http://quantarctica.npolar.no/) data collection. The surface density (taken to be the density of the upper meter) is assumed to be constant over longer times, which is almost always done. In the IMAU-FDM, the surface density is parameterised using the long-term climate: temperature, accumulation, and wind speed.
The passage has been edited to add some details for clarification.

*Was the firn-compaction removed?*
We did not remove a firn compaction as this effect affects mainly short time scales. Over long time scales, which is the main scope of this paper, only a persistent instability in atmospheric conditions would cause an effect of firn compaction. Such an instability, however, wouldn't be detectable even by firn modelling due to the steady-state condition (see next answer).
We added this discussion to Sect. F.2.

*Also, the manuscript several times states that the FDM and SMB time series do not contain long-term changes. This is only partially true. Both FDM and SMB assumes a balance period. The inadequate knowledge or lack of such a balance period could result in an error in the FDM and SMB anomaly trends, but the remaining components of the long-term change will be correctly reconstructed. The cumulative SMB anomalies presented in the manuscript show several examples of these long-term trends (e.g., Fig 13).*
Here, we have to contradict respectfully. For the IMAU-FDM we use, the steady-state assumption is a precondition as described in Ligtenberg et al. (2011). It presumes no overall trend over the modeled period (1979-2016 in this case).
The SMB from RACMO2 does not presume a steady-state. It does not assume a balance period either. RACMO2, the regional climate model, is forced at the lateral boundaries and the atmosphere within the domain is allowed to evolve freely. The simulated weather (and subsequent SMB) is therefore not influenced by any balance assumption. However, in order to get an ice sheet mass balance, this SMB needs to be combined with ice discharge (see e.g. Rignot et al., 2011). As we do not have accurate ice discharge data, which would introduce additional sources of error, we focus on the anomalies in SMB during our comparison. As we reduced the mean rate of each cell, each of the cumulated SMB anomaly curves start and end at the same value. These values may be different from zero as the curve has been adjusted to zero at 09/2010, however, there is no trend.

Nevertheless, we agree that we maybe did not explicitly enough explain that we use SMB anomalies. Therefore, we modified the introduction of this data set to be more precise and introduced the abbreviation SMBA.

***Detailed comments:***
*Page 7. Figure 3. This figure is only referred after the reference to Figure 4. Either the figures order should be changed, or a reference to this figure should be included.*
We agree and moved the Figure to Sect. 3.3.

*Page 9, line 8: I assume that all other corrections were performed and not "omitted"*
The parameters of topography slope and backscatter regression are omitted in the recombination, which has the effect that the resulting time series can be considered "corrected for these effects".
We changed the formulation making it more clear hopefully.

*Page 9, lines 8-14: I can only guess the meaning of "These "res" [residuals] represent the anomalies of typically a single satellite pass towards all respective parameters including the linear rate of elevation change." It needs to be rephrased.*
We agree and added the information about typical repeat cycle periods, which needs to be known to understand this statement.

*Page 11, lines 27-28. As it has been documented, significant changes occurred in ice dynamic during the 8 years used for the correction (e.g., Flament and Remy, 2012). The density of sampling would probably allow using a quadratic approximation to allow for non-linear dynamics. What was the reason for using the linear approximation and what could be the impact of the modeling error?*
This is true for cells where e.g. Envisat and CryoSat observations cover the whole time period. However, we have to consider that there may be also cells where only ICESat observations are available, which sample only some of the months during the laser campaings until 10/2009. Here, an extrapolation including a quadratic term might lead to large errors.

*3.3.3 Merging different techniques: A region with a 30 km diameter is large enough to exhibit variations in thickness change rates. Therefore, as the authors have stated, the residuals of the elevation changes within the region could reflect not only measurement errors but real variations. In my opinion, using a formal error propagation through the entire error assessment, i.e., instead of including equation (4) in Supplement C.4 could provide a better error estimate.*

We agree about the variations within this radius but please consider also, that we used a Gaussian weighting, which has the effect, that a value at 30km distance would have only 0.12% of the weight of a comparable value at the center.

We agree also that equation (4) in C.4 is just a simplification of a full formal error propagation of this averaging, as covariances are not included in the first term. Instead, we added the second term to account for such effects. For a full error propagation, a full variance-covariance matrix would be necessary, including the spatial correlation of the inter-mission offsets. Such correlations would need to be calculated when different missions are combined. However, this combination is a stepwise approach which means that the uncertainties of the combination of the overlapping and non-overlapping missions would first need to be propagated to the combined PLRA time series. After this, another error propagation with a full variance-covariance matrix would be necessary for the combination of the techniques. As in fact, for Seasat and Geosat all measurements are correlated due to the average offset correction, a full variance-covariance matrix would have the dimension of $n \times n$ where $n$ is the amount of all 1km grid cells (ca. 23.000.000). Furthermore, due to the temporal averaging of the PLRA offsets and the time period in the fit of the technique offsets, these dimensions have to be multiplied by the number of months included. Such a matrix is needed for EACH cell for a formally correct error propagation. Handling such a system of matrices is not possible with the computational resources we have. Besides these points, also such a full model of error propagation would still be incomplete as further correlations, e.g. between consecutive cells which are observed by the same satellite orbits would not be accounted for here. To overcome all these problems, we add the second term in Eq. (4), which we consider as an conservative simplified estimate for these effects.

*Page 12, line 9: What is the meaning of the "flow line of outlet glaciers"? Regions of fast-flowing outlet glaciers? Measurements along the flowlines of fast-flowing outlet glaciers?*

Changed to fast-flowing outlet glaciers.

*Page 12: showing annual rates rather than differences relative to the reference time would make it easier to interpret the changes. Otherwise, the sign of the relative anomalies is different for dates preceding the reference dates and those after the reference data. Also, using differences relative to the reference time results in a scaling of the variation of elevation differences within the averaging area, as mentioned in the manuscript, and makes the interpretation of the error difficult.*

We take this comment serious as it has been raised more than once. Nevertheless, we think it is important to show our resulting grids as they are (elevation differences w.r.t a reference epoch and their uncertainties) because this is important to understand the characteristics of the products derived from these grids. This accounts e.g. to the understanding of the standard deviations related to these absolute SECs during the smoothing, the spatial distribution of linear rates over different intervals, or errors related to these rates. Elevation change rates as suggested here can already be found later in the results section (Fig. 9 now)

*Page 12, line 10: I suggest to refer to the area of averaging, rather than to the area of smoothing.*

Changed to "weighted averaging" as we agree that this is more precise but want to make the reader also aware of the fact, that we used weights. Hence, as discussed before, differences on the very edge of the area of averaging have only very small weights and hence also low influences on the results.

*4.1 In situ observations. Briefly describe the measurement type and processing method for both types of observations (e.g., ground GNSS traverse using snowmobile, or airborne laser altimetry). Include the error of the in situ measurements based on literature. Elaborate on the reason of the difference between in situ measurements and new results in the flat interior region.*

Changes have been made accordingly.

*4.2 Firn model. Include details of the particular FDM use. Did it assume the same steady state period as in Ligtenberg et al., 2011? What was the temporal and spatial resolution? How were the*

*modeled FDM changes interpolated to the altimetry grid? Finally, what is the error of the FDM model?*

Details concerning the FDM resolution have been added. We agree that also the interpolation to the altimetry grid is one of the steps which has to be done. As our altimetry (10x10km) has a significantly higher resolution as the FDM (27x27km), errors due to this reprojection can be neglected. This is simply a technical detail and has no influence on the results.

Sensitivity simulations of the FDM show linear relations between a %-change in accumulation and snowmelt (the governing climate processes of firn layer variations), which can be used to quantify the uncertainty of the FDM. However, alternatively, a validation with observations can also provide information on the influence of potentially unmodeled effects. Hence, a cross-validation as performed here provides independent uncertainty estimates.

*It is difficult to understand the comparison method applied between the FDM and the altimetry records. A figure, illustrating the approach would be very useful.*

We added such a figure to the supplement.

*Detailed question: Were both time series detrended independently using linear approximations? In page 14, line 24: "RMS of these anomalies from the altimetry data", while in the figure caption for Fig. 8. b): "RMS of the detrended anomalies of the 1992-2016 altimetry time series". Do you refer to the same RMS in both places? Is it the RMS of the difference between the detrended FDM and detrended altimetry time series computed for each grid cell? Why is the RMS reflect "the magnitude of seasonal and interannual variations"? I assume that both the detrended FDM and the detrended monthly altimetry data reflect seasonal and interannual variations, so the RMS of their difference would be related to the errors and sampling of the two data sets and any short-term variations in ice dynamics.*

Each of the time series was detrended using a trend fit to the respective time series. We replaced the term RMS as it seems to cause confusion here. RMS is simply the description of the method used to calculate the average amplitude of the non-linear signal. In Fig. 8b, the RMS means the root of the mean squared detrended SEC from altimetry. We clarified this in the captions and the text. We agree, that this represents not only interannual but also seasonal changes and added this to the text as well.

*Figure 8. Capital letters and drainage basins in a) are not explained and used in this section. The point density needs a color bar.*

Figure modified accordingly.

*Results, 5.1 Surface elevation changes*
*Table S2. Include the projection system used for the X,Y coordinates and latitude, longitude locations. Expressing the multiyear changes as annual change rates would make it easier to interpret the results, e.g., relative elevation at a later date minus relative elevation at an earlier data divided by the number of years would always give negative values for thinning rates and the normalization would allow a direct comparison of the change rates during different time periods.*

Coordinate system information added. We think, for an interpretation, the reader should refer to the plot (Fig. 10 now). These numbers in the supplement are provided as a reference for the rates in the text. Presenting absolute values of SEC here enables the reader to calculate a rates over any given interval.

*Page 15, line 19: elaborate on the connection between the stable grounding line position and observed elevation changes.*

Done.

*Figure 9. Since elevation changes are shown as relative values, a "floating" elevation change scale would work better than the current vertical axis. What is the difference between the black and colored line in the elevation change time series?*

We are not sure what is meant by a "floating" elevation change scale. We added the explanation that the black bars represents the uncertainty.

*Page 16, line 3: how much longer are the new time series? Clearly articulate on the new findings that are allowed by the extended time period.*

The additional time interval covered depends on the data used in the respective studies. While

Pritchard et al. (2009) or Flament and Rémy (2012) use only one mission, Zwally et al. (2015) processed results for different missions. However, the advantage of our time series is the long-term consistency and the temporal resolution, as mentioned in the text. The following sentences clearly describe the new findings. On the one hand, these are elevation change rates from 40 years instead of a decade. On the other hand, it is the monthly resolution, which sets the rates reported by other authors and the variations as observed previously e.g. by Li et al. (2016) in a 40 year context. We modified the text to express this point even more.

*Page 17, lines 8-13: it is somewhat unclear if this paragraph discusses one event (resulted from two accumulation anomalies) or two distinct elevation increases (in line 10: these event – should be either this event or these events). Also, some of the time series show a continuous increase, while others show two periods of rapid surface elevation increase. This is a nice result, and an enlargement of the figure with a more detailed description would be helpful.*

We edited several parts of this paragraph and think it is much clearer now. We agree that for the points further inland, both events are hard to separate. The spatio-temporal pattern of these events in the altimetry and the FDM could help to understand the reasons, but this would require a detailed analysis. We think that such an analysis should be done very carefully and, hence, this should be done in a separate study. We hope that we can contribute to a deeper understanding with our data set.

*Page 17, line 23-30: I don't understand the computation as described here.*

We edited the description to make it easier to understand.

*Figure 10. Show the locations of the enlargements in Figure 9 to connect the two parts of the section.*

Done.

*Page 18, line 9: this part should be better connected to the changes presented on the Shirase Glacier earlier in this section.*

This was accounted for during the reorganisation of the content of this section.

*Page 19, line 2: how much was the "huge" increase? Add a number!*

Obsolete in the revised version. Numbers are included in the discussion of the time series examples here (Fig. 10c now).

*Page 19, line 4: be more specific about the other regions.*

Obsolete in the revised version.

*5.2 Ice sheet mass time series: See general remarks at the beginning of this review.*

And so refer to the answer given there as well.

*6.1 Multi-mission SEC time series*
*Page 23, line 9: use bias instead of offset to describe the difference between change derived from altimetry data and validation data*

Done.

*Page 23, lines 11-13: the >0.5 correlation values do not mean that "both time series agree very well". It just means that the seasonal and interannual variations of the two time-series correlate well.*

We agree and edited the formulation.

*Page 23, lines 14-17: the good agreement between the detrended time series cannot establish the absolute accuracy of the SEC, especially not in a level of a few cm/yr.*

We modified this to less absolute statements (which clearly depend on very regional conditions).

*Page 24, lines 1-2: These sentences appear to be incomplete: "This due to the mountain ranges just north of 72°S, which lead to many losses of lock of the measurements all the way across this part of the ice sheet. The same applies to the measurements at the APIS."*

Section modified.

**References**

[revised manuscript text omitted]

We arrive at consistent and seamless time series of gridded surface elevation differences with respect to a reference epoch (09/2010). They represent three-month temporal averages sampled every month and an effective spatial resolution of about The resulting monthly grids with a 10 km spatial resolution were obtained by smoothing with a moving window over three months and a spatial gaussian weighting with $2\sigma = 20\,\mathrm{km}$ sampled to a 10 grid. We evaluate our results and their estimated uncertainties by a comparison with independent in situ data sets, results from satellite gravimetry . We evaluate our results and results from regional atmospheric climate modeling their estimated uncertainties by a comparison with independent in-situ and airborne datasets, satellite gravimetry estimates, and regional climate model outputs. We illustrate that these time series of surface elevation change (SEC) allow to study geometry changes and derived mass changes of the AIS in unprecedented detail. For some examples as Pine Island Glacier , Totten Glacier, Shirase Glacier (Dronning Maud Land ) and Lake Vostok, we demonstrate the benefits of the long time series The recent elevation changes of Pine Island Glacier in West Antarctica, Totten Glacier in East Antarctica, and Shirase Glacier of Dronning Maud Land in East Antarctica are put in context with the extended time series from satellite altimetry. Finally, we calculate ice sheet mass balances from these data for the respectively covered regions. A comparison with independent data indicates a high consistency of the different data sets but reveals also remaining discrepancies.

[Figure]

**Figure 1.** Spatial and temporal coverage of the satellite altimetry data used in this study. The colors denote the maximum southern extent of the measurements (dark blue: 72°S, light blue: 81.5°S, orange: 86°S, red: 88°S) and thus the size of the respective polar gap.

While this paper gives some examples for new insights obtained from the presented multi-mission altimetry analysis, it can not fully exploit all potential applications. This will be the scope of future work with this data set.

**2 Data**

**2.1 Altimetry data used**

[revised manuscript text omitted]
 datasets shows that these estimates seem reasonable. In the comparison with the GNSS profiles, the relatively low differences, even in regions which

imply a higher uncertainty, are likely just incidental for the small sample of $\delta\Delta h$ validation data sets supports that the uncertainty estimates are reasonable. For $\Delta h_{ALT}$ we expect higher errors in coastal regions due to the increased uncertainty of the topographic correction in radar altimetry. A similar relation to topography is expected for $\Delta h_{OIB}$ due to the plane fit to the ATM point cloud. In contrast, the errors of the GNSS-derived $\Delta h_{KIN}$ are almost independent of topography. Instead, $\Delta h_{KIN}$ tends to be more uncertain on the plateau, where the soft snow causes large variations of the subsidence of the vehicles into the upper firn layers. The relatively low differences in $\delta\Delta h$ even in regions that imply a higher uncertainty, are likely just incidental for the small sample of validation data along the GNSS profiles.

In conclusion, this validation shows that remaining systematic biases (originating from satellite altimetry or the validation data) are on a centimeter level only and that our uncertainty estimate is realistic. However, we have to stress that less than a decimeter in the observed regions and that our uncertainty estimate is realistic. However, only altimetric SEC within the interval 2001-2016 can be validated in this way. For the earlier missions, no spatially extensive high precision in situ data are available to us publicly available.

**4.2 Firn model**

[revised manuscript text omitted]

grounding line (cf. Fig. 10b). Several smaller glaciers in Wilkes Land also a persistent thinning. We observe SEC rates of -26±10 at Denman Glacier (D), -41±19 at Frost Glacier (F) and -33±12 near Cook Ice Shelf. Rignot (2006) showed that the flow velocity of these glaciers, which are grounded well below sea level, was above the balance velocity for many years. In contrast, the western sector of the EAIS (Coats Land, DML and Enderby Land, basins J"-B) shows thickening over the last 25 years at rates of up to a decimeter per year.

Comparing the long-term elevation changes over 40 years (Fig. 9a) with those over 25 years shows the limitations of the early observations, but also the additional information they provide. There were only relatively few successful observations at the very margins but e.g. for Totten or Denman Glacier, they show similar rates at a distance of about 100 from the grounding line. In DML and Enderby Land (basins A-B in Fig. 9a), the 40 interval shows less positive rates, compared to 1992-2017. Until 2002, a large part of this region even experienced significant thinning (see time series in Fig. 10c and the maps for the sub-intervals in Fig. 9c-g). After that time, especially over the period 2007-2012 (Fig. 9i), this region shows a huge increase in elevation, which relates mainly to the accumulation events in 2009 and 2011. The sub-intervals in Fig. 9c-j demonstrate the effect of interannual snowfall variability on the elevation change rates over shorter time intervals. They show similar variations also in other regions, pointing out that accumulation events have a strong influence on interannual elevation changes over all parts of Antarctica (Horwath et al., 2012; Mémin et al., 2015).

**5.2 Ice sheet mass time series**

[Figure]

**Figure 12.** Mass change ($\Delta M [Gt]$) of the individual drainage basins north of 81.5°S from our combined altimetric time series (blue), GRACE (red) and SMB SMBA (orange). The error bars show the uncertainty estimate $\sigma_\Sigma$ of the altimetry data according to Sect. F.2. The gray color in the background displays the fraction of the area covered by altimetry (up to the top means 100%).

[Figure]

**Figure 13.** Mass change of subregions north of 72°S for several East Antarctic drainage basins from our combined altimetric time series (blue), GRACE (red) and  SMBA (orange). The error bars show the uncertainty estimate $\sigma_\Sigma$ of the altimetry data according to Sect. F.2. The gray color in the background displays the fraction of the area covered by altimetry (up to the top means 100%).

 The surface elevation time series are converted into ice mass changes in order to determine their effect on global sea level. In a first step, the SECs are corrected for uplift rates related to glacial isostatic adjustment (GIA) using coefficients from the IJ05_R2 model (Ivins et al., 2013). This GIA model predicts an uplift of $5 \, \mathrm{mm/yr}$ near the Antarctic

5 Peninsula and rates between -0.5 and $+2 \, \mathrm{mm/yr}$ in East Antarctica. Furthermore we  multiplied the SEC by a scaling factor $\alpha = 1.0205$ to account for elastic solid earth rebound effects (Groh et al., 2012).  The resulting ice sheet thickness changes are multiplied by each cell's area and a density according to a firn/ice mask (McMillan et al., 2014, 2016), depicted in Fig. S10, to obtain a mass change. In regions where ice dynamic processes are assumed to be dominating (e.g. in Amundsen Sea Embayment, Kamb Ice Stream or Totten Glacier), we use a

density of $917\,\mathrm{kg/m^3}$. Elsewhere, we apply the density of near-surface firn as modeled by Ligtenberg et al. (2011), using annual averages of accumulation, 10 m wind speed and surface temperature. We have chosen this straightforward method here, obtained from firn modeling using atmospheric forcing (Ligtenberg et al., 2011). We have chosen this straightforward and robust method here, instead of using modeled temporal variations instead of the firn layer (as e.g. Zwally et al., 2015; Kallenberg et al., 2017) in the volume-to mass conversion. This allows us to compare the time series

5    from altimetry with time series from SMB modeling using the modeled impact of the temporal variations of accumulation, melting and firn compaction on the firn layer (as e.g. Zwally et al., 2015; Kallenberg et al., 2017) in the volume-to-mass conversion. This allows us to keep our altimetry time series independent from the modeled variations in SMB, which is a prerequisite for the interpretation of the comparison of both data sets.

Cumulated mass anomalies over larger regions such as drainage basins or even the total AIS are obtained by summing up the results accordingly. Therefore, we used the basin

10    definitions by Rignot et al. (2011) (updated for Shepherd et al. (2018), see Figs. 9a and 14b). Cells containing no valid data after the gridding (as e.g. where not enough observations were available, in the polar gap or where rocks are predominant) are not considered here. Uncertainty estimates were obtained by propagating the respective We integrate our measurements over larger regions to calculate the cumulative mass anomalies for individual drainage basins and major Antarctic sectors (AIS, WAIS, EAIS, APIS). Our basin delineations are from Rignot et al. (2011), which have been updated for the second Ice Sheet Mass Balance Inter-comparison Exercise (IMBIE-2, Shepherd et al., 2018). Cells that

15    were masked out due to the predominance of rocks or that are considered unobserved after our gridding (due to the polar gap or a lack of valid observations) are not included in these sums. Uncertainty estimates are obtained by propagating the uncertainties of the SEC, the GIA and the firn density to the basin sums for each month (see Sect. F.2 for details). We also include an estimate for the effect of unobserved cells in To account for the lack of information due to unobserved cells, we also add a total estimate for the effect of these cells, based on trends from GRACE, to the error budget.

20    Figures 11a-d show time series for the entire AIS north of 81.5°S (i.e. covered by satellite altimetry since 1992), and the respective subregions EAIS, WAIS and the APIS. Similar time series for the single drainage basins over 1992-2017 are shown in Fig. 12. The full four decade time interval for the coastal areas of the EAIS For the coastal areas of the EAIS the full time interval since 1978 is shown in Fig. 13. These time series use the four decades time series use data north of 72°S only and, hence, provide a nearly consistent observational coverage since 1978. over the whole period. To support the interpretation and evaluate the temporal evo-

[revised manuscript text omitted]

**6    Discussion**

**6.1    Surface elevation changes**

The single-mission time series , obtained in Sect. 3.2, contain satellite-specific calibration biases as well as offsets due to the specific sampling characteristics of different sensor types. In order to form a consistent SEC time series, these biases needed to be determined and corrected. A comparison with in situ data showed that there are no significant offsets between elevation changes from our multi-mission altimetry data and the validation datasets. This comparison, however, could only validate our data in the interval 2001-2016. A quality control for the whole time span was performed by a comparison with a firn model. The correlation of the detrended data sets shows that especially for regions where the interannual variation is large (compared to the measurement noise of the altimeters) both time series agree very well. This comparison even provided independent estimates for the error of the early missions. The average differences between the detrended time series of the FDM and the SEC show that the observations of Geosat and even of Seasat agree with the model results within a few decimeters. For Combining all the single missions consistently, our SEC time series allow to analyze the long-term changes over the full time period of satellite altimetry observations. For 79% of the area of the AIS, this means a time span of 25 years. Over 25% of the ice sheet, largely in the coastal regions of East Antarctica, the time series can be extended back 40 years. Such long-term trends are significantly less affected by short-term variations in snowfall than a trend from a single mission. Furthermore, the period of observation of a single mission is short compared to climatic oscillations as reported e.g. by Mémin et al. (2015). Our extended time series helps to separate elevation change due to climate variations from potentially accelerating volume losses. Also Seasat and Geosat provide important information here. Due to the stability criteria in the calibration, we do not expect significant new insights on the East Antarctic plateau (even as regional variation still may be discernible as we used an ice-sheet-wide average in calibration). However, in the coastal regions of East Antarctica, with SECs of up to several meters w.r.t 2010 (see Fig. 6), this means that also the older data can be used

to calculate elevation change rates with an accuracy better than a centimeter per year (see Fig.S7a also the older data can contribute significant information to study elevation changes in a long-term context of 40 years (cf. the rates in Fig. 9 and their uncertainties in Fig. S9). Unfortunately, in coastal DML west of the ice divide A', the data of Seasat and Geosat are very noisy . This due to the mountain ranges just north of 72°S , which lead to many losses of lock of the measurements all the in these regions. They lead to many signal losses along the way across this part of the ice sheet. The same applies to the measurements at the APIS.

**6.2 Surface elevation changes**

The mean rates of elevation change in Fig. 9 show the regions which experience a significant thinning (Amundsen Sea Embayment, Totten Glacier) or thickening (Kamb Ice Stream) which was already reported by previous publications (e.g. Wingham et al., 2006b; Flament and Rémy, 2012; Helm et al., 2014; Zwally et al., 2015). By combining all the single missions consistently we analyze long-term changes over the full time period covered. For 79% of the area of the AIS, this means a time span of 25 years. For 25%, mainly the coastal regions of East Antarctica, even 40 years are covered. We assume that these long-term trends are significantly less affected by short-term variations in snowfall than a trend from a single mission.

[revised manuscript text omitted]

---

## Author Response (AR3)

**Authors response to referee report #1**

We would like to thank the referee for the help to further improve the clarity of the manuscript and for the very helpful suggestions to improve grammar and wording. All the comments during the whole review process lead to a steady enhancement of the quality of the presentation of our results.

**General comments**

• *The comparisons with GNSS and NASA Operation IceBridge in the supplementary material are helpful. Could use in the main text to separate the OIB data by region and replace the "zoomed out" figure.*

The main text has been modified in accordance with some of the comments below. However, concerning the figure, we think that it is important to present the full extent of validation data before focusing on some regions. Otherwise, the reader could think we are hiding something.

• *The paper is still difficult to follow at times. The paper could be reworked for conciseness and to better separate the Results and Discussion sections.*

In accordance with the detailed comments, several paragraphs of the respective sections has been modified towards more clarity. Also the discussion of our results in comparison with Shepherd et al. (2018) has been moved.

• *Significant figures (e.g. 1.8 ± 0.3 instead of 1.80 ± 0.31)*

We think it is common practice to present results at two significant digits of their standard deviation.

• *Could list some of the mass changes observed in terms of sea level equivalents.*

Done.

**Line-by-line comments**

*P2, L1 Awkwardly constructed sentence. "As a consequence, results derived from a single mission and mean linear rates reported from a single mission have limited significance in characterizing the long-term evolution of the ice sheet"*

Changed.

*P2, L5 could use "have been" instead of "were"*

Done.

*P2, L10–11 I'd recommend something like "Paolo et al. (2016) cross-calibrated ERS-1, ERS-2 and Envisat on each grid cell using overlapping epochs. We use a very similar approach for these missions and data from the low-resolution mode of CryoSat-2." Could also reference Adusumilli et al. (2018) here.*

The section has been modified to better point out whats new in our approach.

*P3, L20 remove "as well"*

Changed.

*P4, L22 "the coarse POCA location"*

Done.

*P4, L31–33 possibly move "especially with a low threshold of 10%" to the end of the sentence.*

Changed.

*P4, L34 possibly add a comma after amplitude and remove the comma after level.*

Done.

*P5, L7 use "and GNSS profiles" instead of "or GNSS profiles"*

Done.

*P5, L28 possibly "which helps confirm the findings of"*

Changed.

*P6, L4 remove "already"*

Done.

*P7, L29 remove "furthermore"*
Done.

*P8, L2 remove "like the first group of authors did"*
Done.

*P8, L8–9 possibly "prefer the simplest viable model in order to keep the number of parameters small as compared to the number of observations"*
Changed.

*P8, L16 change to "(i.e. PLRA, SARin and laser altimetry), the effective surface slope may also differ"*
Done.

*P9, L13 possibly "to exclude any res i that exceed five times"*
Done.

*P12, L17 remove "thus"*
Done.

*P13, L7–8 Awkwardly constructed sentence*
Modified.

*P13, L10–13 Awkwardly constructed sentence. Also remove "hence"*
Modified.

*P13, L15 PPP processing of the GNSS data?*
All details concerning the processing of the profiles can be found in the referenced paper.

*P13, L16–19 possibly "The ground-based GNSS profiles were completed between 2001 and 2015 on traverse vehicles of the Russian Antarctic Expedition. Most of the profiles cover more than 1000km"*
We modified the tense of this sentence but prefer 'observed' instead of 'completed' as 'completed' might indicate that no further profile measurements would be necessary, which is not the case.

*P14, L1 possibly "been used due to poorly determined antenna height offsets."*
Changed.

*P14, L7 remove "nevertheless"*
Done.

*P14, L8 add a comma after terrain*
Done.

*Figure 7 results appear to be saturated in the Operation IceBridge portion of the plot.*
The existence of saturated values is indicated by the arrows at the ends of the color bar. In our opinion, a saturation above $\pm 1\,\mathrm{m}$ is acceptable, as we know (and discuss) that radar altimetry is difficult at the margins. In contrast, we chose the color scale in order to make the smaller differences at a level of some decimeters still visible.
*Could possibly rasterize the OIB data to be comparable spatially with the merged product.*
We already thought about rasterizing the data, however, especially at PIG, this would average out the real spread of the results. While one dh/dt-profile shows significant positive $\delta\Delta h$, compared to our data, another one very close nearby shows the opposite sign. Rasterizing (i.e. averaging) could, hence, make the $\delta\Delta h$ appear smaller than they are.
*The discrepancies compared with OIB in uncrevassed areas is disconcerting.*
We modified Fig. S6, showing PIG and the area around the South Pole now. We discuss, that the large variations in the flat tributaries of PIG may lead to the large spread, while in other flat regions, the differences are significantly smaller.

*P15, L13 The planes used in the Level-4 ATM product are ∼100m across. While the data is affected by slope errors, the larger sources of uncertainty are roughness and crevassing.*
Modified accordingly.

*P15, L14 How often are the vehicle track depths measured when acquiring the GNSS data?*
Usually whenever it was possible, i.e. at each stop of the vehicle. These details concerning the

processing of the validation data can be found in the referenced paper.

*Section 4.2 Could reference Helsen et al. (2008) in this section.*
Helsen et al. (2008) compare linear trends from RACMO with surface elevation change rates from ERS-2. They do not have temporal variation in their altimetric results. In contrast, we remove the rates and compare the remaining temporal variation.

*P15, L27 Remove "furthermore"*
This is an additional fact besides the previous one. Hence, we think that "furthermore" is necessary.

*P16, L1 could add something like "and where large accelerations in ice velocity are observed, such as Pine Island Glacier"*
Modified accordingly.

*P18, L8–9 perhaps "For Totten and Denman Glaciers, the 40-year rates at a distance of approximately 100km inland from the grounding line are similar to the rates over the 1992–2017 interval, which indicates a persistent rate of thinning."*
Changed.

*P18, L9–10 possibly "With our merged time series, elevation change rates can be derived for any sub-interval in time for which there is data available, as shown in Figures 9c-j."*
The sentence has been edited.

*P18, L12–13 Somewhat awkwardly constructed sentence*
Sentence edited.

*P18, L14 for some regions*
This is already indicated by "up to"

*P18, L15–29 why were these regions chosen?*
Some arguments added.

*P25, L9 This sentence could be rewritten "Also Seasat and Geosat provide important information here"*
Modified.

*P27, L6–14 Paragraph seems disconnected and difficult to follow.*
The paragraph has been rewritten. Also the next beginning of the next paragraph was modified to form a better connection.

*P27, L13 remove "nevertheless"*
We think, this is necessary.

*P27, L19 could use "in the peninsula" versus "here"*
Changed.

[revised manuscript text omitted]